# Regenerative neurogenic response from glia requires insulin-driven neuron-glia communication

Neale J Harrison[1†‡], Elizabeth Connolly[1†], Alicia Gascón Gubieda[1§], Zidan Yang[1#], Benjamin Altenhein[2], Maria Losada Perez[3], Marta Moreira[1], Jun Sun[1], Alicia Hidalgo[1*]

[1]Structural Plasticity & Regeneration Group, School of Biosciences, University of Birmingham, Birmingham, United Kingdom; [2]Institute of Zoology, University of Cologne, Cologne, Germany; [3]Instituto Cajal, Consejo Superior de Investigaciones Cientificas (CSIC), Madrid, Spain

*For correspondence:
a.hidalgo@bham.ac.uk

†These authors contributed equally to this work

Present address: ‡ Tomlinson Lab, School of Biosciences, University of Birmingham, Birmingham, United Kingdom; § Institute for Cell and Molecular Biosciences, Newcastle University, Newcastle upon Tyne, United Kingdom; # Max Planck Florida Institute for Neuroscience, Jupiter, United States

Competing interests: The authors declare that no competing interests exist.

**Abstract** Understanding how injury to the central nervous system induces de novo neurogenesis in animals would help promote regeneration in humans. Regenerative neurogenesis could originate from glia and glial neuron-glia antigen-2 (NG2) may sense injury-induced neuronal signals, but these are unknown. Here, we used *Drosophila* to search for genes functionally related to the *NG2* homologue *kon-tiki (kon)*, and identified *Islet Antigen-2 (Ia-2)*, required in neurons for insulin secretion. Both loss and over-expression of *ia-2* induced neural stem cell gene expression, injury increased *ia-2* expression and induced ectopic neural stem cells. Using genetic analysis and lineage tracing, we demonstrate that Ia-2 and Kon regulate *Drosophila* insulin-like peptide 6 (Dilp-6) to induce glial proliferation and neural stem cells from glia. Ectopic neural stem cells can divide, and limited de novo neurogenesis could be traced back to glial cells. Altogether, Ia-2 and Dilp-6 drive a neuron-glia relay that restores glia and reprogrammes glia into neural stem cells for regeneration.

## Introduction

The central nervous system (CNS) can regenerate after injury in some animals, and this involves de novo neurogenesis (*Tanaka and Ferretti, 2009*). Newly formed neurons integrate into functional neural circuits, enabling the recovery of function and behaviour, which is how CNS regeneration is measured (*Tanaka and Ferretti, 2009*). The human CNS does not regenerate after injury. However, in principle it could, as we continue to produce new neurons throughout life that integrate into functional circuits (*Tanaka and Ferretti, 2009*; *Gage, 2019*). Through understanding the molecular mechanisms underlying natural regenerative neurogenesis in animals, we might be able to provoke de novo neurogenesis in the human CNS to promote regeneration after damage or neurodegenerative diseases. Regenerative neurogenesis across animals may reflect an ancestral, evolutionarily conserved genetic mechanism, which manifests itself to various degrees in regenerating and non-regenerating animals (*Tanaka and Ferretti, 2009*). Accordingly, it may be possible to discover molecular mechanisms of injury-induced neurogenesis in the fruit-fly *Drosophila*, which is a powerful genetic model organism.

Regenerative neurogenesis could occur through activation of quiescent neural stem cells, de-differentiation of neurons or glia, or direct conversion of glia to neurons (*Tanaka and Ferretti, 2009*; *Falk and Götz, 2017*). Across many regenerating animals, new neurons originate mostly from glial cells (*Tanaka and Ferretti, 2009*; *Falk and Götz, 2017*). In the mammalian CNS, radial glial cells behave like neural stem cells to produce neurons during development. Remarkably, whereas NG2-glia (also known as oligodendrocyte progenitor cells, OPCs) produce only glia (oligodendrocytes

and astrocytes) in development, they can also produce neurons in the adult and upon injury (*Dimou and Götz, 2014*; *Falk and Götz, 2017*; *Valny et al., 2017*; *Du et al., 2021*) – although this remains controversial. Discovering the molecular mechanisms of a neurogenic response of glia is of paramount urgency.

NG2-glia are progenitor cells in the adult human brain, constituting 5–10% of total CNS cells, and remain proliferative throughout life (*Dimou and Götz, 2014*). In development, NG2-glia are progenitors of astrocytes, OPCs, and oligodendrocytes, but postnatally and upon injury they can also produce neurons (*Dimou and Götz, 2014*; *Torper et al., 2015*; *Falk and Götz, 2017*; *Valny et al., 2017*; *Du et al., 2021*). They can also be directly reprogrammed into neurons that integrate into functional circuits (*Torper et al., 2015*; *Pereira et al., 2017*). The diversity and functions of NG2-glia are not yet fully understood, but they are particularly close to neurons. They receive and respond to action potentials generating calcium signals, they monitor and modulate the state of neural circuits by regulating channels and secreting chondroitin sulphate proteoglycan perineural nets, and they also induce their own proliferation to generate more NG2-glia, astrocytes that sustain neuronal physiology, and oligodendrocytes that enwrap axons (*Dimou and Götz, 2014*; *Sakry and Trotter, 2016*; *Sun et al., 2016*; *Du et al., 2021*). NG2-glia have key roles in brain plasticity, homeostasis, and repair in close interaction with neurons (*Dimou and Götz, 2014*; *Sakry and Trotter, 2016*; *Du et al., 2021*), but to what extent this depends on the *NG2* gene and protein, is not known.

*NG2* (also known as *chondroitin sulphate proteoglycan 4, CSPG4)* is expressed by NG2-glia and pericytes, but not by oligodendrocytes, neurons, or astrocytes (*Cahoy et al., 2008*). NG2 is a transmembrane protein that can be cleaved upon neuronal stimulation to release a large secreted extracellular domain and an intracellular domain (*Sakry et al., 2014*; *Sakry and Trotter, 2016*). The intracellular domain (ICD, NG2$^{ICD}$) is mostly cytoplasmic, and it induces protein translation and cell cycle progression (*Nayak et al., 2018*). NG2$^{ICD}$ lacks a DNA binding domain and therefore does not function as a transcription factor, but it has a nuclear WW4 domain and nuclear localisation signals and can regulate gene expression (*Sakry et al., 2015*; *Sakry and Trotter, 2016*; *Nayak et al., 2018*). It is thought that NG2 functions as a receptor, triggering nuclear signalling in response to ligands or partners (*Sakry et al., 2014*; *Sakry and Trotter, 2016*). NG2 protein is abundant in proliferating NG2-glia and glioma (*Sakry et al., 2015*; *Sakry and Trotter, 2016*; *Nayak et al., 2018*). It is also required for OPC proliferation and migration in development and in response to injury (*Kucharova and Stallcup, 2010*; *Kucharova et al., 2011*; *Binamé et al., 2013*). Given the close relationship of NG2-glia with neurons, it is anticipated that key partners of NG2 are produced from neurons, but these remain largely unknown.

The fruit-fly *Drosophila* is particularly powerful for discovering novel molecular mechanisms. The *Drosophila NG2* homologue is called *kon-tiki (kon)* or *perdido* (*Estrada et al., 2007*; *Schnorrer et al., 2007*; *Pérez-Moreno et al., 2017*). Kon functions in glia, promotes glial proliferation and glial cell fate determination in development and upon injury, and promotes glial regeneration and CNS injury repair (*Losada-Perez et al., 2016*). Kon works in concert with the receptor Notch and the transcription factor Prospero (Pros) to drive the glial regenerative response to CNS injury (*Kato et al., 2011*; *Losada-Perez et al., 2016*). It is normally found in low levels in the larval CNS, but injury induces a Notch-dependent increase in *kon* expression in glia (*Losada-Perez et al., 2016*). Together, Notch signalling and Kon induce glial proliferation. Kon also initiates neuropile glial differentiation and *pros* expression, and Pros maintains glial cell differentiation (*Griffiths and Hidalgo, 2004*; *Kato et al., 2011*; *Losada-Perez et al., 2016*). This glial regenerative response to injury is homeostatic and time-limited, as two negative feedback loops halt it: Kon represses Notch, and Pros represses *kon* expression, preventing further cell division (*Kato et al., 2011*; *Losada-Perez et al., 2016*). The relationship between these genes is also conserved in the mouse, where the homologue of *pros, Prox1*, is expressed together with *Notch1* in NG2-glia (*Kato et al., 2015*). Following cell division, Prox1 represses NG2-glia proliferation and promotes oligodendrocyte differentiation (*Kato et al., 2015*). Together, Notch, Kon, and Pros form a homeostatic gene network that sustains neuropile glial integrity throughout life and drives glial regeneration upon injury (*Hidalgo and Logan, 2017*; *Kato et al., 2018*). As Kon is upregulated upon injury and provokes glial proliferation and differentiation, it is the key driver of the glial regenerative response to CNS injury.

A critical missing link to understand CNS regeneration was the identification of neuronal partners of glial NG2/Kon that could induce regenerative neurogenesis. We had observed that injury to the *Drosophila* larval CNS also resulted in spontaneous, yet incomplete, repair of the axonal neuropile

(*Kato et al., 2011*). This strongly suggested that injury might also induce neuronal events, such as axonal regrowth or generation of new neurons. Thus, we asked whether Kon may interact with neuronal factors that could contribute to regenerative neurogenesis after injury. Here, we report that relay of insulin signalling involving neuronal Ia-2 and glial Kon drives in vivo reprogramming of neuropile glia into neural stem cells.

## Results

### Genetic analysis reveals Ia-2 is a key neuronal factor interacting with Kon

To search for neuronal factors that might interact with glial *kon*, we carried out genetic screens aimed at identifying genes expressed in neurons that had non-autonomous effects on glia. We exploited the fact that overexpression of *kon* elongates the larval ventral nerve cord (VNC) (*Losada-Perez et al., 2016*), and tested whether RNAi knock-down of candidate genes in neurons or glia could rescue this phenotype (*Figure 1—figure supplements 1* and *2*). To validate the approach, we first tested genes predicted or known to interact with *kon* and/or *NG2* (*Schnorrer et al., 2007*; *Pérez-Moreno et al., 2017*). Indeed, knock-down of known interactors, such as *integrins* (*Pérez-Moreno et al., 2017*), factors involved in Notch signalling (e.g. *Mtm, Akap200*), secretases (i.e. *kuz, kul*) that cleave both Notch and NG2/Kon (*Sakry and Trotter, 2016*), and phosphatases *Prl1* and *Ptp99A* (*Song et al., 2012*), all rescued the phenotype, validating the approach (*Figure 1—figure supplements 1* and *2*). We tested knock-down of other genes encoding phosphatases and transmembrane proteins expressed in neurons. Knocking-down phosphatases *ptp99A, ptp69D,* and *ptp4E* from neurons rescued the phenotype, but most prominent was knock-down of phosphatase *lar,* a negative regulator of insulin signalling (*Figure 1—figure supplement 2A–D*). Notably, knock-down of other insulin related factors including *Akt* and *ia-2* also caused some rescue (*Figure 1—figure supplement 2A–D*). However, multiple genes can affect VNC length, and these rescue phenotypes may not necessarily reflect specific gene interactions. Thus, we next asked whether altering *kon* function affected the expression of a group of genes selected from the above screens. Kon can influence gene expression, as *kon* mutations cause loss of glial gene expression (*Losada-Perez et al., 2016*). Using quantitative real-time reverse transcription PCR (qRT-PCR) on dissected larval CNSs, we found that *kon* knock-down in neurons (with $kon^{c452}$, *elavGAL4>UAS-konRNAi*) or glia (with $kon^{c452}$, *repoGAL4>UAS-konRNAi*) had no effect on the expression of most phosphatases, including *lar,* or other tested genes. By contrast, it resulted in an approximately three fold increase in *ia-2* mRNA levels (*Figure 1—figure supplement 3A*). Conversely, overexpression of full-length *kon* in either neurons or glia downregulated *ia-2* mRNA levels by 25% (*Figure 1—figure supplement 3B*). We validated these results by increasing the repeats of the most promising subset of genes (*Figure 1—figure supplement 3C,D*), and this confirmed the strongest effect of *kon* loss of function (LOF) and gain of function (GOF) on *ia-2* (*Figure 1A*). Accordingly, Kon function in glia prevents *ia-2* expression. Next, we asked whether knock-down or overexpression of *ia-2* in neurons (with *elavGAL4*) had any effect on *kon* mRNA levels, but none did (*Figure 1B*). However, overexpression of *ia-2* in glia (with *repoGAL4>ia-2[GS11438]*) decreased *kon* mRNA levels (*Figure 1B*). As Kon functions in glia (*Losada-Perez et al., 2016*), these data indicated that *kon* and *ia-2* restrict each other's expression to glia or neurons, respectively, and/or that Ia-2 is restricted to neurons. Either way, these data showed that *ia-2* and *kon* interact genetically.

Our genetic and qRT-PCR based screens had identified genetic interactions between *kon* and *lar,* and *Akt* and *ia-2.* LAR is involved in neuronal axon guidance and is responsible for de-phosphorylating, and thus inactivating insulin receptor signalling (*Mooney et al., 1997*; *Wills et al., 1999*). Akt is a key effector of insulin receptor signalling downstream (*van der Heide et al., 2006*). Ia-2 is a highly evolutionarily conserved phosphatase-dead transmembrane protein phosphatase required in dense core vesicles for the secretion of insulin, insulin-related factor-1 (IGF-1) and neurotransmitters; it also has synaptic functions and influences behaviour and learning (*Cai et al., 2001*; *Harashima et al., 2005*; *Hu et al., 2005*; *Henquin et al., 2008*; *Cai et al., 2009*; *Nishimura et al., 2010*; *Cai et al., 2011*; *Carmona et al., 2014*). Rather unexpectedly, our findings suggested that Kon is involved in insulin signalling.

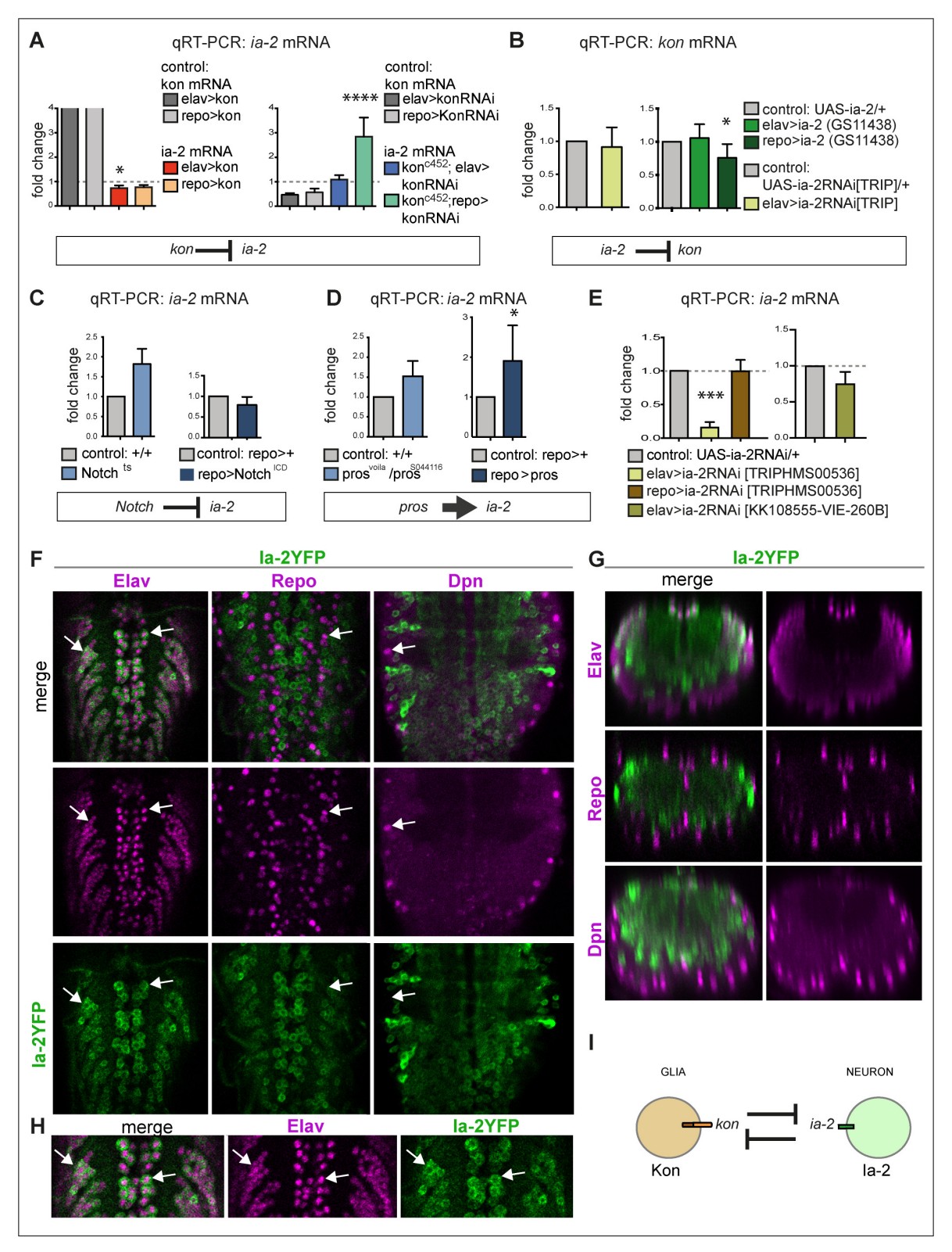

**Figure 1.** *ia-2* interacts genetically with *kon, Notch,* and *pros.* (**A**) Quantitative real-time PCR (qRT-PCR) showing that gain of *kon* function reduced *ia-2* mRNA levels by 25% (one-way ANOVA p=0.045), whereas loss of *kon* function in glia caused practically a threefold increase in *ia-2* mRNA levels (*genotype: kon^c452^/UASkonRNAi; repoGAL4/+;* one-way ANOVA p<0.0001). Post hoc Dunnett's test multiple comparisons to control. N = 4 replicates. (**B**) qRT-PCR showing that overexpression of *ia-2* in glia downregulated *kon* mRNA levels. Left: Unpaired Student's t-test with Welch correction p=0.457. *Figure 1 continued on next page*

*Figure 1 continued*

Right: one-way ANOVA p<0.045, post hoc Dunnett's test multiple comparisons to control. N = 4–6 replicates. (C) Ia-2 is functionally related to Notch: qRT-PCR showing that *ia-2* mRNA levels increased in $N^{ts}$ mutant larvae at the restrictive temperature of 25°C. Unpaired Student's t-test with Welch correction. Left: p=0.4123; Right: p=0.2182. N = 3 replicates. (D) *ia-2* is functionally related to *pros*: qRT-PCR showing that overexpression of *pros* in glia increased *ia-2* mRNA levels by twofold. Unpaired Student's t-test with Welch correction. Left: p=0.1368; Right: p=0.0428. N = 3 replicates. (E) qRT-PCR showing that *UAS-ia-2* RNAi[TRIPHMS00536] knock-down in neurons (with *elavGAL4*) lowered *ia-2* mRNA levels to 20%, whereas in glia it has no effect, meaning that *ia-2* is expressed in neurons. A second *UAS-ia-2RNAi[KK108555-VIE-260B]* line lowered mRNA levels by 25%. One-way ANOVA p=0.0004, post hoc multiple comparisons to control Dunnett's test. N = 3 replicates. (F–H) Fusion protein Ia-2YFP revealed expression exclusively in neurons, as all Ia-2YFP+ cells were also Elav+, but Repo⁻ and Dpn⁻. Genotype: ia-2[CPTI100013]. N = 4–16 larval ventral nerve cords (VNCs). (I) Illustration showing that *kon* and *ia-2* functions are restricted to glia and neurons, respectively, and they mutually exclude each other. (G) Transverse views; (F and H) horizontal views; (H) higher magnification views. With more than two sample types, asterisks indicate multiple comparison post hoc tests to controls: *p<0.05, **p<0.01, ***p<0.001, ****p<0.0001. For full genotypes and further statistical analysis details, see *Supplementary file 1*.

The online version of this article includes the following figure supplement(s) for figure 1:

**Figure supplement 1.** Modifier genetic screens identify genes interacting with *kon*.
**Figure supplement 2.** Modifier candidate genetic screens identify genes encoding transmembrane phosphatases and insulin signalling factors as interacting with *kon*.
**Figure supplement 3.** Loss and gain of *kon* function prominently affected *ia-2* expression.

To ask whether and how Ia-2 might relate to the Kon-Notch-Pros glial regenerative gene network, we tested whether LOF or GOF for *Notch* or *pros* might affect *ia-2* expression. With qRT-PCR on dissected larval CNSs, we found that *Notch^{ts}* mutants had an almost twofold increase in *ia-2* expression, whereas *Notch^{ICD}* overexpression in glia (*repoGAL4>Notch^{ICD}*) caused no significant effect (*Figure 1C*). Like Kon, Notch also functions in glia (*Griffiths and Hidalgo, 2004*; *Kato et al., 2011*; *Losada-Perez et al., 2016*), thus the genetic inference is that *ia-2* expression in glia is prevented by Notch. Ia-2 mRNA levels increased slightly (albeit not significantly) in *pros* mutant larvae, and most prominently, when *pros* was overexpressed in glia (*Figure 1D*). The loss of function phenotype could be indirect: in glial cells Pros and Notch depend on each other (*Griffiths and Hidalgo, 2004*; *Kato et al., 2011*), thus loss of *pros* causes the downregulation of *Notch*, which would increase *ia-2* expression. Instead, the stronger effect of *pros* GOF on *ia-2* indicated that Pros could directly regulate *ia-2* expression. Importantly, Pros is a transcription factor found in glia, type I and II neuroblasts, ganglion mother cells (GMCs), and some neurons (*Bayraktar et al., 2010*). Thus, Pros could regulate *ia-2* expression in any of these cell types. Most importantly, these data meant that *ia-2* participates in the *kon, Notch, pros* gene network that drives the regenerative response to CNS injury.

The above data suggested that *ia-2* expression is normally excluded from glia. To test what cells express *ia-2*, we knocked-down *ia-2* with RNAi in either neurons or glia and measured *ia-2* mRNA levels with qRT-PCR in dissected larval CNSs. *ia-2-RNAi* knock-down in glia (*repoGAL4>UASia-2RNAi^{TRIPHMS00536}*) had no effect, however knock-down in neurons (*elavGAL4>UASia-2RNAi^{TRIPHMS00536}*) downregulated *ia-2* transcripts to about 20% of wild-type levels (*Figure 1E*). A second *UAS-ia-2 RNAi* line (line *UAS-ia-2RNAi^{KK108555-VIE-260B}*) had a milder effect, but still reduced *ia-2* expression by 25% (*Figure 1E*). These data meant that *ia-2* is expressed in neurons. To visualise *ia-2* expression in vivo, we used a transgenic protein fusion of Ia-2 to yellow fluorescent protein (YFP), Ia-2YFP^{CPTI100013} (*Lowe et al., 2014*; *Lye et al., 2014*), from now on called Ia-2YFP. Ia-2YFP+ cells did not have the glial marker Repo (*Figure 1F,G*). They did not have Deadpan (Dpn) either (*Figure 1F, G*), which is the general marker for neuroblasts as well as intermediate neural progenitors of type II neuroblast lineages (*Boone and Doe, 2008*). All Ia-2YFP+ cells had the pan-neuronal marker Elav (*Figure 1F–H*). This demonstrated that *ia-2* is expressed exclusively in neurons.

Altogether, these data showed that Ia-2 and Kon function within the regenerative gene network and are restricted to neurons and glia, respectively (*Figure 1I*).

## Alterations in Ia-2 levels induced ectopic cells expressing the neural stem cell marker Dpn

Next, we carried out a functional analysis of *ia-2* in the CNS. As *kon* knock-down increased *ia-2* mRNA levels, we sought to verify this using Ia-2YFP. Ia-2YFP+ appeared undistinguishable from wild type when *kon* was knocked-down in glia (*kon^{c452}/ia-2YFP; repoGAL4>kon-RNAi*) (*Figure 2A*). However, as Ia-2YFP is normally in all neurons, a potential effect could have been missed. Thus, we

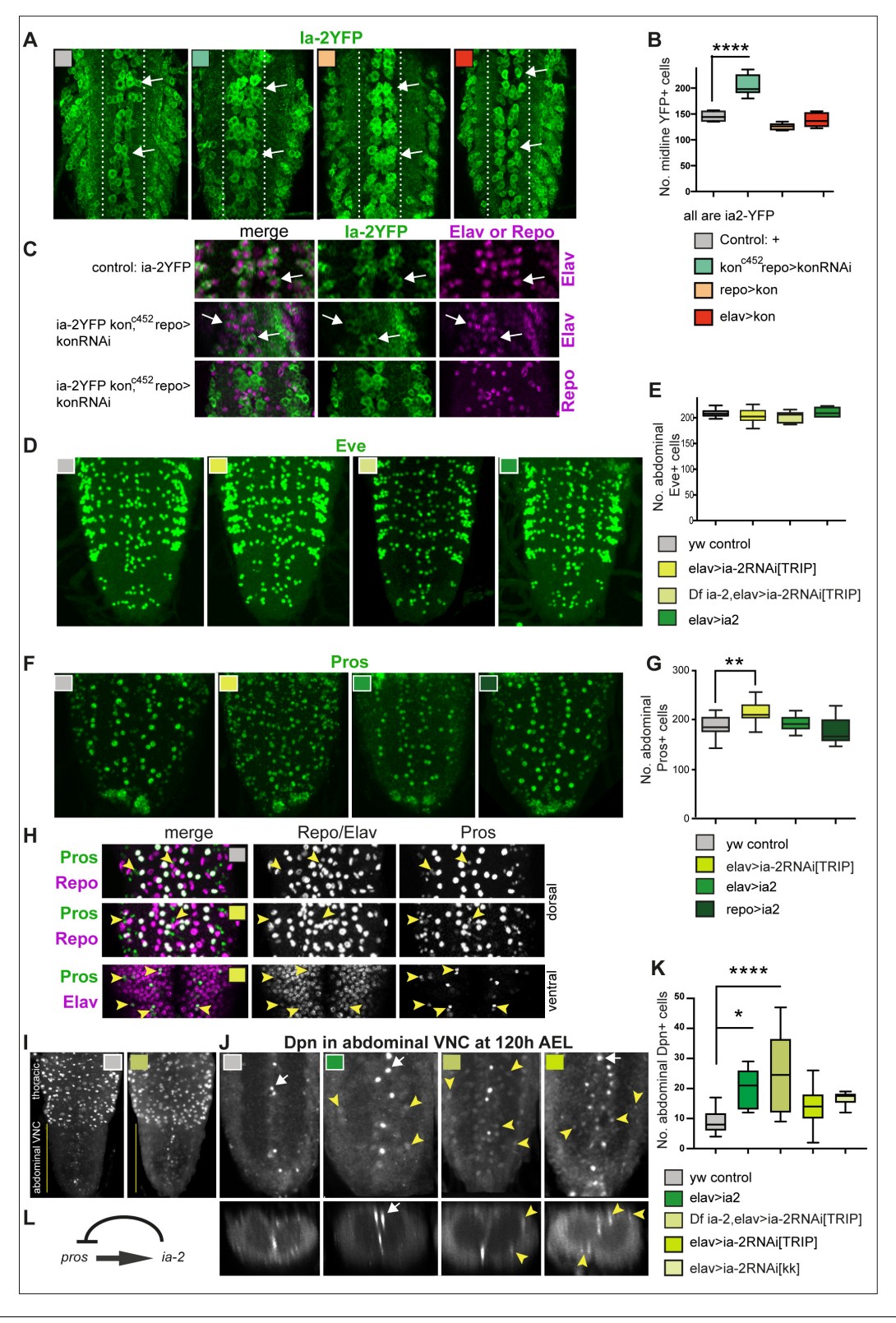

**Figure 2.** *ia-2* influences neural cell fate non-autonomously. (**A and B**) Loss of *kon* function in glia (*kon^c452^/UASkonRNAi; repoGAL4/+*) increased the number of Ia-2YFP+ cells along the midline. One-way ANOVA p<0.0001, post hoc Tukey's test. N = 5–8 ventral nerve cords (VNCs). (**C**) The ectopic Ia-2YFP+ cells in *kon* loss of function were Elav+ and not Repo+. N = 5–7 VNCs. (**D and E**) Neither loss nor gain *of ia-2* function affected the number of Eve+ neurons. One-way ANOVA p=0. 2374. N = 7–12 VNCs. (**F and G**) Loss of *ia-2* function (*elavGAL4>UASia-2RNAi[TRIPHSM00536]*) increased Pros+

*Figure 2 continued on next page*

*Figure 2 continued*

cell number, and supernumerary cells were small. Kruskal–Wallis ANOVA p=0.0003, post hoc Dunnett's test. N = 8–26 VNCs. (H) Small Pros+ cells in *ia-2* knock-down (genotype as in **F and G**) did not have the glial marker Repo, but could have the neuronal marker Elav. (**I–K**) Dpn+ cells visualised at 120 hr AEL, after developmental neuroblasts have disappeared. (I) Dpn signal in thorax was strong as normal, and in abdomen lower Dpn was found ectopically in *ia-2* loss or gain of function. (**J and K**) Both loss and gain of *ia-2* function increased the number of abdominal Dpn+ cells, which also were in ectopic locations. (**L**) Genetic inference: *ia-2* negatively regulates *pros*, most likely non-autonomously. All images are horizontal views, except for **I** (bottom row) which are transverse views. One-way ANOVA p=0.0002, post hoc Dunnett. N = 7–15. Data shown in box-plot s: box represents 50% values around with median, and whiskers 25% top and bottom values. Asterisks indicate multiple comparison post hoc tests to a fixed control: *p<0.05, **p<0.01, ***p<0.001, ****p<0.0001. For further statistical analysis details, see ***Supplementary file 1***.

The online version of this article includes the following figure supplement(s) for figure 2:

**Figure supplement 1.** Alterations in *ia-2* levels cause no obvious neuronal phenotypes.

focused on the midline, where a limited number of dorsal Ia-2YFP+ neurons can be counted. Indeed, *kon* loss of function in glia increased the number of Ia-2YFP+ cells along the midline (***Figure 2A,B***). The ectopic ia-2YFP cells had the pan-neuronal marker Elav and did not have the glial marker Repo (***Figure 2C***), meaning they were neurons. Ia-2YFP+ midline cells were unaffected by *kon* overexpression in either neurons or glia (***Figure 2A,B***, *elavGAL4>kon* and *repoGAL4>kon*). Thus, in the absence of *kon*, ectopic Ia-2YFP+ neurons were found at the midline. Loss of *kon* function prevents glial differentiation (***Losada-Perez et al., 2016***) and could result in more Ia-2YFP+ neurons also in other locations. The increase in neurons could explain why *ia-2* mRNA levels increased with *kon* loss of function (see ***Figure 1A***). However, the mRNA levels for *ptp99A*, −*69F*, and *10D* (***Figure 1—figure supplement 3***), also known to function in neurons, were not increased. Either way, these data confirmed that Kon and Ia-2 are mutually exclusive in glia and neurons, respectively.

To ask what function Ia-2 might have in neurons, we altered *ia-2* expression and visualised the effect using standard neuronal markers. *ia-2* knock-down in neurons (*elavGAL4>ia-2RNAi-*[TRIPHMS00536]) had no detectable effects on FasII or BP102 (***Figure 2—figure supplement 1A,B***). It did not change Eve+ neuron number either (***Figure 2D,E***). As Pros activates *ia-2* expression (***Figure 1D***), we asked whether *ia-2* might affect Pros. Overexpression of *ia-2* in either neurons or glia had no effect on Pros+ cells (***Figure 2F,G***). By contrast, *ia-2* knock-down in neurons (*elavGAL4>ia-2RNAi* [TRIPHMS00536]) increased the number of small Pros+ cells (***Figure 2F,G***). GMCs and neurons are generally smaller than glia. In fact, these small Pros+ cells lacked the glial marker Repo, but had the neuronal marker Elav (***Figure 2H***). Altogether, data showed that reduced *ia-2* function caused ectopic Pros+ neurons, and potentially also GMCs. Genetic inference is that *ia-2* represses *pros* (***Figure 2L***).

Pros is normally found in the nuclei of some neurons, some glia (astrocytes and midline), and all GMCs. In neuroblasts, Pros is cytoplasmic, and nuclear translocation of Pros drives the switch from neural stem cell (neuroblast) to progenitor (GMC) cell fate (***Choksi et al., 2006***). One of the target genes repressed by Pros is the marker common to all *Drosophila* neural stem cells, Dpn (***Choksi et al., 2006***). Thus, we asked whether altering *ia-2* function might affect Dpn. Dpn was visualised in normally high levels, in thoracic neuroblasts of third instar larvae (***Figure 2I***). Both *ia-2* GOF (*elav>ia-2*) and LOF (*Df(2L)ED7733/+; elav>ia-2RNAi* [TRIPHMS00536]), and *elavGAL4* knock-down using two different RNAi lines, *UAS-ia-2RNAi* [TRIPHMS00536] and *UAS-ia-2RNA* [KK108555-VIE-260B], in neurons increased the total number of abdominal VNC Dpn+ cells (***Figure 2I–K***). Supernumerary abdominal Dpn+ cells were located at ectopic positions, along the midline and around the neuropile, positions normally occupied by glial cells (***Figure 2J***). The ectopic Dpn+ cells were distinct from normal abdominal larval neural stem cells, which are ventro-lateral and further away from the neuropile. Furthermore, they were visualised 120 hr after egg laying (AEL), after the disappearance of developmental abdominal neural stem cells. Dpn levels in these ectopic abdominal cells were lower than in thoracic Dpn+ neuroblasts (compare with ***Figure 2I***). Altogether, alterations in the levels of neuronal Ia-2 induced neural stem cell marker expression ectopically.

These data showed that interference with normal neuronal Ia-2 levels upregulated Pros (a marker for progenitor cells, neurons, and glia) and Dpn (the general neural stem cell marker). This effect was non-autonomous, as neurons themselves seemed unaffected. As Ia-2 and Kon are functionally related but confined to either neurons or glia, respectively, this suggested that communication between neurons and glia was involved in inducing an ectopic neural stem cell state.

## Injury induces *ia-2* expression and a regenerative neurogenic response

Above data had shown that altering *ia-2* levels upregulated the neural stem cell marker Dpn (*Figure 2I–K*). Since CNS injury induced the upregulation of *kon* expression (*Losada-Perez et al., 2016*), we asked whether injury might affect *ia-2* expression and, consequently, induce a neurogenic response. To this end, crush injury was carried out at 74–76 hr AEL in early third-instar larval VNCs labelled with the endoplasmic reticulum GFP marker G9 (*Figure 3A–C*), using a previously established protocol (*Losada-Perez et al., 2016*). qRT-PCR in injured VNCs revealed approximately a two-fold increase in *ia-2* mRNA levels at 5–7 hr post-injury, which recovered homeostatically by 24 hr post-injury (*Figure 3B*). This paralleled the effect of injury on *kon* expression (*Losada-Perez et al., 2016*). Thus, CNS injury caused an increase in *ia-2* expression.

Since increased *ia-2* levels induced ectopic Dpn+ cells (*Figure 2I–K*), and *ia-2* was upregulated in injury, we asked whether injury induced neural stem cells. We focused on the abdominal VNC only, which has three neuroblasts per hemi-segment in ventro-lateral positions, in early third instar larvae. Crush injury in the abdominal VNC at 74–76 hr AEL resulted in ectopic abdominal Dpn+ cells by 5–7 hr later (*Figure 3A,D*). These were more numerous than the normal developmental abdominal larval neuroblasts, and included cells located in dorsal positions, which are not normally occupied by them (see *Sousa-Nunes et al., 2011*; *Froldi et al., 2015*). The numerous Dpn+ cells could correspond to injury-induced divisions of neuroblasts normally found during larval development. To test whether injury might induce ectopic neural stem cells distinct from developmental neuroblasts, we next carried out crush injury at three later time points: (1) at 96 hr AEL and analysed the VNCs 6 hr post-injury (PI, 102 hr AEL), when in control VNCs, abdominal hemi-segments have 0 or 1 Dpn+ cells remaining (*Figure 3E–I*); (2) at 105 hr and analysed 24 hr PI (129 hr AEL), when in controls there are no ventro-lateral neuroblasts, only Dpn+ cells along the midline (*Figure 3J*); and (3) at 117 hr AEL and analysed the VNCs 12 hr PI (129 hr AEL), taking advantage of the delayed pupariation of injured larvae (*Figure 3K–O*). At 129 hr AEL there were no remaining abdominal ventro-lateral neural stem cells in intact controls, only some Dpn+ cells along the midline (*Figure 3M,N*). Normal thoracic Dpn + neuroblasts were present in all samples (*Figure 3F,L*). Injury induced at the three later time points caused ectopic abdominal Dpn+ cells at the lesion site (*Figure 3G,H,M,N,P*). Most, ectopic Dpn+ cells lacked Ia-2YFP (*Figure 3G,H,M,N*). Importantly, most ectopic abdominal Dpn+ cells surrounded the neuropile, and some were dorsal, in positions never occupied by developmental neural stem cells (*Figure 3H,N*). Ectopic abdominal Dpn+ cells were located surrounding the lesions (*Figure 3G,M,P*). To take into account that Dpn might also be induced in cells at a distance from the lesion site, we also counted the total number of abdominal Dpn+ cells in all injured samples (*Figure 3G–O*). The number of abdominal Dpn+ cells increased significantly in injured samples compared to controls (*Figure 3I,J,O*). The incidence of ectopic Dpn+ cells at the injury site decreased as larval age at the time of injury increased (*Figure 3P*). This most likely means that in older injured larvae there was not enough time between injury and pupariation for cells to divide further. Altogether, these data showed that injury induced ectopic neural stem cells that were distinct from developmental neuroblasts. Since *ia-2* levels increased upon injury, and *ia-2* GOF induced neural stem cells, this suggested that *ia-2* was responsible for the increase in Dpn+ cells caused by injury.

## Neuronal Ia-2 and glial Kon regulate Dilp-6

The above data raised the question of how Ia-2 might induce ectopic neural stem cells. Ia-2 is highly evolutionarily conserved and it functions in dense core vesicles to release insulin and neurotransmitters (*Harashima et al., 2005*; *Kim et al., 2008*; *Nishimura et al., 2010*; *Cai et al., 2011*). There are eight *Drosophila* insulin-like-peptides (Dilps) and Ia-2 affects only Dilp-6 (*Kim et al., 2008*). *dilp-6* is expressed in cortex and blood–brain barrier CNS glia, and activates neural stem cell proliferation following a period of quiescence in normal larval development (*Chell and Brand, 2010*; *Sousa-Nunes et al., 2011*). Thus, we asked whether the increase in Dpn+ cells in *ia-2* LOF and GOF observed above involved *dilp-6*.

We first visualised *dilp-6* expressing cells in wandering larvae using *dilp6-GAL4* (*Chell and Brand, 2010*; *Sousa-Nunes et al., 2011*) to drive expression of the nuclear reporter Histone-YFP (His-YFP). Most *dilp-6>his-YFP+* cells were also Repo+, but they did not surround the neuropile and lacked the neuropile glial marker Pros (*Figure 4A,B*). Therefore, most *dilp-6* expressing cells in the abdominal

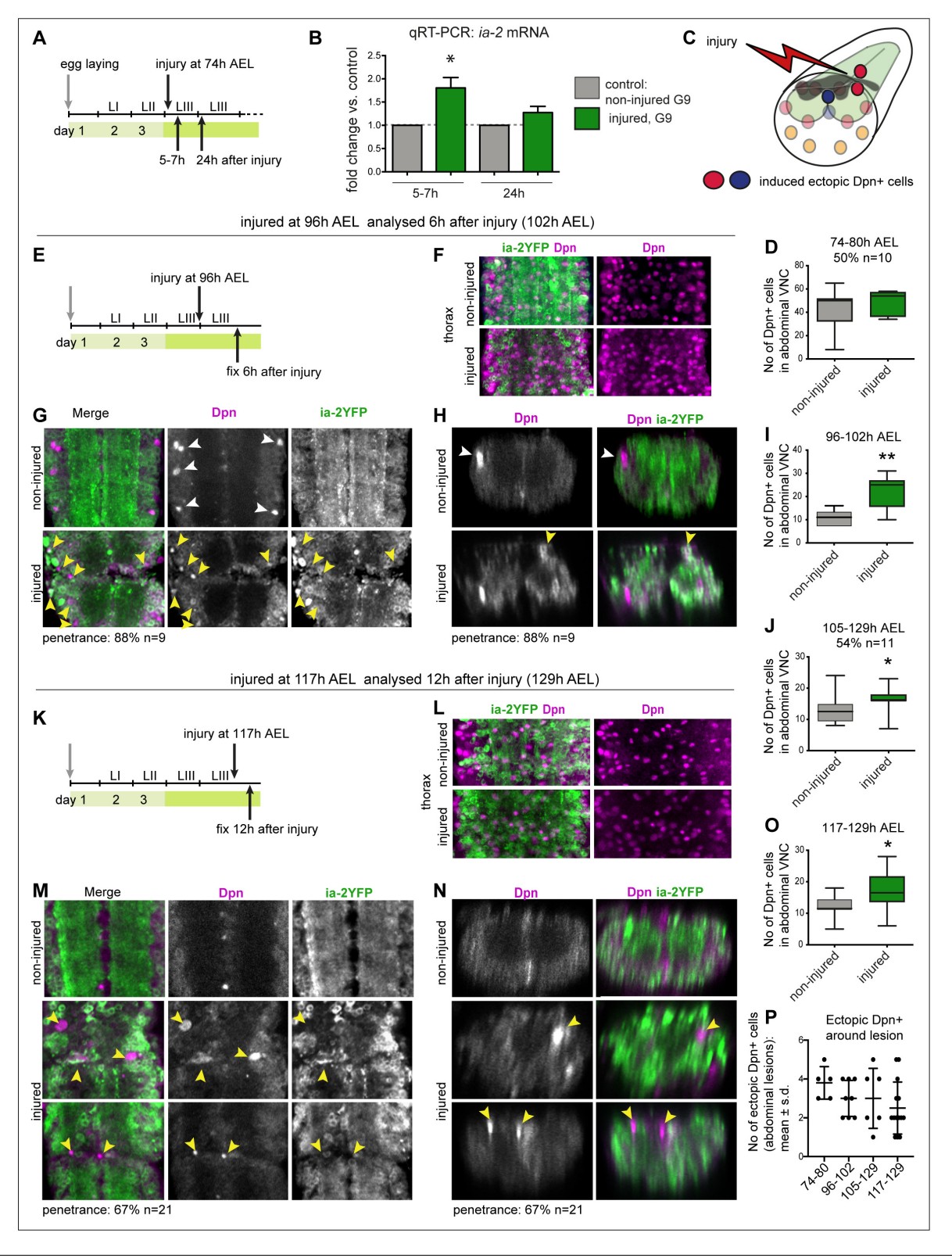

**Figure 3.** Injury induced *ia-2* expression and ectopic Dpn+ cells. (A, E, and K) Time course of crush-injury experiments in the larval abdominal ventral nerve cord (VNC), indicating the age of the larvae (after egg laying, AEL) when crush was applied (top arrows), followed by various recovery periods, and when they were dissected and fixed (bottom arrows). (C) Diagram showing that crush injury induced ectopic Dpn+ cells. (A, B, and D) Crush injury in the larval abdominal VNC at 74 hr AEL: (B) increased the levels of *ia-2* mRNA at 5–7 hr post-injury, which recovered homeostatically by 24 hr,

*Figure 3 continued on next page*

*Figure 3 continued*

detected by qRT-PCR. N = 3 biological replicates. (D) Induced ectopic abdominal Dpn+ cells by 5–7 hr post-injury (74–80 hr AEL) in half of injured VNCs (penetrance 50%, N = 10), and increased (albeit not significantly) the total number of abdominal Dpn+ cells inthose samples. Mann–Whitney U-test, p=0.24. (F and L) Thoracic Ia-2YFP and Dpn signal were normal in thoracic neuroblasts, in samples shown in (G and H) and (M and N), respectively. (E–I) Injury at 96 hr AEL caused ectopic abdominal Dpn+ cells by 6 hr post-injury (arrowheads). Most Dpn+ cells were Ia-2YFP⁻. At this stage, some developmental neuroblasts could still remain (white arrowheads), but ectopic abdominal Dpn+ cells were dorsal (yellow arrowheads, H). Most injured VNCs had ectopic abdominal Dpn+ cells (88% penetrance N=9). (I) Unpaired Student's t-test p=0.0063. (J) Injury at 105 hr AEL and fixation at 129 hr AEL, when no developmental neuroblasts remain, induced a significant increase in abdominal Dpn+ cells (54.5% penetrance N = 11). Mann–Whitney U-test p=0.0375. (K–O) Injury at 117 hr AEL caused ectopic abdominal Dpn+ cells by 12 hr post-injury (129 hr AEL). . Ectopic abdominal Dpn+ cells were found in ectopic dorsal positions (yellow arrowheads, N). This stage is devoid of developmental neural stem cells. Over two thirds of injured VNCs had ectopic Dpn+ cells (67% penetrance N=21). (O) Student's t-test p=0.0302. (I,J,O) All abdominal Dpn+ cells were counted in all injured samples. (P) Temporal profile of number of ectopic Dpn+ cells surrounding the lesions, number in X-axis indicate time points of injury and fixation. (F, G, L, and M) Horizontal views, (H and N) transverse views. (D, I, J, O ) Graphs show box-plots . (P) Shows dot plots, with mean and error bars (±s.d.) indicated. *p<0.05, **p<0.01. For full genotypes and further statistical analysis details, see *Supplementary file 1*.

larval VNC were cortex and surface glia, as previously reported (*Chell and Brand, 2010*; *Sousa-Nunes et al., 2011*). Some *dilp6>his-YFP+* cells were Repo⁻ Elav+ and thus were neurons (*Figure 4A,B*). Therefore, *dilp-6* is expressed in some neurons per VNC segment and mostly in non-neuropile glia.

Kon function is required for glial cell fate (*Losada-Perez et al., 2016*), so we used qRT-PCR to ask whether altering Kon levels might affect *dilp-6* expression. Overexpression of full-length *kon* mildly increased *dilp-6* mRNA levels (albeit not significantly), but *kon* knock-down in glia significantly reduced *dilp-6* mRNA levels (*Figure 4C,D*). This effect could be indirect, as glial proliferation and differentiation are impaired with *kon* loss of function (*Losada-Perez et al., 2016*), or perhaps Kon regulates *dilp-6* expression. Either way, *dilp-6* expression depends on *kon* in glia.

Dilp-6 is a ligand for the *insulin receptor* (*InR*), which functions at least in CNS neurons and neuroblasts (*Fernandez et al., 1995*; *Song et al., 2003*; *Sousa-Nunes et al., 2011*; *Fernandes et al., 2017*). To revisit what cells might receive Dilp-6, we visualised *InR* expression, using multiple available GAL4 lines to drive *his-YFP,* and tested co-localisation with glial and neuronal markers. At 72 hr AEL, *InR^{NP2552}>his-YFP+* cells comprised some Elav+ neurons and some Repo+ glia, including dorsal neuropile glia and surface glia (*Figure 4E,F*). The distribution was stochastic, most likely due to the insertion of GAL4 into an intron. To verify whether *InR* is expressed in glia, we searched published single cell RNAseq data from the larval CNS (*Brunet Avalos et al., 2019*). Indeed, of 152 Repo+ glial cells, 97 were also InR+, that is, 63% of glial cells express *InR*. Thus, *InR* is expressed in both neurons and glia.

Altogether, these data indicated that in the third instar larva Dilp-6 is produced and secreted by Ia-2 from some neurons, it is mostly produced in non-neuropile glia, and it is received by InR in neurons and glia, including neuropile glia.

## A positive neuron-glia communication loop boosts Dilp-6 production from glia

The above data strongly suggested that a neuron-glia communication loop might serve to amplify Dilp-6. A limiting step could be Kon, as glial *dilp-6* expression depends on *kon*. Kon is required for glial gene expression (*Losada-Perez et al., 2016*), but whether this depends on the nuclear translocation of its intracellular domain, Kon^{ICD}, is unknown. In *Drosophila,* Kon had been reported to lack a nuclear localisation signal (*Schnorrer et al., 2007*). In mammals, NG2^{ICD} positively regulates the expression of multiple genes, including downstream targets of mTOR (*Sakry et al., 2015*; *Nayak et al., 2018*), but whether this requires nuclear NG2^{ICD} is also unknown. Altogether, whether NG2 or Kon regulates glial gene expression through nuclear events remained unsolved. Thus, to ask whether Kon^{ICD} might function in the nucleus, we generated a HA-tagged form of Kon^{ICD} (Kon^{ICD-HA}). Glial overexpression of *kon^{ICD-HA}* (*repoGAL4>UAS-Kon^{ICD-HA}*) revealed distribution of anti-HA in glial cytoplasms and in nuclei, co-localising with the glial nuclear transcription factor Repo, in both embryos and larvae (*Figure 5A* and *Figure 5—figure supplement 1*). Thus, Kon^{ICD} is distributed in the cytoplasm and nucleus, from where it could regulate gene expression.

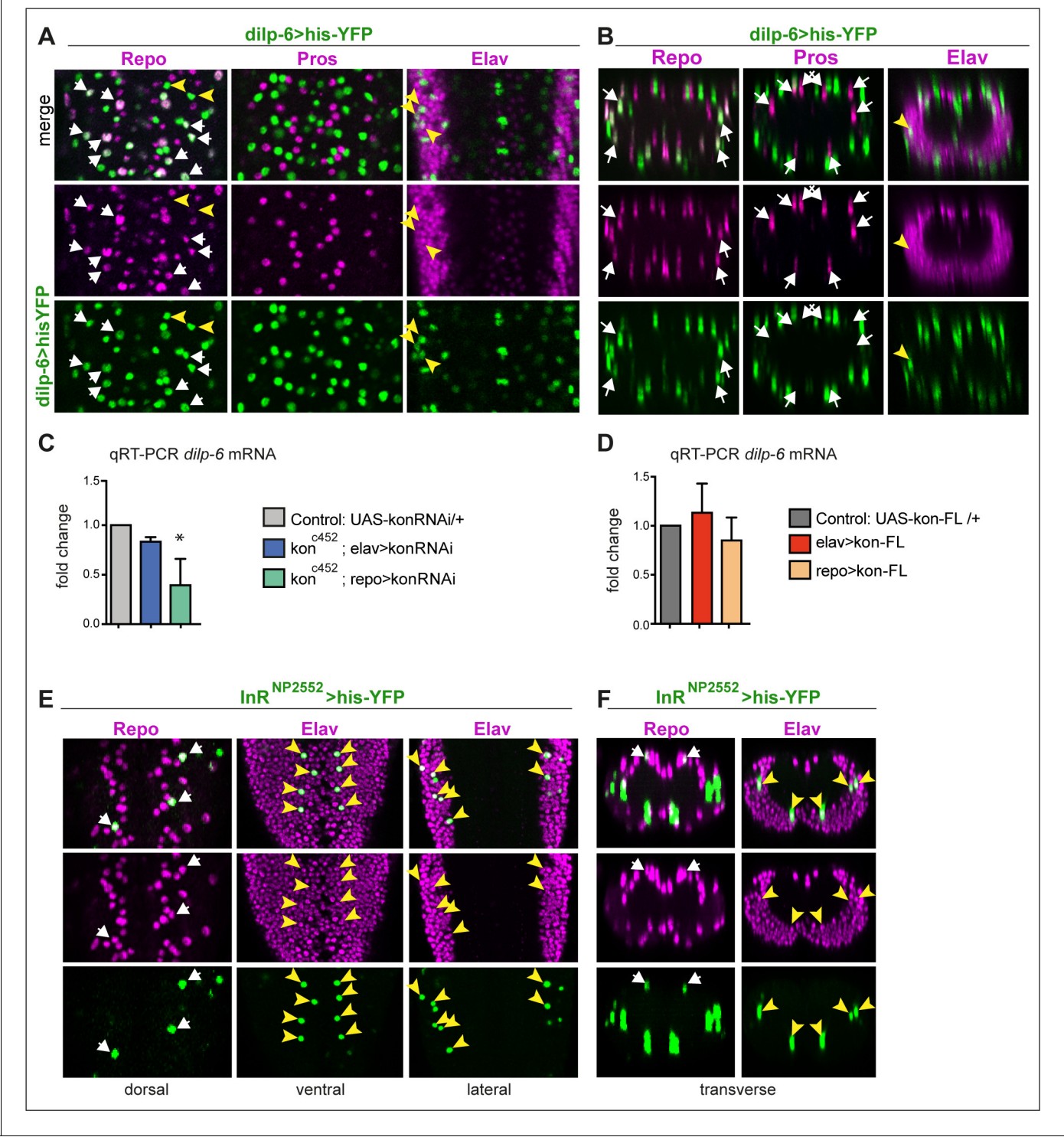

**Figure 4.** *dilp-6* is expressed in neurons and cortex glia and received by neuropile glia. (**A and B**) *Dilp-6GAL4>UAShisYFP* cells are mostly Repo+ Pros⁻ glia that do not surround the neuropile (white arrows), and from position appear to be cortex and surface glia. No YFP+ cells have Pros. Some cells are Repo⁻ Pros⁻ Elav+ (yellow arrowheads) meaning they are neurons. (**C and D**) qRT-PCRs showing that: (**C**) *kon* knock-down in glia *(kon^{c452}/ UASkonRNAi; repoGAL4/+)* downregulates *dilp-6* mRNA levels; (**D**) overexpression on *kon* does not cause a significant effect. N = 3 replicates for both. (**C and D**) One-way ANOVA, only differences in (**C**) for *dilp-6* mRNA significant p=0.0362, *p<0.05. (**E and F**) *inR* expression visualised with reporter *InR^{NP2552}GAL4> UAShistoneYFP* is expressed stochastically in some dorsal Repo+ neurople glia (white arrows), and other glia, and in some Elav+

*Figure 4 continued*
neurons (yellow arrowheads). (**A and E**) Horizontal views of the abdominal ventral nerve cord (VNC); (**B and F**) transverse views. For full genotypes and further statistical analysis details, see *Supplementary file 1*.

Next, we asked whether Kon$^{ICD}$ is functional. Since NG2 and Kon are responsible for glial proliferation both in mammals and *Drosophila* (*Kucharova and Stallcup, 2010*; *Losada-Perez et al., 2016*), we used glial cell number as a read-out of Kon$^{ICD}$ function. First, we tested whether cleaved Kon$^{ICD}$ could induce glial proliferation, like full-length Kon does (*Losada-Perez et al., 2016*). We overexpressed *kon$^{ICD}$* in glia and automatically counted glial cells labelled with the nuclear marker his-YFP, using DeadEasy software (*Forero et al., 2012*). Overexpression of *kon$^{ICD}$* in glia increased glial cell number (*UAShisYFP; repoGAL4>UASkon$^{ICD}$*, *Figure 5B,C*), meaning that Kon$^{ICD}$ can induce glial proliferation. As full-length Kon also promotes glial proliferation (*Losada-Perez et al., 2016*), these data meant that full-length Kon is normally cleaved, releasing Kon$^{ICD}$ to promote glial proliferation.

In principle, Dilp-6 amplification could occur if it was first secreted from neurons by Ia-2 to activate InR in glia, and InR signalling in turn drove the Kon-dependent upregulation of *dilp-6* expression in glia (*Figure 5F*). To test whether Dilp-6 activates InR in glia, which activates Kon, we asked: (1) whether overexpression of *dilp-6* could mimic the increase in glial cell number caused by Kon$^{ICD}$, and (2) whether this could be rescued by overexpression of a dominant negative form of the *insulin receptor (InR$^{DN}$)* in glia. We found that overexpression of *dilp-6* in glial cells increased glial cell number comparably to Kon$^{ICD}$ (*Figure 5B,C*). Furthermore, this was rescued with concomitant overexpression of *InR$^{DN}$* in glia (*Figure 5B,C*). These data meant that Dilp-6 activates InR signalling in glia and induces glial proliferation.

Dilp-6 and InR signalling reactivate quiescent developmental neural stem cells (*Chell and Brand, 2010*; *Sousa-Nunes et al., 2011*), but Kon functions in glia (*Losada-Perez et al., 2016*). To further verify whether Kon function is restricted to glia, we asked whether Kon might also be required in neural stem cells during development at 72 hr AEL, when normally there are neural stem cells in both thorax and abdomen of larvae (*Figure 5D*). RNAi *kon* knock-down in neural stem cells with *inscutable-GAL4 (ins-GAL4>UAS-konRNAi)* did not affect the number or distribution of abdominal developmental Dpn+ cells at 72 hr AEL (*Figure 5D,E*), meaning that Kon is not required for neural stem cell development. Since glial proliferation depends on Kon (*Losada-Perez et al., 2016*), the fact that *dilp-6* alone could reproduce the increase in cell number caused by *kon$^{ICD}$*, and this depended on InR in glia, strongly suggested that InR signalling can activate Kon cleavage downstream in glia.

To conclude, altogether these data suggested that Ia-2 triggers the release of Dilp-6 from neurons, which then is received by glial cells, where InR signalling activates Kon, which in turn induces glial proliferation enabling further production of Dilp-6. Thus, a non-autonomous relay from neuronal Ia-2 to glial Kon promotes glial proliferation and induces a positive feedback loop that amplifies Dilp-6 production from glia (*Figure 5F*).

## Ia-2 and Dilp-6 can induce neural stem cells from glia

So far, our data had shown that: alterations in Ia-2 levels caused either by genetic manipulation or injury induced ectopic neural stem cells; Ia-2 is required for the neuronal secretion of Dilp-6, which is received and amplified in cortex glia under the control of Kon; and secreted Dilp-6 is received by InR also in neuropile glia. As Dilp-6 activates quiescent developmental neural stem cells (*Chell and Brand, 2010*; *Sousa-Nunes et al., 2011*), this raised the question of whether the Ia-2-Kon-Dilp-6 loop not only produced more glia but could also induce a neurogenic response from glia.

To ask whether Kon, Ia-2, or Dilp-6 could be responsible for inducing ectopic neural stem cells from glia, we overexpressed them in glia (with *repoGAL4*), and analysed Dpn at 120 hr AEL, after the disappearance of developmental abdominal neural stem cells. Dpn was detected normally in thoracic neuroblasts in all samples (*Figure 6A* and *Figure 6—figure supplement 1*). Interestingly, overexpression of a dominant negative form of the *InR (InR$^{DN}$)* in glia together with *dilp-6* reduced the levels of Dpn in thoracic neuroblasts (*Figure 6A*).

Overexpression of *kon-FL* did not induce ectopic abdominal Dpn+ cells (*Figure 6B–E* and *Figure 6—figure supplement 1*). To test whether Kon$^{ICD}$ might be required instead, we overexpressed

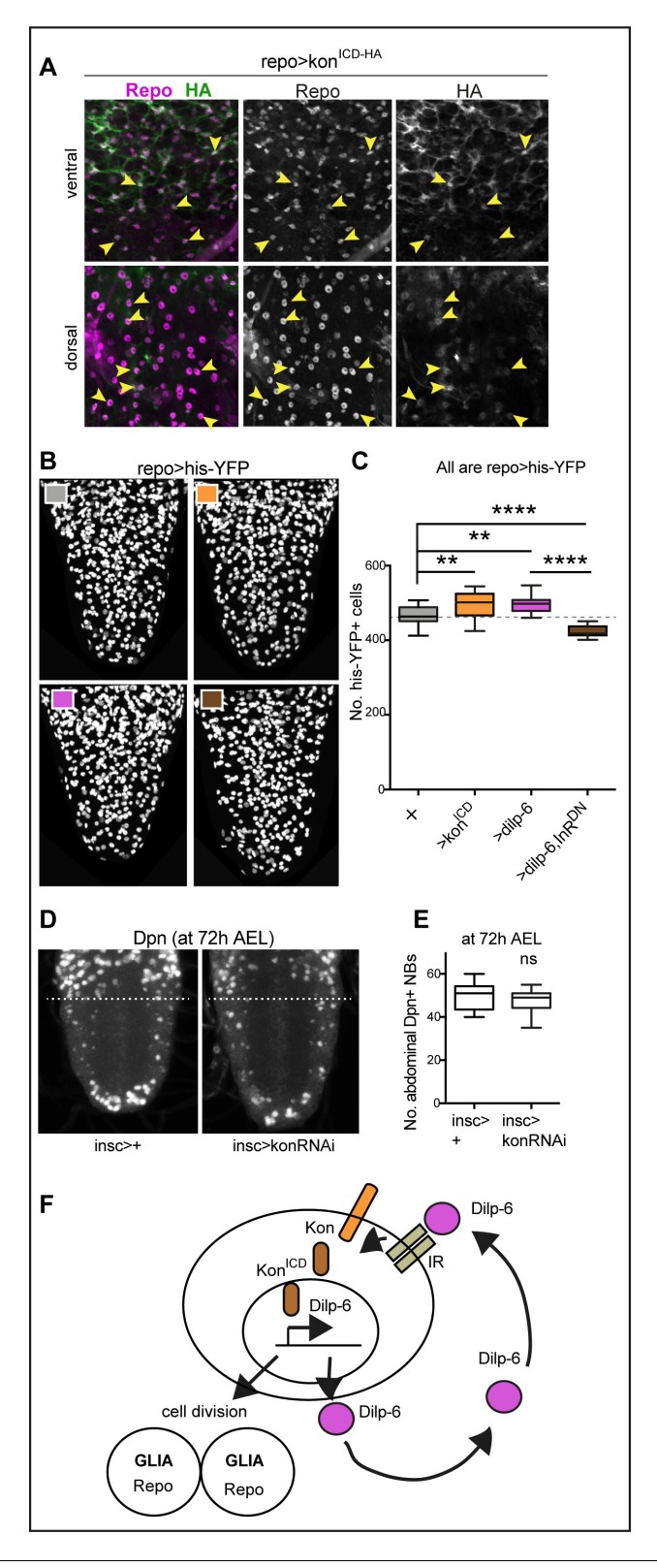

**Figure 5.** Ia-2, Kon, and Dilp-6 are linked though a neuron-glia communication loop. (**A**) Overexpressed HA-tagged Kon[ICD] in glia (*repoGAL4>UASkon[ICD]::HA*) visualised with anti-HA antibodies in third instar wandering larvae, localises to both glial cytoplasms and nuclei (arrows). (**B and C**) Overexpression of the intracellular domain of *kon* (*kon[ICD]*) or *dilp-6* increased glial cell number, visualised with *repoGAL4>UAShistone*-YFP, and quantified

*Figure 5 continued on next page*

*Figure 5 continued*

automatically with DeadEasy in (**C**). Overexpression of a dominant negative form of the insulin receptor rescues the increase in cell number caused by Dilp-6 (*repo>hisYFP, dilp-6, InRDN*), meaning that autocrine InR signalling regulates glial proliferation. Box-plots. One-way ANOVA p<0.0001, post hoc Tukey's test multiple comparisons between all samples. N = 15–28 ventral nerve cords (VNCs). (**D and E**) Third star larvae at 72 hr AEL to visualise abdominal developmental neuroblasts: *kon-RNAi* knock-down in neural stem cells with *insGAL4* does not affect Dpn+ cell number. Box-plots. Unpaired Student's t-test, p=0.3111. N = 10 VNCs. (**F**) Illustration summarising that a positive feedback autocrine loop involving Dilp-6, InR, and Kon promotes both glial proliferation and Dilp-6 production. All images are horizontal views. Asterisks refer to multiple comparison post hoc tests, all samples vs. all: **p<0.01, ****p<0.0001. All graphs show box-plots. For full genotypes and further statistical analysis details, see ***Supplementary file 1***.

The online version of this article includes the following figure supplement(s) for figure 5:

**Figure supplement 1.** Over-expressed HA-tagged kon^ICD^ localised to nuclei in embryos.

---

kon^ICD^ in glia, but it did not induce ectopic Dpn+ cells either (***Figure 6—figure supplement 2A***). We verified whether *repo>Kon^ICD^* resulted in increased *kon^ICD^* expression, and it did, between 15- and 20-fold (***Figure 6—figure supplement 2B***). However, *kon^ICD^* overexpression in glia (*repo>kon^ICD^*) did not significantly alter mRNA levels for *dilp-6* nor *dpn* either (***Figure 6—figure supplement 2C***). These data meant that Kon^ICD^ does not directly function as a transcription factor. Instead, it may function as a co-factor of an unknown transcription factor. Kon^ICD^ could also participate in the nuclear import–export shuttle, as loss of *kon* function prevented nuclear translocation of Repo (***Losada-Perez et al., 2016***). Thus, most likely Kon regulates *dilp-6* indirectly by influencing glial cell fate and it does not induce ectopic Dpn+ cells.

By contrast, overexpression of *ia-2* induced ectopic abdominal Dpn+ cells prominently along the midline but also in lateral locations surrounding the neuropile, ordinarily occupied by glia (***Figure 6B–E*** and ***Figure 6—figure supplement 1***). Overexpression of *dilp-6* had a stronger effect, and there were many ectopic Dpn+ cells surrounding the neuropile (***Figure 6B–E*** and ***Figure 6—figure supplement 1***). Dpn levels in ectopic cells were generally lower than in normal thoracic neural stem cells. These data showed that both Ia-2 and Dilp-6 can induce *dpn* expression, potentially in glia. However, Kon alone cannot, meaning that insulin signalling is required to induce neural stem cells. Since Ia-2 drives Dilp-6 production and secretion, this suggested that ultimately Dilp-6 induced ectopic neural stem cells.

To further test whether Ia-2 upregulated *dpn* ectopically via Dilp-6, we carried out epistasis analysis. Overexpression of *ia-2* together with *dilp-6* knock-down in glia (*ia-2YFP, repoGAL4>UAS-ia-2, UAS-dilp-6RNAi*), rescued the number of abdominal Dpn+ cells (***Figure 6A–E*** and ***Figure 6—figure supplement 1***), demonstrating that Ia-2 induces ectopic neural stem cells via Dilp-6. Furthermore, overexpression of *ia-2* together with *kon* RNAi in glia (*ia2-YFP, repoGAL4>UAS-ia-2, UAS-konRNAi*) also rescued the Dpn+ phenotype (***Figure 6B–E*** and ***Figure 6—figure supplement 1***), confirming that *dilp-6* expression depends on Kon in glia (see ***Figure 4C***) and that Kon and Dilp-6 engage in a positive feedback loop (see ***Figure 5***). Finally, the ectopic Dpn+ phenotype was also rescued by overexpression of *dilp-6* together with *InR^DN^* in glia (***Figure 6B–E*** and ***Figure 6—figure supplement 1***, *ia-2YFP repoGAL4>UAS-dilp6, UAS-InR^DN^*), meaning that ectopic neural stem cells depend on InR signalling in glia. Together, these data showed that Ia-2 induces ectopic abdominal neural stem cells via Dilp-6 and InR signalling in glia, and that ectopic Dpn cells originated from glia (***Figure 6F***).

The observation that Ia-2 and Dilp-6 could induce neural stem cell marker expression from glia was important. Thus, we sought to further verify it in two ways. Firstly, we used a second anti-Dpn antibody aliquot, from the Wang Lab (***Huang and Wang, 2018***; ***Zhang et al., 2019***), as well as their protocol. This revealed thoracic Dpn signal in normal neuroblasts, in late third instar larvae (***Figure 7A–F***). In the abdomen, Dpn could be detected in some cells at low levels in ectopic positions corresponding to neuropile glia (***Figure 7A,G***). Ectopic abdominal Dpn levels increased significantly when *dilp-6* was overexpressed in glia (with *repoGAL4*, ***Figure 7A,G,J***). *dilp-6* overexpression also increased abdominal Dpn+ cell number (***Figure 7G,M***), reflecting either higher levels and/or that Dpn+ cell proliferated. Secondly, we asked whether directly manipulating downstream effectors of insulin signalling – Ras and PI3Kinase – in glia might also influence *dpn* expression. Since InR signalling can trigger multiple signalling pathways downstream, we overexpressed *dilp-6* together with

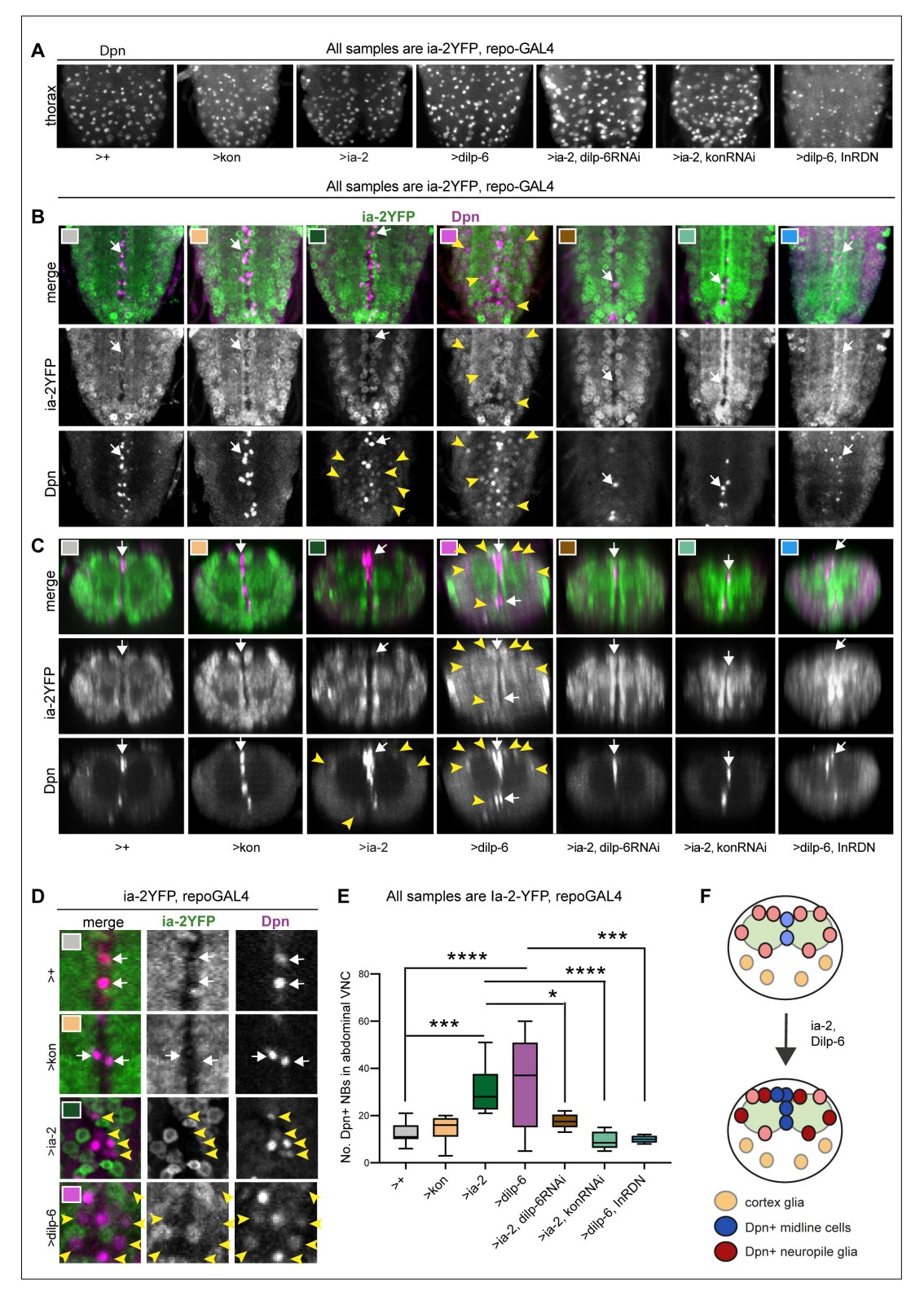

**Figure 6.** Ia-2 and Dilp-6 induce ectopic neural stem cells from InR signalling in glia. All samples were analysed at 120 hr AEL, after disappearance of abdominal developmental neuroblasts. (**A**) Dpn signal in thorax was normally strong and clear, except with the overexpression of both *dilp-6* and *InR^DN* in glia, which reduced Dpn levels and NB size. (**B and C**) Overexpression of *ia-2* and *dilp-6*, but not *kon-full-length*, induced Dpn+ cells in the abdominal ventral nerve cord (VNC), at the midline and in lateral positions. Ectopic abdominal Dpn was at lower levels than normal thoracic signal in

*Figure 6 continued on next page*

*Figure 6 continued*

NBs. (D) Ectopic Dpn+ cells did not express Ia-2YFP (arrowheads). (E) Overexpression of *ia-2* or *dilp-6* increased abdominal Dpn+ cell number. Quantification of all abdominal VNC Dpn+ cells, and genetic epistasis analysis showing that: the increase in Dpn+ cell number caused by *ia-2* overexpression was rescued by *dilp-6 RNAi* and *kon-RNAi* knock-down in glia, meaning that *ia-2* requires Dilp-6 and glial Kon to induce Dpn; and preventing insulin signalling with InR$^{DN}$ in glia rescued the increase in Dpn+ cell number caused by *dilp-6* overexpression, meaning that Dilp-6 induced Dpn via InR signalling in glia. One-way ANOVA p<0.0001, post hoc Tukey's test multiple comparisons all samples vs. all. N = 5–13 VNCs. (F) Illustration showing that Ia-2 and Dilp-6 can induce Dpn via InR signalling in glial cells. (A and B) Horizontal views; (C) transverse views; (D) higher magnification. Graphs show quantifications in box-plots. Asterisks refer to multiple comparison post hoc tests: *p<0.05, ***p<0.0001, ****p<0.0001. For full genotypes and further statistical analysis details, see *Supplementary file 1*.

The online version of this article includes the following figure supplement(s) for figure 6:

**Figure supplement 1.** Dpn in thoracic neuroblasts of specimens shown in *Figure 6*.

**Figure supplement 2.** Kon$^{ICD}$ does not induce *dilp-6* nor *dpn* expression.

the activated forms of Ras (Ras$^{V12}$) and PI3Kinase (Dp110CAAX, hereby called activated PI3K). In order to prevent potential embryonic lethality, to separate Dilp-6 function from its developmental function activating quiescent neuroblasts, and to separate further developmental neuroblasts from ectopic Dpn in glia, we used *tubGAL80$^{ts}$* to conditionally overexpress these factors in glia after larval L1 hatching. Samples were analysed in late third instar larvae, just before pupariation, when no normal neuroblasts remain in the abdomen. Dpn was normal in thoracic neuroblasts, with genotype-specific modulation (*Figure 7A–F*). In the abdominal VNC, overexpression of activated *ras$^{V12}$* in glia (*tubGAL80$^{ts}$, repo>act-ras$^{V12}$*) increased the levels of ectopic Dpn (*Figure 7B,H,K*), and joint overexpression of *dilp-6* and *activated-ras$^{V12}$* (*tubGAL80$^{ts}$, repo>dilp6, act-ras$^{V12}$*) increased Dpn levels further (over twofold, *Figure 7B,H,K*). Remarkably, whereas activated Ras$^{V12}$ did not alter abdominal Dpn+ cell number, together Dilp-6 Ras$^{V12}$ resulted in a dramatic threefold increase (*Figure 7B,H,N*). It appeared that more than all glial cells now expressed *dpn*. These ectopic Dpn+ cells were located throughout the VNC, also invading and disrupting the neuropile (*Figure 7H*). Thus, activated Ras with Dilp-6 induced Dpn+ cell proliferation. Overexpression of activated PI3K in glia (*tubGAL80$^{ts}$, repo>act-PI3K*) had no effect on abdominal Dpn+ levels (*Figure 7I,L*), but levels increased nearly fourfold when *dilp-6* was also overexpressed (*tubGAL80$^{ts}$, repo>dilp-6, act-PI3K*) (*Figure 7I,L*). Interestingly, and contrary to Ras$^{V12}$, activated PI3K induced Dpn only surrounding the neuropile, in positions normally occupied by neuropile glia (astrocytes, ensheathing glia, and midline glia). Co-overexpression of *dilp-6 act-PI3K* increased the number of Dpn+ cells (*Figure 7I,O*), but not as dramatically as Dilp-6 Ras$^{V12}$. These data demonstrated that insulin signalling induces *dpn* expression ectopically in late larval glial cells and promotes their proliferation.

To further test whether the ectopic Dpn+ cells originated from glia, we first asked whether ectopic abdominal Dpn colocalised with the glial marker Repo, in larvae at 120 hr AEL, after the disappearance of developmental abdominal neural stem cells. Lateral ectopic Dpn+ cells observed with *dilp-6* overexpression were Repo+ (*Figure 8A,B*), consistent with originating from glial cells. Dpn levels were lower than in normal neural stem cells. By contrast, ectopic midline Dpn+ cells were not Repo+. Midline glia do not normally express *repo*, but express *wrapper (wrp)*. Overexpression of *dilp-6* resulted in Dpn+ cells along the midline that also had Wrp (*Figure 8C,D*), showing that ectopic midline Dpn+ cells were midline glia. Thus, there are two distinct populations of ectopic Dpn+ cells: latero/dorsal Repo+ around the neuropile and midline Wrp+ cells, meaning that Dpn was induced in neuropile glia (class known as 'astrocytes') and midline glia.

However, not all Dpn+ cells were Repo+ or Wrp+, as some did not express either of these markers (*Figure 8E,F*, white arrows; genotype: *repoGAL4>his-YFP, dilp-6*). This could mean that either some ectopic Dpn+ did not originate from glia, or that as glial cells reprogrammed into neural stem cells, they switched off glial gene expression. To test whether ectopic neural stem cells originated from glia, we used the cell-lineage marker G-TRACE. This GAL4-dependent tool results in the permanent labelling of GAL4/UAS-expressing cells and their lineage. Thus, as glial cells become neural stem cells, the glial *repo* promoter would be switched off, but G-TRACE would enable their visualisation as well as that of all their progeny cells. Cells that were originally glia but may no longer be so would be labelled in green (GFP+), and recently specified glial cells would be labelled in red (RFP+). G-TRACE expression in glia with *repoGAL4* together with *dilp-6* caused larval lethality and thus could not be analysed. By contrast, overexpression of both G-TRACE and *ia-2* in glia (*repoGAL4>G-*

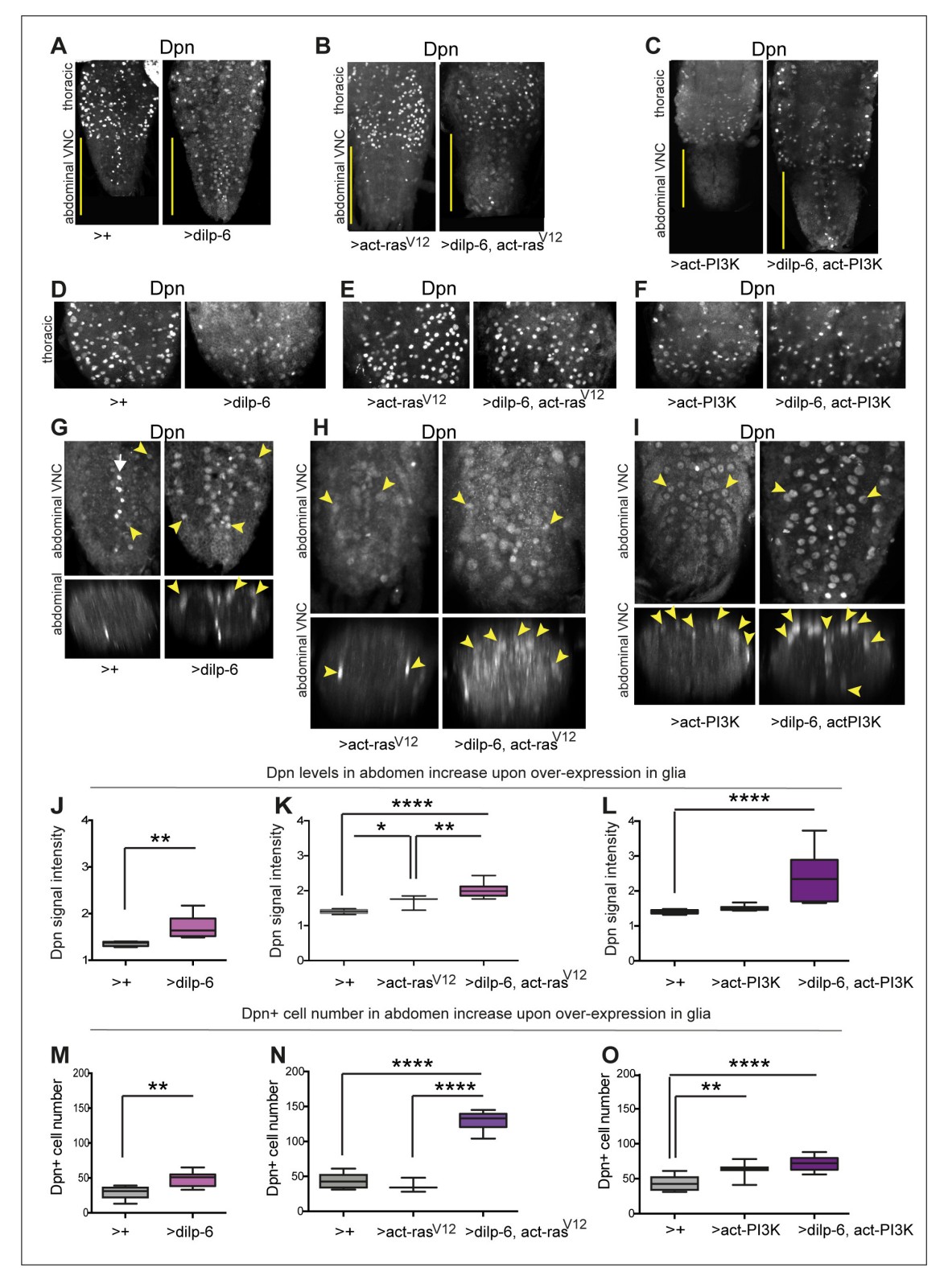

**Figure 7.** Activation of Ras or PI3K downstream of insulin signalling in glia induces ectopic Dpn+ cells. (**A–C**) Full ventral nerve cord (VNC) projections of VNCs showing normal thoracic levels of Dpn, and genotype-specific modulations, using anti-Dpn from Wang Lab. (**D–F**) Projections of thoracic sections, showing Dpn signal in normal NBs. (**G, J, and M**) Overexpression of *dilp-6* with *repoGAL4* induced ectopic abdominal Dpn (**G**). Some ectopic Dpn+ cells were also observed at low levels in control samples. (**J**) Dilp-6 increased ectopic abdominal Dpn levels, unpaired Student's t-test; and (**M**)

*Figure 7 continued on next page*

Figure 7 continued

abdominal Dpn+ cell number, Mann–Whitney U-test. (H, I, K, L, N, and O) *tubGAL80ts repoGAL4* was used to overexpress *dilp-6* together with activated *rasV12* or *PI3K* in glia to prevent embryonic lethality (48 hr 18°C, followed by 30°C until dissection ). (H, K, and N) Overexpression of *dilp-6* with activated *rasV12* induced ectopic Dpn in possibly all glia (H). Ectopic Dpn+ cells were large and neuropile integrity was disrupted. (K) Ectopic abdominal Dpn levels increased. One-way ANOVA, p<0.0001, Tukey's multiple comparison test. (N) Abdominal Dpn+ cell number increased threefold, indicating widespread Dpn+ cell proliferation. One-way ANOVA, p<0.0001, Tukey's multiple comparison test. (I, L, and O) Overexpression of *dilp-6* with activated *PI3K* induced ectopic abdominal Dpn in neuropile glia only (astrocytes and midline glia). Cells and VNC were large. (L) Ectopic abdominal Dpn levels increased. One-way ANOVA, p<0.0001, Tukey's multiple comparison test. (O) Abdominal Dpn+ cell number increased, One-Way ANOVA p<0.0001, Tukey's multiple comparison test. Graphs show quantifications in box-plots. Asterisks refer to multiple comparison tests: *p<0.05, **p<0.01, ****p<0.0001. N=3-10 VNCs. For full genotypes, sample sizes and further statistical analysis details, see *Supplementary file 1*.

TRACE, *ia-2*) revealed G-TRACE+ Dpn+ cells around the neuropile, at 120 hr AEL (*Figure 8G,H*). Most, if not all, of these cells had GFP, but also RFP (*Figure 8G,H*). These data demonstrate that ectopic Dpn+ originates from glial cells. Since RFP was also present, glial cell fate had not been suppressed, and instead glial cells may have been in the process of reprogramming.

Altogether, these data showed that Ia-2 and Dilp-6 can induce de novo formation of neural stem cells from neuropile and midline glial cells.

## In vivo reprogrammed glial cells can divide and generate neurons

To ask whether ectopic neural stem cells can divide to generate neurons, we used the S-phase marker PCNA-GFP, and overexpressed *dilp-6* specifically at the third instar larva using GAL4 under the control of a heat-shock promoter. We heat-shocked larvae at 110.5 hr AEL at 37°C for 30 min, and then kept them at 25°C for 9 hr, when they were dissected and fixed, to visualise Dpn+ and PCNA-GFP at 120 hr AEL. In control wandering third instar larvae, a few PCNA-GFP+ cells could be observed along the midline, but not in lateral positions (*Figure 9A,A'*). Overexpressed *dilp-6* resulted in ectopic Dpn+ PCNA-GFP+ cells in lateral positions around the neuropile (*Figure 9B,B', D*), as well as along the midline (*Figure 9C,C',D*). Some of the dividing midline cells were glia, as Dilp-6 overexpression resulted in PCNA-GFP+Wrp+ cells along the midline (*Figure 9E–H*), and in an increase in Wrp+ cells (*Figure 9G*). Thus, Dilp-6 overexpression induced proliferation and increased the number of both Repo+ glia (see *Figure 5*) and Wrp+ midline glia (*Figure 9E–H*). Upon overexpression of either *dilp-6* or *ia-2*, some of the dividing PCNAGFP+ Wrp+ cells also had Dpn (*PCNA-GFP, hsGAL4>ia-2*, or *dilp-6, Figure 9I–J'*), showing that Dpn+ cells of glial origin can divide. To further verify whether Wrp+ Dpn+ cells could divide, we used the mitotic marker anti-phospho-Histone-H3. Overexpression of *ia-2* in glia (*repo>Ia-2*) induced proliferation of Wrp+ Dpn+ cells, as labelled with pH3 (*Figure 9K,K'*). These data demonstrate that Ia-2 and Dilp-6 glial-reprogrammed neural stem cells can divide.

To ask whether the reprogrammed, proliferating Dpn+ cells might result in de novo neurogenesis, we first visualised cells using the *pros*-promoter, which drives expression in neural stem cells, GMCs, neurons, and glia. We reasoned that this promoter would be less likely to be silenced through a cell-state transition. FlyBow was used as a reporter to visualise *pros* expressing cells. Interestingly, this also revealed that the small Pros+ cells are generally neurons (*Figure 10A*). Overexpression of *dilp-6* with *pros-GAL4 (pros^voila^GAL4>UAS-FlyBow, UASdilp-6)* resulted in groups of GFP+ cells (at 120 hr AEL) that comprised one GFP+ cell, one GFP+Dpn+ Elav⁻ cell, and one GFP+-Dpn⁻ Elav+ cell (*Figure 10A,B*). These data were consistent with Dilp-6 reprogrammed glia becoming neurogenic.

To further verify that neurons could be generated by Dilp-6 from glia, we used a lineage-tracing method. We overexpressed *dilp-6* and *flippase (FLP)* in glia, to flip-out a stop codon placed between the *actin* promoter and GAL4, to swap the expression of the reporter GFP from being controlled by the glial *repo* promoter, to the constant *actin* promoter (*actin>y+STOP>GAL4 UAS-GFP/UAS-FLP; repoGAL4/dilp-6*). Thus, as reprogrammed glial cells switched off the glial *repo* promoter and switched on neural stem cell gene expression, they and their progeny cells would still be visible with GFP. Larvae were analysed at 120 hr AEL. In this genetic background, overexpression of *dilp-6* resulted in lateral ectopic Dpn+ cells that were also GFP+ (*Figure 10C,D*, at 120 hr AEL). This showed that, like with Ia-2 and G-TRACE (*Figure 8G,H*), ectopic Dpn+ cells induced by Dilp-6 originated from glia. Furthermore, there were groups of two to three GFP+ cells, some of which were

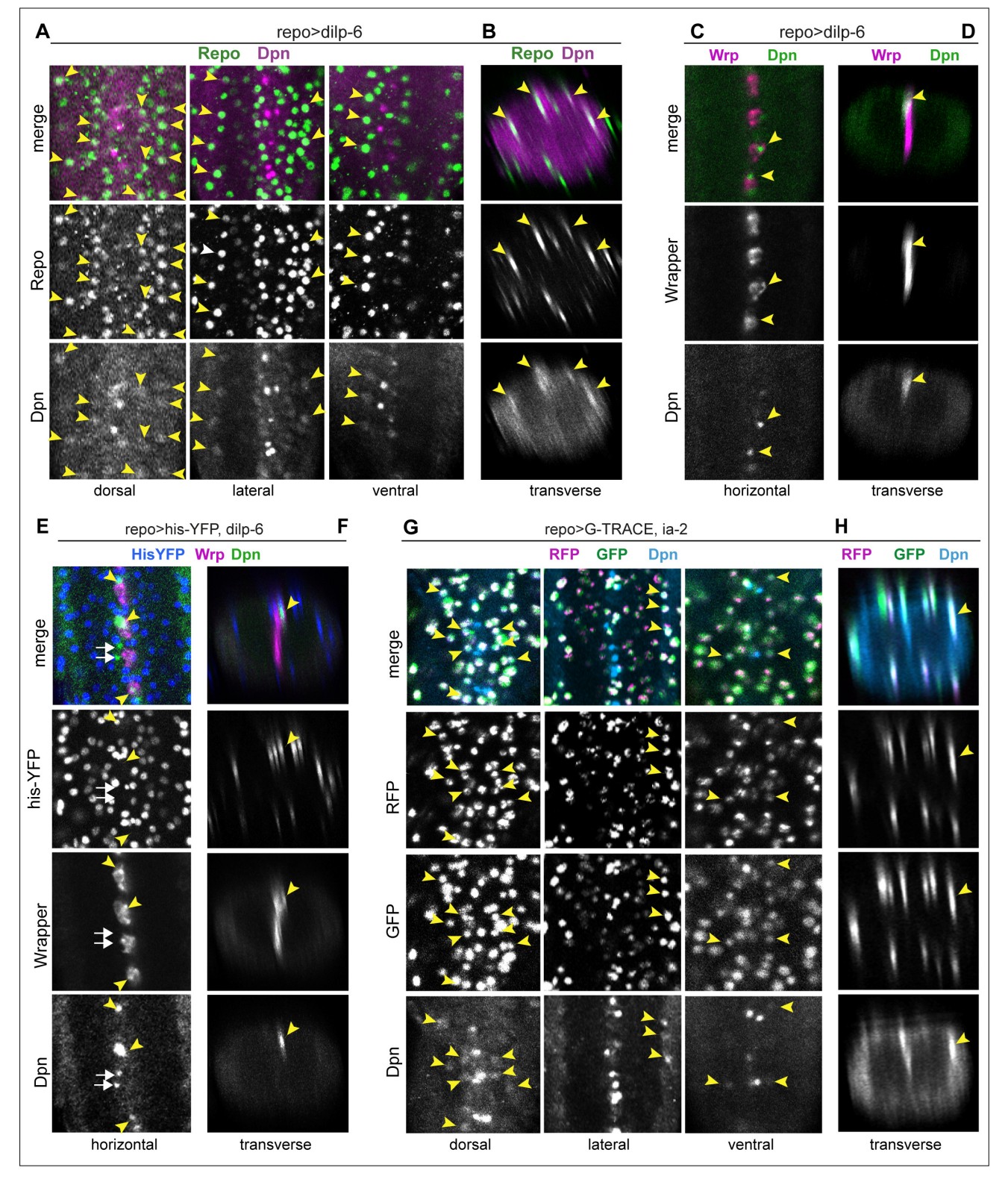

**Figure 8.** Ia-2 and Dilp-6 induced ectopic neural stem cells originate from glia. All samples were analysed at 120 hr AEL, after disappearance of abdominal developmental neuroblasts. (**A and B**) Overexpression of *dilp-6* from glia (*repoGAL4>UAS-dilp-6*) induced Dpn expression in Repo+ neuropile glial cells (arrowheads). N = 10 ventral nerve cords (VNCs). (**C and D**) Overexpressed *dilp-6* also induced Dpn in Wrp+ midline glia (arrowheads). N = 6 VNCs. (**E and F**) When *dilp-6* was overexpressed, and all glia except midline glia were visualised with nuclear *repoGAL4>Histone-*

*Figure 8 continued on next page*

*Figure 8 continued*

YFP and midline glia with anti-Wrp, Dpn+ YFP⁻Wrp⁻ cells were found, which therefore were not glia (white arrows; yellow arrowheads point to Dpn+Wrp+ cells). N = 6 VNCs. (**G and H**) G-TRACE expression in glia with *repoGAL4* revealed with GFP cells that were originally glia or originated from a glial cell lineage, even if they switched off the glial *repo* promoter, and with RFP newly generated glial cells. Dpn colocalised in neuropile glia with both GFP and RFP, meaning that Dpn+ cells originated from glia, and at that point in time these cells still retained active the glial *repo* promoter. N = 8 VNCs. (**A, C, E, and G**) Horizontal and (**B, D, F, and H**) transverse views. For full genotypes and sample sizes, see *Supplementary file 1*.

Elav+, meaning they were neurons (*Figure 10C,D*). Importantly, GFP+ Elav+ cells were found near ectopic Dpn+ cells (*Figure 10C,D*). These data meant that glial-derived Dpn+ cells could produce neurons. We did not find any larger clusters, suggesting that neurogenesis was limited. Pupariation occurs soon after 120 hr AEL, potentially limiting and altering further cellular events.

## Discussion

A critical missing link to understand how to induce CNS regeneration in non-regenerating animals such as humans had been to identify factors that interact with NG2 to induce regenerative neurogenesis. NG2-glia are abundant progenitor cells present throughout life in the adult human brain and can respond to injury (*Dimou and Götz, 2014*; *Torper et al., 2015*; *Valny et al., 2017*). Thus, they are the ideal cell type to manipulate to promote regeneration. However, whether NG2-glia can give rise to neurons is highly debated, and potential mechanisms remained unknown (*Dimou and Götz, 2014*; *Viganò and Dimou, 2016*; *Falk and Götz, 2017*; *Valny et al., 2017*; *Du et al., 2021*). Here, using *Drosophila* in vivo functional genetic analysis we have identified neuronal Ia-2 as a genetic interactor of the NG2 homologue Kon, and show that it can induce a neurogenic response from glial cells via insulin signalling.

We provide evidence that Ia-2, Kon, and Dilp-6 induce a regenerative neurogenic response from glia (*Figure 11*). In the un-injured CNS, Kon and Ia-2 are restricted to glia and neurons, respectively (*Figure 11A*). Ia-2 is required for neuronal Dilp-6 secretion (*Cai et al., 2001*; *Kim et al., 2008*), Dilp-6 is produced by some neurons and mostly glia, and its production depends mostly on Kon regulated glia. Alterations in Ia-2 levels, increased Dilp-6, and concerted activation of Ras or PI3Kinase downstream of insulin signalling induced ectopic neural stem cells from glia. Both loss and gain of *ia-2* function induced ectopic Dpn cells. Ia-2 depends on Pros and in turn negatively regulates Pros. Pros controls the switch from neural stem cell to progenitor state (*Choksi et al., 2006*). In this way, cell–cell interactions involving Ia-2 can influence neural progenitor cell fate. *ia-2* loss of function would also cause a decrease in Dilp-6 secretion from neurons, but not from glia, as *kon* mRNA levels were unaffected, and *dilp-6* expression depends mostly on glial *ko*n. As neuronal Ia-2 and glial Kon mutually exclude each other, perhaps loss of *ia-2* function might increase *kon*-dependent Dilp-6 production. As Ia-2 is required for Dilp-6 secretion (*Cai et al., 2001*; *Harashima et al., 2005*; *Kim et al., 2008*), *ia-2* GOF would increase Dilp-6 release triggering the Dilp-6 amplification loop. Conceivably, either way Dilp-6 increased and this induced Dpn. Upon injury, levels of *kon* (*Losada-Perez et al., 2016*) and *ia-2* expression increased (*Figure 11C*). Ia-2 drives secretion of Dilp-6 from neurons, Dilp-6 is received by glia, and a positive feedback amplification loop drives the further Kon and InR dependent production of Dilp-6 from cortex glia (*Figure 11B, C*). Dilp-6 can then both promote glial proliferation to generate more glia and induce the neural stem cell marker Dpn in neuropile glia – the subset known as '*Drosophila* astrocytes' and midline glia (*Figure 11B,C*). Ectopic Dpn+ cells were induced from glia both upon injury and genetic manipulation of Ia-2, Dilp-6, Ras, and PI3Kinase. Importantly, these glial-derived neural stem cells could divide, as revealed by the S-phase marker PCNA-GFP and the mitotic marker pH3, and could generate neurons, albeit to a rather limited extent. Altogether, Dilp-6 is relayed from neurons to cortex and then to neuropile glia. This neuron-glia communication relay could enable concerted glio- and neuro-genesis, matching interacting cell populations for regeneration (*Figure 11C,D*). Interestingly, Dilp-6 is also involved in non-autonomous relays between distinct CNS cell populations to activate neural stem cells and induce neuronal differentiation in development (*Sousa-Nunes et al., 2011*; *Fernandes et al., 2017*).

We have demonstrated that ectopic neural stem cells originate from glia. Regenerative neurogenesis could occur via direct conversion of glia into neurons, glial de-differentiation, or neuronal de-differentiation. Neuronal de-differentiation occurs both in mammals and in *Drosophila* (*Froldi et al.,*

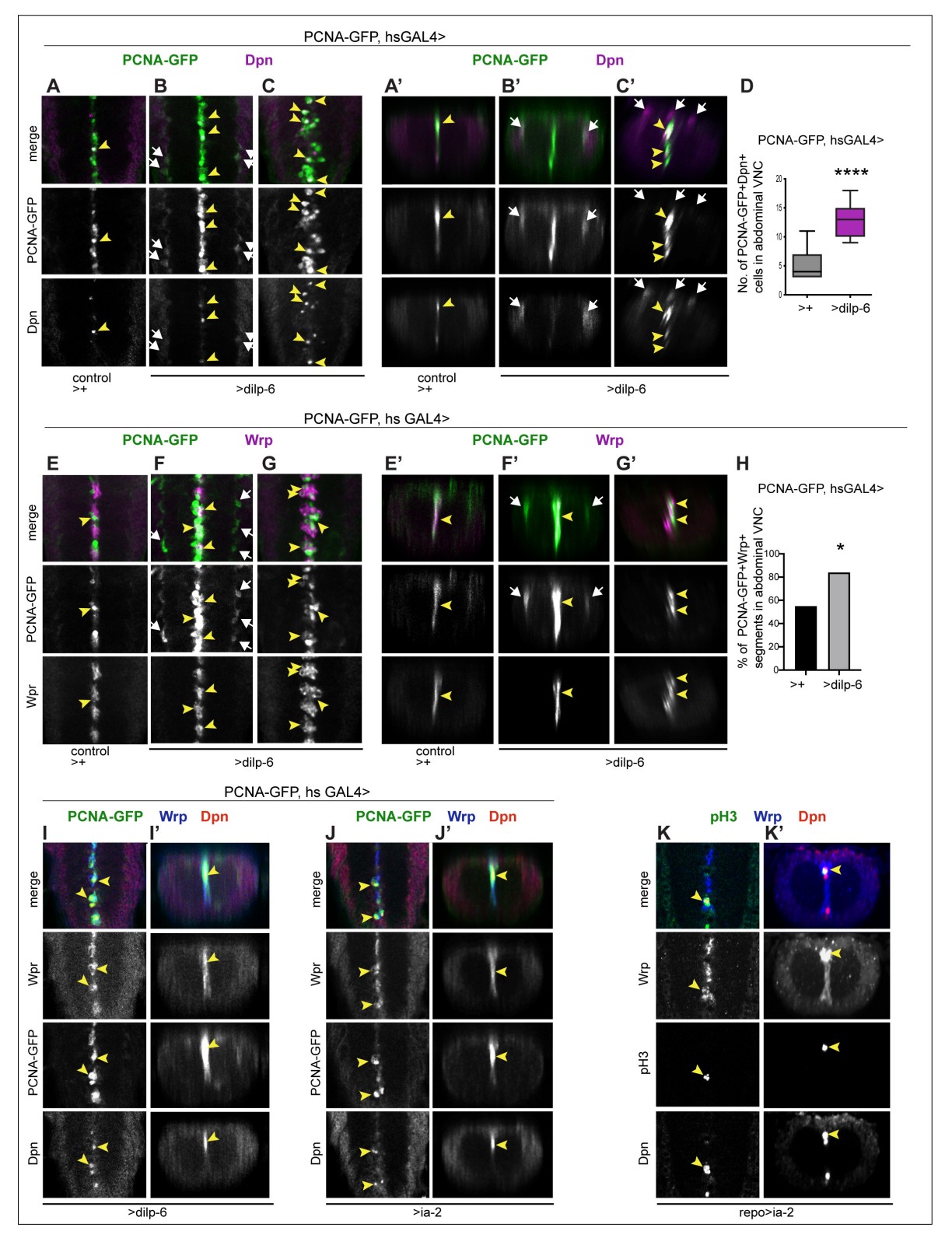

**Figure 9.** Ia-2 and Dilp-6 induced ectopic neural stem cells can divide. All samples were analysed at 120 hr AEL, after disappearance of abdominal developmental neuroblasts. (**A–C' and E–G'**) Cell proliferation was visualised with the S-phase marker PCNA-GFP, quantification in (**D and H**). *dilp-6* expression was induced in all cells with *heat-shock-GAL4*, raising the temperature to 37°C for 30 min at the end of the third instar larval stage at 110.5 hr AEL, and then larvae were kept at 25°C for 9 hr, visualising Dpn+ and PCNA-GFP at 120 hr AEL. (**A–C'**) Overexpression of *dilp-6* resulted in Dpn+

*Figure 9 continued on next page*

*Figure 9 continued*

PCNA-GFP+ cells laterally around the neuropile (**B and B'** white arrows) and along the midline (**C and C'** yellow arrowheads), showing that these ectopic Dpn+ cells were in S-phase. Quantification box-plots in (**D**), Student's t-test. There were also some Dpn+ cells that were not dividing (white arrows in **C'**). (**E–G'**) Overexpression of *dilp-6* resulted in PCNA-GFP+ Wrp+ midline glia (yellow arrowheads) that therefore were dividing. In (**G**) there is a notable increase in the number of Wrp+ cells. In (**E and E'**) lateral PCNA-GFP+Wrp⁻Dpn+ cells around the neuropile (white arrows) most likely correspond to neuropile glia. (**H**) Quantification showing phenotypic penetrance: percentage of segmentally repeated Wrp+ cell clusters that contain PCNAGFP+ cells. Fisher's exact test p=0.0276. (**I–J'**) Overexpression of either *dilp-6* or *ia-2* with *hsGAL4* upregulated the S-phase marker PCNA-GFP in Wrp+ Dpn+ midline cells, meaning these ectopic Dpn+ cells were dividing. Penetrance: >*dilp-6* 25% N = 4; >*ia-2*: 18% N = 11 ventral nerve cords (VNCs). (**K and K'**) Overexpression of *ia-2* in glia with *repoGAL4* induced non-autonomously proliferation of ectopic Wrp+ Dpn+ cells, visualised with the mitotic marker pH3. Penetrance: 60% N = 10 VNCs.

*2015*). However, in most animals, neural stem cells in the adult CNS and upon injury are generally distinct from developmental ones, and can originate from hemocytes, but most often, glial cells (*Tanaka and Ferretti, 2009*; *Dimou and Götz, 2014*; *Falk and Götz, 2017*; *Simões and Rhiner, 2017*; *Du et al., 2021*). In the mammalian brain, radial glia in the hippocampus respond to environmental challenge by dividing asymmetrically to produce neural progenitors that produce neurons (*Shtaya et al., 2018*); and astrocytes and NG2-glia can generate neurons, particularly in response to stroke, excitoxic injury, and genetic manipulations (*Heinrich et al., 2014*; *Dimou and Gallo, 2015*; *Péron and Berninger, 2015*; *Du et al., 2021*). Furthermore, genetic manipulation can lead to the direct conversion of NG2-glia into neurons (*Torper et al., 2015*; *Pereira et al., 2017*). Our findings that Dilp-6 and InR signalling can induce *dpn* expression are reminiscent of their functions in the induction of neural stem cells from quiescent progenitors in development (*Chell and Brand, 2010*; *Sousa-Nunes et al., 2011*; *Gil-Ranedo et al., 2019*). However, the Dpn+ cells induced upon injury and after development are distinct from the developmental neural stem cells normally induced by Dilp-6 in multiple ways. Firstly, in injuries carried out in third instar larvae, the induced neural stem cells were more numerous than normal neural stem cells. Secondly, in injuries carried out late in wandering larvae, Dpn+ cells were found after normal developmental neural stem cells have been eliminated through apoptosis (*Bello et al., 2003*). Thirdly, Dpn+ cells were found in dorsal ectopic locations not normally occupied by developmental neural stem cells. In all injury and genetic manipulation experiments involving overexpression of either *ia-2*, *dilp-6*, or *PI3K*, ectopic Dpn+ cells were located along the midline and surrounding the neuropile, in positions normally occupied by glia. Remarkably, concerted overexpression of *ras* and *dilp-6* induced Dpn in potentially all glial cells and more, consistently with further Dpn+ cell proliferation. Consistent with our findings, ectopic neuroblasts were also observed upon co-expression of activated *ras^{V12}* and knock-down of *PTEN* in glia, within glioma models in *Drosophila* (*Gangwani et al., 2020*). We demonstrated that ectopic Dpn+ originated from glia, most particularly neuropile glia (midline glia and '*Drosophila* astrocytes'). Firstly, ectopic Dpn+ cells did not have Ia-2YFP, which is expressed in all neurons. Secondly, overexpression of *ia-2* or *dilp-6*, alone or in combination with *ras* and *PI3K*, in glia dramatically increased Dpn levels, meaning that insulin signalling induces *dpn* expression in glia. Thirdly, ectopic Dpn+ cells surrounding the neuropile occupied positions of astrocytes and had the pan-glial marker Repo, and Repo⁻ Dpn+ along the midline had the midline glia marker Wrp. Fourthly, the glial origin of the ectopic Dpn+ cells was demonstrated using two cell-lineage tracing methods (G-TRACE and glial activation of the *actin* promoter) whereby the expression initiated from the glia *repo* promoter was turned permanent despite cell state transitions. Consistently with our findings, TRAP-RNA analysis of the normal third instar larva revealed expression of *dpn* and multiple genes involved in neuroblast polarity, asymmetric cell division, neuroblast proliferation, and neurogenesis in glia (*Huang et al., 2015*). And single cell RNAseq analysis of the larval CNS revealed that in normal larvae some Repo+ glial cells can express *dpn*, or other neuroblast markers like *wor* and *ase* (*Brunet Avalos et al., 2019*). Our findings show that basal or potential expression of neuroblast genes in glia is switched on and amplified by insulin signalling. We conclude that Ia-2 and Dilp-6 could reprogramme glial cells in vivo into neural stem cells.

Our data showed that the ectopic *ia-2* and *dilp-6* induced neural stem cells could divide and generate neurons. In fact, concomitant overexpression of *dilp-6* and *PI3K*, and most prominently *dilp-6* and *ras*, dramatically increased Dpn+ cell number. Dilp-6 induced glial-derived Dpn+ cells could express the S-phase marker PCNA-GFP, and Ia-2 induced Wrp+ Dpn+ cells that were pH3+ in

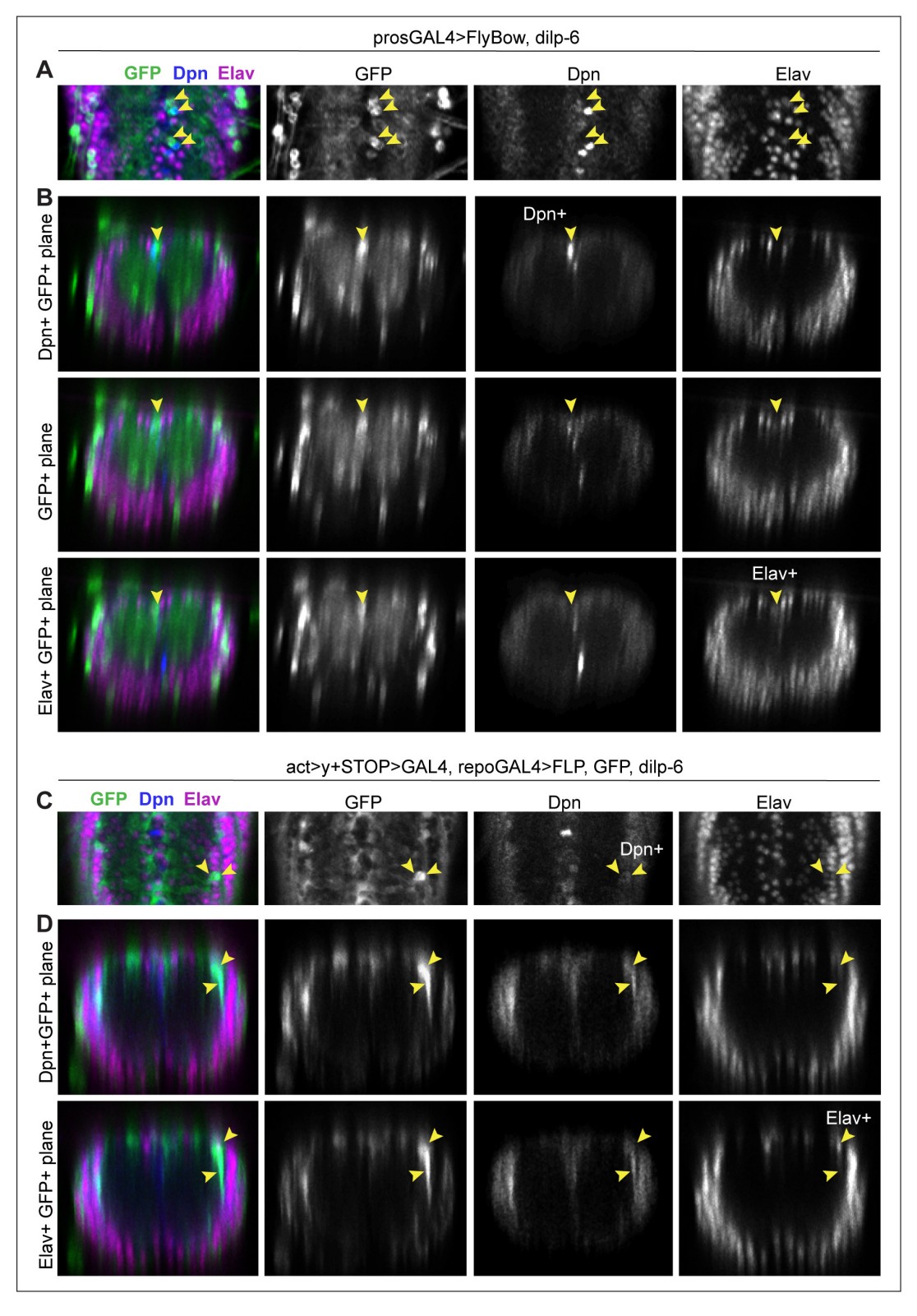

**Figure 10.** Neurons were detected from glial-derived neural stem cells. All samples were analysed at 120 hr AEL, after disappearance of abdominal developmental neuroblasts. (**A and B**) prosGAL4>FlyBow can reveal expression of neural stem cells, ganglion mother cells, neurons, and glia. Overexpression of *dilp-6* with *prosGAL4* resulted in clusters of 3 GFP+ cells along the midline that comprised one GFP+Dpn+ neural stem cell (top row in **B**), a GF+Dpn⁻Elav⁻ progeny cell (middle row, **B**) and one GFP+Elav+ progeny neuron (bottom row, **B**). (**C and D**) Progeny cells of a glial cell-lineage

*Figure 10 continued on next page*

*Figure 10 continued*

were visualised with GFP, expressed originally under the control of the glial *repo* promoter, then switched using Flipase, to the permanent *actin* promoter activated only in glial cells (*act>y+STOP>UAS-GFP/UAS-FLP; repoGAL4/UAS-Dilp-6*). Overexpression of *dilp-6* resulted in clusters of two to three GFP+ cells that comprised a GFP+Dpn+ neural stem cell and two progeny GFP+Elav+ neuronal progeny cells. (**A and C**) Horizontal and (**B and D**) transverse views. For full genotypes, sample sizes, and statistical details, see *Supplementary file 1*.

mitosis. We could not detect mitotic cells surrounding the neuropile, but mitosis is brief, and could have easily been missed. The Dilp-6 induced ectopic Dpn+ cells could generate neurons that could be traced with GFP expression from their glial origin. Thus, ectopic neural stem cells induced by Dilp-6 can divide and produce neuronal progeny cells. However, the clusters of GFP+ cells originating from the in vivo reprogrammed glial cells were rather small, indicating that although neurogenesis was possible in late larvae, it was extremely constrained. This could be due to the fact that in the third instar larva, time is rather limited by pupariation. Injury and genetic manipulation in late larvae may not allow sufficient time for cell lineages to progress, before pupariation starts. Pupariation and metamorphosis bring in a different cellular context, which could interfere with regenerative neuronal differentiation. Alternatively, Ia-2 and Dpn may not be sufficient to carry neurogenesis through either. For instance, gain of *ia-2* function resulted only in Dpn+ but not Pros+ or Eve+ cells, suggesting that Ia-2 and Dpn are not sufficient for neuroblasts to progress to GMCs and neurons. In fact, ectopic Dpn+ cells still had Repo. Furthermore, we did not detect other ectopic neuroblast markers, such as Wor or Ase in glia. Nevertheless, RNA seq data revealed expression of neuroblast markers, including *dpn, wor,* and *ase* in some glia in normal larval CNS, meaning they could potentially be further regulated (*Brunet Avalos et al., 2019*; *Huang et al., 2015*). Still, to generate neurons, glia may not only require the expression of neural stem cell markers like *dpn*, but also perhaps receive other yet unknown signals (*Figure 11B*). In mammals, injury creates a distinct cellular environment that prompts glial cells to generate different cell types than in the un-injured CNS. For instance, elevated Sox-2 is sufficient to directly reprogramme NG2-glia into neurons, but only upon injury (*Heinrich et al., 2014*). Whereas during normal development NG2-glial cells may only produce oligodendrocyte lineage cells, upon injury they can also produce astrocytes and neurons (*Dimou and Gallo, 2015*; *Huang et al., 2018*). This suggests that there are injury-induced cues for neuronal differentiation. In the future, it will be compelling to find out what signals could enhance neurogenesis from glial cells reprogrammed in vivo by insulin signalling.

Our work has revealed a novel molecular mechanism driving a regenerative neurogenic response from glia, involving Kon/NG2 and insulin signalling. Ia-2 induces an initial secretion of Dilp-6 from neurons, Dilp-6 is received by glia, and a positive feedback loop amplifies the Kon-dependent production of Dilp-6 by cortex glia, Dilp-6 is then relayed to neuropile glia, resulting in the in vivo reprogramming of glial cells into neural stem cells (*Figure 11C*). This mechanism can induce both glial regeneration and neural stem cells from glia, potentially also neurons, matching interacting neuronal and glial cell populations. The incidence of neuropile glia conversion to Dpn+ cells was variable, meaning the process is stochastic. However, all glia converted when activated Ras or PI3K were combined with Dilp-6, meaning levels of insulin signalling matter. Such a mechanism may also operate in mammals. In fact, Ia-2 has universal functions in dense core vesicles to release insulin (*Cai et al., 2001*; *Harashima et al., 2005*; *Kim et al., 2008*; *Nishimura et al., 2010*; *Cai et al., 2011*). Insulin-like growth factor 1 (IGF-1) induces the production of astrocytes, oligodendrocytes, and neurons from progenitor cells in the adult brain, in response to exercise (*Nieto-Estévez et al., 2016*; *Mir et al., 2017*). The transcription factor Sox-2 that can switch astrocytes to neural stem cells and produce neurons is a downstream effector of InR/AKT signalling (*Mir et al., 2017*). NG2 also interacts with downstream components of the InR signalling pathway (e.g. PI3K-Akt-mTOR) to promote cell cycle progression and regulate the expression of its downstream effectors in a positive feedback loop (*Sakry et al., 2015*; *Nayak et al., 2018*). Together, all of these findings indicate that Ia-2, NG2/Kon, and insulin signalling have a common function across animals in reprogramming glial cells into becoming neural stem cells.

Intriguingly, *dpn* was mostly induced in neuropile associated glial cells and was only induced in other glial types with overexpression of active Ras$^{V12}$ together with Dilp-6. Thus, perhaps prominently neuropile glia have neurogenic potential. Of the neuropile glia, *Drosophila* 'astrocytes' and midline glia express *Notch, pros,* and *kon*, as well as *InR*. The cells frequently called 'astrocytes'

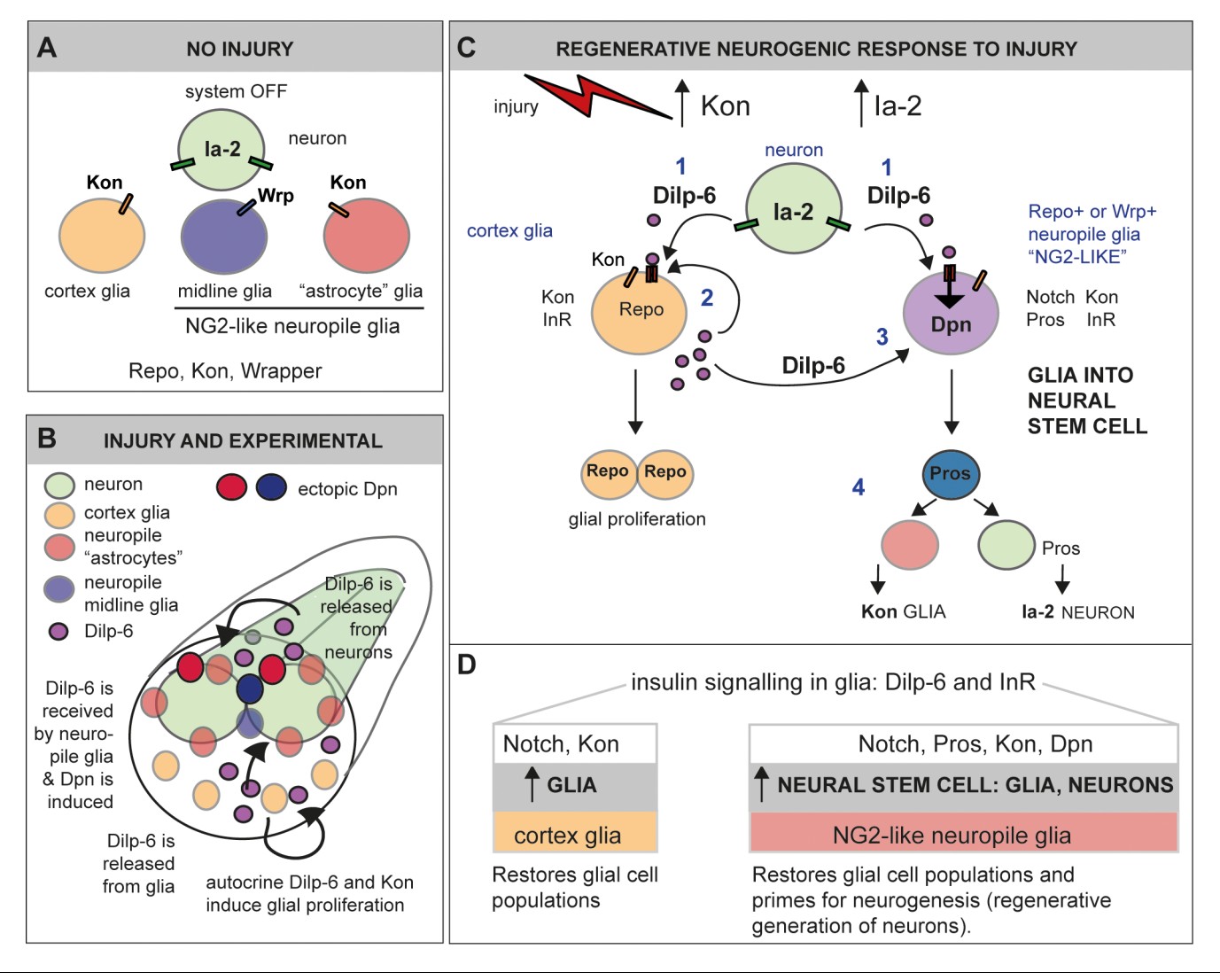

**Figure 11.** Ia-2 and Dilp-6 drive a regenerative neurogenic response to central nervous system (CNS) injury. (**A**) In the abdominal larval ventral nerve cord (VNC), neurons have Ia-2, glia have Kon, and Ia-2 and Kon are mutually exclusive; non-midline glia have the transcription factor Repo and midline glia the membrane protein Wrapper. In the normal, uninjured abdominal VNC, InR is in glial cells and some neurons; Ia-2 expression is constantly present in neurons; *kon* is switched off, and there are no neural stem cells (neuroblasts). (**B**) Diagram showing that Dilp-6 can be secreted from neurons, amplified and secreted by cortex glia, and received by all glial types. Dilp-6 production and secretion depend on Kon and Ia-2, which increase in injury. (**C**) Injury to the abdominal VNC provokes a dramatic surge in Ia-2 and Kon. This drives the initial secretion of Dilp-6 from neurons (1). Secreted Dilp-6 binds InR in glia, and InR signalling may facilitate cleavage and activation of Kon. Kon^ICD activates glial proliferation (2). In an autocrine Kon and InR dependent manner, Dilp-6 sets off a positive feedback loop that amplifies Dilp-6 production from cortex glia (2). Once secreted, Dilp-6 and InR signalling cause the upregulation of Dpn+ in neuropile glia – including Notch+ Pros+ lateral (astrocytes) and Wrp+ midline glia (3). Neuropile glia can stochastically switch on Dpn. Glial-derived Dpn+ neural stem cells can divide and generate new neurons – although to a rather limited extent (4). After cell division, Kon may determine whether daughter cells become glia, to the exclusion of Ia-2. (**D**) Insulin signalling involving Ia-2, Dilp-6, and InR can increase cell number of various glial cell types – including cortex glia, neuropile astrocytes and midline glia – induce neural stem cells, and potentially generate new neurons. The neurogenic potential of glia may depend on the availability of Notch and Pros and the downregulation of Kon. Together, these genes can potentially induce neurogenesis and gliogenesis, matching cell populations for regeneration.

share features with mammalian NG2-glia (*Losada-Perez et al., 2016*; *Hidalgo and Logan, 2017*; *Kato et al., 2018*). In mammals, the combination of Notch1, Prox1, and NG2 is unique to NG2-glia and is absent from astrocytes (*Cahoy et al., 2008*). Perhaps Ia-2 and Dilp-6 can only induce neural stem cells from NG2-like glia bearing this combination of factors. Notch activates glial proliferation and *kon* expression in *Drosophila* (*Losada-Perez et al., 2016*), and in the mammalian CNS, Notch

promotes NG2-glia proliferation and maintains the progenitor state (*Yamamoto et al., 2001*; *Ables et al., 2010*; *Piccin et al., 2013*; *Falk and Götz, 2017*). In *Drosophila*, Notch and Pros also regulate *dpn* expression: Notch activates *dpn* expression promoting stemness, and Pros inhibits it, promoting transition to GMC and neuron (*Vaessin et al., 1991*; *San-Juán and Baonza, 2011*; *Babaoğlan et al., 2013*; *Bi and Kuang, 2015*). Thus, only glial cells with Notch and Pros may be poised to modulate stemness and neuronal differentiation. We showed that *InR* is expressed in neuropile glia, which was confirmed by publically available single cell RNAseq data (*Brunet Avalos et al., 2019*). Insulin signalling represses FoxO, which represses *dpn,* and thus ultimately activates *dpn* expression (*Siegrist et al., 2010*). As Notch and insulin signalling positively regulate *dpn* expression (*Vaessin et al., 1991*; *Siegrist et al., 2010*; *San-Juán and Baonza, 2011*; *Babaoğlan et al., 2013*; *Bi and Kuang, 2015*), and injury induces a Notch-dependent upregulation of Kon (*Losada-Perez et al., 2016*), which enables *dilp-6* expression, and of Ia-2, which secretes Dilp-6, our data indicate that Notch-Kon/NG2-insulin synergy triggers the activation of *dpn* expression. Importantly, we found no evidence that Kon functions in neural stem cells. Thus, perhaps induced neural stem cells can generate only glia from daughter cells that inherit Kon, on which Repo and glial cell fate depend, or generate neurons, from daughter cells that lack Kon, but have Pros, on which Ia-2 depends (*Figure 11C*). Thus, upon injury, Notch, Pros, Kon/NG2, Ia-2, and insulin signalling function together to enable the regenerative production of both glial cells and neural stem cells from glia (*Figure 11C,D*). Intriguingly, developmental neural stem cells are thought to be eliminated through upregulation of Pros, induction of cell cycle exit, and terminal differentiation into glia (*Maurange et al., 2008*). Our findings imply that such termination may not be final.

To conclude, a neuron-glia communication relay involving Ia-2, Dilp-6, Kon, and InR is responsible for the induction of neural stem cells from glia, their proliferation, and limited neurogenesis. Neuronal Ia-2 and Dilp-6 trigger two distinct responses in glia: (1) in cortex glial cells, insulin signalling boosts Kon-dependent amplification of Dilp-6, glial proliferation, and glial regeneration. (2) In neuropile-associated NG2-like glial cells, insulin signalling unlocks a neurogenic response, inducing neural stem cell fate. As a result, these genes can drive the production of both glial cells and neurons after injury, enabling the matching of interacting cell populations, which is essential for regeneration.

## Materials and methods

### Fly stocks and genetics

Fly stocks used are listed in Key Resources Table. Stocks carrying combinations of overexpression and RNAi, or RNAi and mutants, etc., were generated by conventional genetics. $N^{ts}$ mutants were raised at 18°C to enable normal embryogenesis and switched to 25°C from larval hatching to the third instar larval wandering stage to cause N loss of function. For experiments in *Figure 7B–D,F,G, I,J,L,M*, eggs from crosses to *tubGAL80^{ts} repoGAL4* were kept at 18°C for 48 hr and moved to 30°C after L1 larval hatching. For all experiments, larvae bearing balancer chromosomes were identified by either using the fluorescent balancers CyO Dfd-YFP and TM6B Dfd-YFP or using the fused balancer SM6a- TM6B Tb⁻, which balances both the second and third chromosomes, and discarded. For the genetic screens, larvae with fluorescent VNCs (i.e. *repoGAL4>UAS-FlyBow* or *elavGAL4>UAS-FlyBow*) were selected.

### Crush injury in the larval VNC

Crush injury in the larval CNS was carried out as previously reported (*Losada-Perez et al., 2016*), and only lesions in the abdominal VNC were analysed. Larval collections were staged by putting the G0 flies in an egg laying chamber for 2 hr, then collecting the F1 larve some time later, as indicated next. Larvae were placed on a chilled petri-dish with agar over ice. Crush injury was carried out by pinching with fine forceps the GFP-bearing VNCs under UV light using a fluorescence dissecting microscope: (1) at 74–76 hr AEL; VNCs were then left to carry on developing at 25°C and were dissected either 5–7 hr or 24 hr post-injury (PI); (2) at 96 hr, kept at 25°C and dissected and fixed 6 hr PI; (3) at 105 hr AEL, kept at 25°C and dissected 24 hr PI; (4) at 117 hr AEL, kept at 25°C, and dissected 12 hr PI. Dissected and fixed VNC were then processed for antibody stainings following standard procedures.

## Molecular cloning

The UAS-konICD-HA construct was generated from EST LD31354 via PCR amplification with Kappa HiFi PCR kit (Peqlab) and subsequent cloning using the Gateway cloning system (Invitrogen) according to manufacturer's instructions. Primers used were kon[ICD] fwd comprising the CACC-sequence at the 5'-end (CCACAGGAAACTGAGAAAGCACAAGGC) for direct cloning of the PCR product (482 bp) into the entry vector pENTR/D-Topo, and kon[ICD] rev (AAACCTTACACCCAATACTGATTCC) including the endogenous stop-codon, underlined. Destination vector was pTHW for tagging the ICD on the N-terminus with HA, including a 5xUAS cassette and P-element ends for transformation. These destination vectors were developed by the Murphy-Lab at Carnegie Institution of Science, Baltimore, MD, USA, and can be obtained from the Drosophila Genomics Resource Center at Indiana University, USA. Transformant fly strains were generated by BestGene Inc, Chino Hills, CA, USA following a standard tranposase-mediated germline transformation protocol.

A UAS-ia-2 construct was generated using Gateway cloning (Invitrogen, as above). Ia-2 cDNA was generated by reverse-transcription PCR of purified mRNA from Oregon R flies and cloned into pDONR. Subsequently, a standard PCR amplification was performed using Phusion High-Fidelity (Fisher Scientific), primers Ia-2F (5'-ATGGCACGCAATGTACAACAACGGC) and ia-2-stopR (5'-CTTCTTCGCCTGCTTCGCCGATTTG), and the resulting PCR product (3918 bp) was cloned into pGEM-T Easy Vector (Promega). Subsequently, a Phusion High Fidelity PCR amplification was carried out using Gateway primers Ia-2attB F1 (5'-ggggacaagtttgtacaaaaaagcaggcttcATGGCACGCAATGTACAACAACGGC) and Ia-2attB R1 (5'-ggggaccactttgtacaagaaagctgggtcCTTCTTCGCCTGCTTCGCCGATTTG), and plasmid pGEM-ia-2 as template. Using Gateway cloning, the PCR product (3979 bp) was cloned first into pDONR[221] and subsequently into the pUAS-gw-attB destination vector, for $\phi$C31 transgenesis. The construct was injected by BestGene Inc to generate transgenic flies bearing UAS-ia-2 at the attP2 landing site.

## Quantitative real time reverse transcription PCR (qRT-PCR)

qRT-PCR was preformed according to standard methods and as previously described (*Losada-Perez et al., 2016*), with the following alteration. For each sample, 10 third instar larvae were used per genotype per replicate. At least three independent biological replicates were performed for all experiments other than in *Figure 1—figure supplement 3A and B* where two replicates were carried out on all candidates and those of interest where taken forward to carry out two further replicates. For a list of the primers used in this study please see Key Resources Table below.

## Immunostainings

Immunostainings were carried out following standard procedures, with the following modifications. For anti-Dpn, we used: (1) 20 min dissection in PBS into 4% formaldehyde on ice followed by 45 min fixation at room temperature (anti-Dpn from J. Skeath); (2) 5 min dissection in PBS into 4% formaldehyde on ice, followed by 10 min fixation at room temperature (anti-Dpn from H. Wang) (*Huang and Wang, 2018*; *Zhang et al., 2019*). The following primary antibodies were used: mouse anti-Repo (1:100, DSHB); guinea pig anti-Repo (1:1000, Ben Altenhein); rat anti-Elav (1:250, DSHB); mouse anti-FasII ID4 (1:500, DSHB); mouse anti-Prospero (1:250, DSHB); guinea pig anti-Dpn (1:1000, gift of J. Skeath); guinead pig anti-Dpn (1: 1000 gift of H. Wang); mouse anti-Eve 3C10 (1:20, DSHB); rabbit anti-phospho-histone-H3 (1:250); rabbit anti-HA (1:1600, Cell Signalling Technology); rabbit anti-GFP at 1:250 (Molecular Probes); mouse anti-Wrapper (1:5, gift of G. Tear). Secondary antibodies were Alexa conjugated: Donkey anti-rabbit 488 (1:250, Molecular Probes), goat anti-rabbit 488 (1:250, Molecular Probes), goat anti-rabbit 647 (1:250, Molecular Probes), goat anti-mouse 488 (1:250, Molecular Probes), goat anti-mouse 546 (1:250, Molecular Probes), goat anti-mouse 647 (1:250, Molecular Probes), goat anti-rat 546 (1:250, Molecular Probes), goat anti-guinea pig 488 (1:250, Molecular Probes), goat anti-guinea pig 633 (1:250, Molecular Probes), and goat anti-rat 647 and 660 (1:250, Molecular Probes).

## Microscopy and imaging

Image data were acquired using a Zeiss LSM710 laser scanning confocal microscope, with 25× lens and 1.00 zoom, an Olympus Fluoview FV1000, 20× lens, and a Leica SP8 laser scanning confocal microscope, with a 20× lens, 1.25 zoom. All images were taken with resolution 512 × 512 or 1024 ×

1024, step 0.96 µm and 1–3× averaging for all samples except for cell counting with DeadEasy that have no averaging.

Images were analysed using ImageJ. Images of horizontal sections are projections from the stacks of confocal images that span the thickness of the entire VNC, using ImageJ. Transverse views were generated using the Reslice option. Images were processed using Adobe Creative Suite 6 Photoshop and compiled with Adobe Illustrator.

## Automatic cell counting

Glial cells labelled either with anti-Repo or with repoGAL4>UAShistone-YFP were counted automatically in 3D across the thickness of the VNC using DeadEasy Larval Glia software, as previously described. Prospero+ and Dpn+ cells were counted manually in 3D (i.e. not in projections), as the signal was noisy for DeadEasy.

## Statistical analysis

Statistical analysis was carried out using Graphpad Prism. All data in this work are continuous, except for the PCNAGFP data in *Figure 9H* which are categorical. The latter were analysed with a non-parametric Fisher's exact test. For all other data, tests to determine whether data were distributed normally and variances were equal were initially carried out, and thereafter if so, parametric one-way ANOVA tests were carried out when comparing more than two sample types group. Multiple comparison corrections were carried out with post hoc Dunnett tests comparisons to set controls, or Bonferroni comparisons of all samples against all. Box plots were used to represent the distribution of continuous data, where the line within the box represents the median of the data distribution, the box comprises the 25 percentiles above and below the median, and the whiskers the lowest and highest 25 percentiles.

# Acknowledgements

We thank our labs and C Rezaval for discussions and comments on the manuscript; S Corneliussen, T Schunke, and S Dietz for technical help; Y Fan, A Gould, Y Jan, J Skeath, F Schnorrer, and H Wang for reagents; A Di Maio and Birmingham Advanced Light Microscopy for assistance; Bloomington *Drosophila* Stock Centre for fruit-flies and Developmental Studies Hybridoma Bank, Iowa for antibodies.

# Additional information

## Funding

| Funder | Grant reference number | Author |
| --- | --- | --- |
| Biotechnology and Biological Sciences Research Council | BB/L008343/1 | Neale J Harrison Marta Moreira Alicia Hidalgo |
| Biotechnology and Biological Sciences Research Council | BB/R00871X/1 | Marta Moreira Alicia Hidalgo |
| Biotechnology and Biological Sciences Research Council | MIBTP Studentship | Elizabeth Connolly |
| MSCA | TOLKEDA | Jun Sun Alicia Hidalgo |

The funders had no role in study design, data collection and interpretation, or the decision to submit the work for publication.

## Author contributions

Neale J Harrison, Conceptualization, Data curation, Formal analysis, Supervision, Validation, Investigation, Visualization, Methodology, Writing - review and editing; Elizabeth Connolly, Data curation, Formal analysis, Validation, Investigation, Visualization, Methodology, Writing - review and editing; Alicia Gascón Gubieda, Data curation, Formal analysis, Validation, Investigation, Visualization,

Writing - review and editing; Zidan Yang, Formal analysis, Validation, Investigation, Visualization, Writing - review and editing; Benjamin Altenhein, Conceptualization, Data curation, Formal analysis, Supervision, Validation, Investigation, Visualization, Writing - review and editing; Maria Losada Perez, Resources, Methodology, Writing - review and editing; Marta Moreira, Resources, Data curation, Methodology, Writing - review and editing; Jun Sun, Formal analysis; Alicia Hidalgo, Conceptualization, Resources, Formal analysis, Supervision, Funding acquisition, Investigation, Methodology, Writing - original draft, Project administration, Writing - review and editing

### Author ORCIDs
Neale J Harrison https://orcid.org/0000-0001-6821-4089
Elizabeth Connolly https://orcid.org/0000-0002-5716-8889
Marta Moreira https://orcid.org/0000-0002-4779-4077
Jun Sun https://orcid.org/0000-0002-1539-9937
Alicia Hidalgo https://orcid.org/0000-0001-8041-5764

### Decision letter and Author response
Decision letter https://doi.org/10.7554/eLife.58756.sa1
Author response https://doi.org/10.7554/eLife.58756.sa2

## Additional files

### Supplementary files
• Supplementary file 1. Genotypes, sample sizes, and statistical analysis details. This table contains full genotypes for all experiments, sample sizes used, and statistical analysis details including normality tests, tests applied, and multiple comparison correction tests.

• Transparent reporting form

### Data availability
All data generated or analysed during this study are included in the manuscript and supporting files.

The following previously published datasets were used:

| Author(s) | Year | Dataset title | Dataset URL | Database and Identifier |
|---|---|---|---|---|
| Avalos CB, Maier GL, Bruggmann R, Sprecher SG | 2019 | Single cell transcriptome atlas of the *Drosophila* larval brain | https://www.ncbi.nlm.nih.gov/geo/query/acc.cgi?acc=GSE134722 | NCBI Gene Expression Omnibus, GSE134722 |

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
