## [Decision Letter]

**Acceptance summary:**

The observation that Ia-2 and Dilp-6 drive a neuron-glia relay that restores glia, and reprograms glia into neural stem cells for regeneration is a valuable contribution that will spark interest in the fly neurobiology community.

**Decision letter after peer review:**

[Editors’ note: the authors submitted for reconsideration following the decision after peer review. What follows is the decision letter after the first round of review.]

Thank you for choosing to send your work, "Regenerative neurogenesis is induced from glia by Ia-2 driven neuron-glia communication", for consideration at *eLife*. Your article has been reviewed by three peer reviewers, and the evaluation has been overseen by Hugo Bellen as the Reviewing Editor and Utpal Banerjee as the Senior Editor. Although the work is of interest, we regret to inform you that the findings at this stage are too preliminary for further consideration at *eLife*.

The reviewers feel that several conclusions made throughout the manuscript are not fully supported by data presented, and some conflicting results need to be addressed or resolved. Several reviewers also recommended that there is the backbone of interesting story but the number of experiments requested cannot be performed in a few months. Finally, we also found the manuscript difficult to read and it really needs editing. The reviewers also mention this. We did not feel 2 months is adequate time to address all of the reviewers' concerns. If you are willing to address most of the reviewers comments and resubmit the manuscript along with a detailed rebuttal letter, we will send this work back to the current reviewers. Please note that the reviewers would like all their comments addressed and the resubmission option is the same as a new submission except that we will try our best to find the same reviewers.

Reviewer #1:

In this manuscript, Harrison et al. identify a novel molecular mechanism involving islet antigen (Ia-2), Kon/NG2 and Dilp-6 to induce regenerative neurogenesis from glia. Through a genetic screen and qRT-PCR analysis the authors clearly demonstrate Ia-2 as a functional neuronal partner of Kon and that Ia-2 can participate in the Kon, Notch and Pros gene network that drives a regenerative response to CNS injury. Using crush injury experiments, they further show that Ia-2 levels are significantly upregulated during injury and changes in Ia-2 levels induced the formation of ectopic neural stem cells. Interestingly, these ectopic neural stem cells originated from glia. Finally, they demonstrate a neuron-glia communication loop involving Ia-2, Dilp-6,Kon and InR. Ia-2 was capable of inducing the secretion of Dilp-6 from neurons, which is taken up by glia and amplified through a kon-dependent positive feedback loop to induce both glial regeneration and neural stem cell proliferation.

Overall, this is an very interesting study with novel findings on regenerative neurogenesis. The experiments are well performed and supported by proper quantification. However, the following points may strengthen the manuscript.

1) The authors show that both ia-2 gain of function and loss of function in neurons increased the number of abdominal VNC Dpn+ cells and ia-2 was up-regulated during injury. The fact that both the ia-2 lof and gof resulted in a similar phenotype seems counter-intuitive. It would be helpful if the authors can provide an explanation for this.

2) Over-expression of full length kon in neurons and glia resulted in the down-regulation of ia-2 mRNA levels but had no effect on dpn or dilp-6 expression. It would be interesting to know whether over-expression of konICD could cause a change in dpn or dilp-6 expression.

3) Similarly, since InR signaling can activate kon cleavage downstream in glia, could over-expression of konICD induce ectopic Dpn+ cells?

4) In general, the writing needs to be improved. Certain sentences were long and hard to follow which made the reading difficult.

Reviewer #2:

This manuscript addresses an important problem – how to achieve neuronal regeneration in humans – using an excellent model system, *Drosophila*, where there are superb tools for identifying and characterizing signaling pathways.

The manuscript jumps off from the observation that the evolutionarily conserved transmembrane/nuclear glial protein Kon-tiki (fly)/NG2 (mouse) may promote "regenerative neurogenesis" via an unknown ligand.

Overall the paper is clearly written and the figures are appropriate.

I have a single major comment that affects the whole paper, and multiple minor comments. Overall, the paper needs additional experiments before resubmission.

1) Throughout the paper conclusions are not fully supported by the data shown. Most results paragraphs end with a conditional conclusion, but then in the headers and subsequent sections it is assumed to be proven. The manuscript would be strengthened by either dampening down conclusions to match the data, or by adding additional data to justify the conclusions. A major example is the claim of “regenerative neurogenesis” in the title, Abstract, and Results… yet there are no data presented showing the production of newly-induced neurons – only increased Dpn+ cells (without showing these cells express other stem cell markers, proliferation markers, or produce neuronal progeny).

2) Calling kon and ia-2 "partners" suggests a direct physical interaction. This has not been shown. There could be a dozen genes acting between these two. Please add direct interaction experiments or use wording that accurately reflects the unknown relationship between the two proteins. Multiple examples in the text; where it is stated "ia-2 is a functional partner of Kon."

3) Figure 2F,G says "ectopic Pros+ cells might be GMCs or neurons" but this result should be resolved by staining for Elav protein (neurons) and Asense (neuroblasts and GMCs) and Worniu (neuroblasts).

4) The text says loss of Ia-2 "destabilizes cell fate" but this statement is extremely vague; it seems to hide the fact that the actual phenotype is not known.

5) The use of ectopic Dpn+ cells to indicate “induced neurogenesis” is premature. The authors at a minimum need to assay the ectopic Dpn+ cells for other neuroblast (stem cell) markers (e.g. Wor, Ase) to validate stem cell identity, and PH3 or PCNA to validate their proliferative state. To make their conclusion solid it would be ideal to induce clones containing these ectopic Dpn+ cells and show the clone includes neurons.

6) Figure 3D is unconvincing. The images show large speckles in the Dpn channel that are similar to the “ectopic Dpn+ cells” plus the number of ectopic Dpn+ cells per brain is not quantified (that I could find).

7) Either show data or cite prior work showing that InR-gal4 mimics the InR endogenous expression pattern.

8) Figure 5B. The increased number of glia is not convincing. All panels look the same to me. The statistics show much greater significance between panels than I trust. It would be good to give the raw numbers in a supplement so that the statistics could be validated.

Reviewer #3:

The manuscript by Harrison et al. describes a neuronal-glial communication loop that requires Ia-2, Dilp-6, InR and kon and in some circumstances, can induce the formation of Dpn+ cells from glia. Overall, the manuscript presents an interesting signaling relationship between neurons and glia that could influence neurogenesis. However, the manuscript was not easy to read or interpret and covers many details not essential for the final conclusions. Significant revisions are required to clarify and simplify the manuscript before publication. In addition, ensure that conclusions are appropriate for the data shown and do not overreach. Finally, specific suggestions are below.

1) The authors state that kon overexpression (which has a striking VNC overgrowth phenotype) decreases the transcript of ia-2 (not significantly?) but RNAi of ia-2 suppresses kon_on_ overexpression phenotypes, which is suspicious. Do multiple RNAi lines for ia-2 give the same suppression of kon overexpression? How about the other ia-2 phenotypes presented throughout the manuscript? It is possible that off target affects could account for these phenotypes and a second RNAi line should be used to verify ia-2 phenotypes. Since the authors start with kon overexpression phenotypes in the VNC and suppression, the reader is left with a conflict that is not resolved. How does ia-2 knockdown suppress kon overexpression phenotypes, which reduces the levels of ia-2 in the first place? If the authors choose the keep this data in the manuscript, they must address this discrepancy.

2) For Figure 1A and supplemental, with what genotype were the samples normalized/compared? The authors show the control in B but not A. Conclusions are made that are not necessarily supported. Since the authors state that kon overexpression only marginally decreased ia-2 mRNA levels, too much emphasis is placed on this result, which leaves the reader with concerns mentioned in 4, above. Knockdown of kon may not specifically regulate Ia-2 mRNA levels but may be the result of extra Ia-2 positive cells shown in Figure 2.

3) Pros does not simply activate ia-2 as removal of pros or Pros expression causes ia-2 mRNA increase. This is difficult to interpret and is not a simple ARROW (below 1D). Perhaps analysis of Notch and Pros is unnecessary here as it complicates the message of the manuscript.

4) When analyzing loss of ia-2, sometimes Df plus RNAi is used, sometimes only RNAi is used. Please be consistent with the strategy to remove ia-2 if you wish to compare phenotypes.

5) What is the difference between Figure 3B and D? You provide numbers for controls in the text, so it is unclear why numbers are not provided for the injury model. Also, what is the percentage that lack Ia-2YFP?

6) The authors make strong statements about RNA changes in ia-1 LOF and GOF models (Figure 4E-F), but no changes are statistically significant. Perhaps this data belongs in supplemental or not include at all.

7) Is Figure 5E the same stage as 2J? If so, the controls have very different number of Dpn + cells. If this is due to the genetic background or the GAL4, it is not the correct driver to use.

8) In the section "Ia-2 and Dilp6 can induce neural stem cells from glia", it does not seem that everything is included or referenced correctly. I don't see images of "repoGAL4>UAS-ia-2, UAS-dilp-6-RNAi", "repoGAL4>UAS-ia-2, UAS-konRNAi", or "repoGAL4>UAS-dilp-6, UAS-InRDN" even though 6A-D is referenced for each genotype. Ideally, the authors should control for GAL4 dilution effects and use the same ia-2-YFP in all comparisons (epoGAL4>UAS-dilp-6, UAS-InRDN does not have this transgene).

9) Dpn staining in Figure 7 is not clear. All arrows do not convincingly point to a Dpn + cell because of the high background. Use a different color for Dpn in C/D as blue cannot be seen. Better Dpn staining is presented in other figures. As this experiment is important, perhaps an image with less background or better Dpn staining can be shown.

10. What is the direct evidence that there is Ia-2 regulated Dilp-6 signaling from neurons to glia?

[Editors’ note: further revisions were suggested prior to acceptance, as described below.]

Thank you for submitting your article "Regenerative neurogenic response from glia requires insulin driven neuron-glia communication" for consideration by *eLife*. Your article has been reviewed by three peer reviewers, and the evaluation has been overseen by a Reviewing Editor and Utpal Banerjee as the Senior Editor. The following individual involved in review of your submission has agreed to reveal their identity: Hongyan Wang (Reviewer #1).

The reviewers have discussed the reviews with one another and the Reviewing Editor has drafted this decision to help you prepare a revised submission.

The reviewers feel that while many comments were adequately addressed, several essential points remain problematic and do not support the conclusions of the manuscript. These include experiments with technical difficulties or uninterpretable results. If you are willing and able to address the reviewers' concerns you may resubmit a revised manuscript with a detailed rebuttal letter.

It is required that you address the following to be reconsidered for publication:

1) Additional NSC markers must be optimized. Dpn staining in some figures is unconvincing and must be verified using additional markers.

2) Cell markers routinely used by many labs must work (Wor, Ase, Elav)

3) Dpn is a reagent that works well and should be clear.

Please note that this decision letter does not guarantee that this manuscript will be accepted. At least one reviewer feels that their concerns were not addressed in the revised manuscript and that it is in everyone's best interest that the authors can prove that the inability to provide the appropriate expression patterns is not indicative of a deeper problem with the underlying hypothesis.

1) The main conclusion of the paper (glia transform into NSCs which produce neurons) but is not supported by data: only one NSC marker used out of many available.

The authors tried two additional NSC markers but did not observe staining, despite these reagents working for many labs in many publications. "We did not consider these results satisfactory enough to present."

This is a major flaw, especially how unusual the Dpn staining looks like in the ectopic Dpn+ cells (very speckly). Failure to show additional NSC markers very concerning is a real issue; also no evidence for asymmetric cell division at mitosis (a hallmark of these NSCs).

2) There is no evidence for proliferation of the ectopic Dpn+ cells. The authors state that ectopic Dpn+ cells expressed the S phase marker PCNA:GFP and can be labeled with the mitotic marker pH3.

However, only panes 8A-C show PCNA^+^ Dpn+ cells, which are increased following dilp-6 overexpression. No data in the figure shows ectopic Dpn+ cells that are pH3. The rest of the figure shows glial markers and PCNA or pH3, which is irrelevant to the question of whether ectopic Dpn+ cells can divide.

3) To show evidence that ectopic Dpn+ cells produce neuronal progeny, the authors used the pros-Gal4 line to drive flybow expression, and observed a small cluster of cells that included one Dpn+ and one Elav+ cell. As the authors say "this does not prove these cells are related by lineage, but is consistent with it."

This does not show Dpn+ cells are producing neurons.

4) The authors also used "flip out" genetics to permanently mark glial cells.

The genetics shown in the figure, legend, and reviewer response will not specifically label glia. The genotype is: actGAL4>y+>UASGFP/UAS-FLP; repoGAL4/Dilp-6. This would induce Flp widely, in all cells due to ubiquitous expression of actin-gal4. Most likely, the authors wrote down the wrong genotype in the figure, legend, Materials and methods, and reviewer response – it is probably actin promotor-FRT-stop-FRT-GFP. They cite Table 1 for more information on genotypes but there is no Table 1 provided.

5) In order to call kon and ia-2 partners, a direct physical interaction should be shown. The authors could not get the biochemical experiments to work for various reasons. Changed text from "partners" to "functional neuronal partner."

The continued use of “partner” is inappropriate. The most accurate description of their relationship is that they show “genetic interactions” – so the first results header should be changed from "Ia-2 is a functional partner of Kon" to "Ia-2 and kon show genetic interactions."

6) Saying ectopic Pros+ cells are GMCs or neurons is premature and can be definitively resolved by staining for Wor or Dpn (neuroblast-specific), Ase (neuroblast and GMC), and Elav (neurons). All have been extensively used by many labs. The authors could not get the stains to work.

This is unsatisfactory.

7) The text says loss of ia-2 "destabilizes cell fate" – which is a vague term that obscures the phenotype. The authors changed text to "… upregulated GMC and NSC markers."

They looked at Dpn but no other NSC marker, and Pros is not a specific GMC marker, also being expressed in neuropile glia near the midline (which is worrying).

8) Dpn staining in Figure 3D is unconvincing; everything looks speckly. The authors state that Dpn staining is speckly in their hands.

Many labs have used Dpn to mark neuroblasts, it is a very reliable reagent. The authors have good Dpn staining in other figures; this suggest to me that the ectopic Dpn+ cells are different from the normal Dpn+ NPCs, leading to different protein localization/levels. This concern is reinforced by the failure of the authors to show the ectopic Dpn+ cells express any other NSC marker.

9) Ectopic Dpn+ cells were not quantified due to due to the disruption and variability of the abdominal crush procedure. The authors only counted the VNCs in which they could see ectopic Dpn+ (cells).

Cherry-picking only VNCs that show ectopic Dpn+ cells is inappropriate.

10) In response to InR-Gal4 expression concerns, the authors state "we do not know whether (InR-gal4) represents the endogenous expression pattern". It labels sparse patterns of neurons and sporadic glial cells.

The authors directly state in the revised manuscript "we visualized InR expression using available GAL4 lines to drive his-YFP" but in the reviewer response they acknowledge this is not accurate.

---

## [Author Response]

[Editors’ note: the authors resubmitted a revised version of the paper for consideration. What follows is the authors’ response to the first round of review.]

Reviewer #1:Overall, this is an very interesting study with novel findings on regenerative neurogenesis. The experiments are well performed and supported by proper quantification. However, the following points may strengthen the manuscript.1) The authors show that both ia-2 gain of function and loss of function in neurons increased the number of abdominal VNC Dpn+ cells and ia-2 was up-regulated during injury. The fact that both the ia-2 lof and gof resulted in a similar phenotype seems counter-intuitive. It would be helpful if the authors can provide an explanation for this.

We do not fully understand why *ia-2* loss of function induced ectopic Dpn+, but data were unambiguous. Reasoning through it, *ia-2* loss of function would cause a decrease in Dilp-6 from neurons (Cai et al., 2001; Kim et al., 2008), but not from glia, as k_on_ mRNA levels were unaffected (Figure 1B) and *dilp-6* expression depends mostly on glial k_on_ (Figure 4C). *ia-2* loss of function caused an increase in Pros+ small cells that were not glia (Figure 2F-H), and this effect was non-autonomous, since neuronal number was unaffected (Figure 2D). This suggested that perhaps cell-cell interactions involving Ia-2, between neurons and other cell types, is required to prevent a reversion to progenitor or neural stem cell fate. Alternatively, as the genetic data suggest that neuronal *ia-2* and glial k_on_ mutually exclude each other, perhaps loss of *ia-2* function resulted in an increase in k_on_ function that we could not detect, and with it an increase in Dilp-6. The gain of function effect was clearer. Ia-2 is required for Dilp-6 secretion (Harashima et al., 2005; Cai et al., 2001), thus *ia-2* gain of function, as well as the increase in Ia-2 levels with injury, presumably resulted in increased release of Dilp-6. Conceivably, perhaps either way Dilp-6 increased and this induced Dpn.

2) Over-expression of full length k_on_ in neurons and glia resulted in the down-regulation of ia-2 mRNA levels but had no effect on dpn or dilp-6 expression. It would be interesting to know whether over-expression of konICD could cause a change in dpn or dilp-6 expression.

We agree with this reviewer that this was an interesting question. Thus, we carried out qRT-PCRs to measure levels of *dilp-6* and *dpn* mRNA in wandering 3^rd^ instar larvae of genotypes: control *UAS-kon^ICD^/+* and test *repoGAL4>UAS-kon^ICD^* and *elavGAL4>UAS-kon^ICD^* using n=10 VNC and 3 biological replicates each. The qRT results with the full-length form had been very variable for *dpn*, and had not shown an up-regulation. With *kon^ICD^*, *dpn* mRNA was upregulated, but this result was still very variable. Unfortunately, as the results were variable and we could not resolve the discrepancy between the full-length and the cleaved forms, we decided to leave them out.

3) Similarly, since InR signaling can activate k_on_ cleavage downstream in glia, could over-expression of konICD induce ectopic Dpn+ cells?

We also thought this was interesting. We attempted this, and carried out anti-Dpn stainings in wandering 3^rd^ instar larvae at 120h AEL, of genotypes: control *UAS-kon^ICD^/+* and test *repoGAL4>UAS-kon^ICD^*. Over-expression of *kon^ICD^* did not induce ectopic Dpn+ cells. However, as this effect was normally stochastic, anti-Dpn doesn’t reliably result in unambiguous signal and ectopic NBs generally have lower levels than normal NBs, we couldn’t resolve whether this was absence of evidence or evidence of absence. The qRT-PCR results above were variable, but resulted in increased dpn mRNA, which contradicted the lack of ectpic Dpn+ in the stainings. So together, these two data sets did not help resolve whether *UAS-kon^ICD^* could induce Dpn or not or how efficiently. Either way, we could not conclude and had to leave out.

We tried to address a similar question with a different approach: does lack of k_on_ in null mutant clones in glia result in the conversion of glial cells into neuroblasts or neurons? For this, we carried out Twin-spot MARCM mutant clones in glia with *repoGAL4*, whereby the twin clones resulting from mitotic recombination would be either *kon^c452^*mutant (RFP) or wild-type (GFP), and the control clones would be wild-type both for RFP and GFP. At first, the *kon^c452^* mutant phenotype looked interesting, as glial cells converted into neuronal cell fate. However, so did control wild-type clones. This meant that the Twin-spot MARCM lines provided by Janelia were rather strange. So we had to leave this out.

We have now edited the text to accommodate this unresolved point: *“…releasing Kon^ICD^ to pro*mote glial proliferation, although we were unable to verify whether it regulates gene expression”.

4) In general, the writing needs to be improved. Certain sentences were long and hard to follow which made the reading difficult.

The text has been modified extensively, and we very much hope that this reviewer finds it more accessible now.

Reviewer #2:This manuscript addresses an important problem – how to achieve neuronal regeneration in humans – using an excellent model system, *Drosophila,* where there are superb tools for identifying and characterizing signaling pathways.The manuscript jumps off from the observation that the evolutionarily conserved transmembrane/nuclear glial protein Kon-tiki (fly)/NG2 (mouse) may promote "regenerative neurogenesis" via an unknown ligand.Overall the paper is clearly written and the figures are appropriate.I have a single major comment that affects the whole paper, and multiple minor comments. Overall, the paper needs additional experiments before resubmission.1) Throughout the paper conclusions are not fully supported by the data shown. Most results paragraphs end with a conditional conclusion, but then in the headers and subsequent sections it is assumed to be proven. The manuscript would be strengthened by either dampening down conclusions to match the data, or by adding additional data to justify the conclusions. A major example is the claim of “regenerative neurogenesis” in the title, Abstract, and Results… yet there are no data presented showing the production of newly-induced neurons – only increased Dpn+ cells (without showing these cells express other stem cell markers, proliferation markers, or produce neuronal progeny).

Please allow us to break this paragraph down into two distinct points.

First point: I’m afraid there has been a slight misunderstanding when the reviewer stated “Most results paragraphs end with a conditional conclusion…”. In the previous version, we had ended each section by first drawing a conclusion based on the data, followed by introducing a new question – as it is most common in science that solving a question raises a further question. This new question was then solved in the following section. This style was meant to create a suspense, anticipation and ease the reading. We have done it for other manuscripts, where it worked, but we do understand that it doesn’t work for all readers. Thus, to satisfy this reviewer, for the revised version we have finished all sections with a straightforward conclusion, and we have moved the questions to the beginning of new sections.

Second point: claim of “regenerative neurogenesis” in the title, Abstract, and Results… yet there are no data presented showing the production of newly-induced neurons – only increased Dpn+ cells (without showing these cells express other stem cell markers, proliferation markers, or produce neuronal progeny). Other examples below.

This reviewer is correct that we had previously not provided any evidence for de novo neurogenesis. Thus, we have removed the word “neurogenesis” from the title – changing it to “neurogenic response from glia” – have considerably restricted the use of this word and taken care to use it more accurately.

An important point had been raised, i.e. whether the ectopic neural stem cells could complete de novo neurogenesis. For the revision, we have carried out more experiments – as suggested by this reviewer – and provide novel data to address this point:

1) Neural stem cell markers: We attempted to visualise the ectopic neural stem cells using other neural stem cell markers, anti-Worniu and anti-Asense. We did this in wandering third instar larvae at 120h AEL, of genotypes: control *repoGAL4/+,* and over-expression of *dilp-6* in glia: repoGAL4/UAS-dilp-6. Unfortunately, in our hands, these antibodies resulted in very high background, and we did not consider these results satisfactory enough to present.

However, we provide new data that show that not only lateral neuropile glia, but also midline can convert into Dpn+ neuroblasts. We demonstrate that all ectopic Dpn+ cells originate from glia. These new data are now reported in Figure 7C-F showing Wrp+ Dpn+ cells at the midline, and revised version text.

2) Proliferation: we tested whether over-expression of *dilp-6* or *ia-2* could cause the ectopic neural stem cells to divide, testing this using the G1-phase marker PCNA-GFP and the mitosis marker anti-phospho-histone-H3 (pH3). We found that:

a) New Figure 8A-D, PCNA-GFP: Over-expression of *dilp-6* at 110.5h AEL with *heat-shock GAL4* in late third instar larvae induced PCNAGFP in ectopic Dpn+ cells located around the neuropile. Genotype: *PCNAGFP, hsGAL4/+;; UAS-dilp-6/+.*

b) New Fig8ure A-H, PCNA-GFP: Over-expression of *dilp-6* with at 110.5h AEL *heat-shock GAL4* in late third instar larvae induced PCNAGFP in Wrp+ midline glia.

Genotype: *PCNAGFP, hsGAL4/+;; UAS-dilp-6/+*

c) New Figure 8I-L, pH3: Over-expression of *ia-2* with *repo-GAL4* induced mitosis, visualised in Wrp+ pH3+midline cells. Genotype: *repoGAL4>UAShisYFP, UAS-ia-2.* The increase was not statistically significant, but visualising mitosis is challenging as it is very brief.

Thus, we demonstrate that ectopic Dpn+ cells can be induced from two types of glial cells, Repo+ neuropile glia and Wrp+ midline glia, and these can divide. These findings are now reported in revised version text.

3) Neurogenesis: we carried out two new experiments to test whether the ectopic Dpn+ cells can generate neurons:

a) Use *pros* promoter to visualise progeny neurons, data provided in new Figure 9A,B: we had shown that ectopic Dpn+ cells originated from glial cells (Figure 6,7). Thus, we would not be able to use a glial promoter to visualise their progeny cells, since as glia reprogrammed into neural stem cells, they would stop responding to glial gene expression (e.g. driven by Repo, controlled by the *repo* promoter). Thus, to visualise the progeny cells of re-programmed glia, now Dpn+, we used the *pros* promoter instead. As the *pros* gene can be expressed in any nervous system cell type, we reasoned that it could remain switched on as cells went through a cell-state transition. With *prosGAL4> UASFlyBow* we small clusters of cells, like shown in Figure 9A,B: one cell was Dpn+ and one was Elav+. Although this does not demonstrate that these cells are related by lineage, it would be compatible with it.

b) Lineage tracing, new Figure 9C,D: To test whether Dpn+ cells could result in Elav+ cells through lineage, we used flies bearing *actinGAL4>y+ stop>UASGFP, repoGAL4* crossed to *UASFLP, UAS dilp-6*. Here, Flippase is over-expressed only in glia, where it removes the stop codon downstream of the actin promoter, thus switching reporter gene expression from being under the control of a glial promoter to a house-keeping general promoter, of *actin*. This labels all progeny cells of glia and reprogrammed glia. We found that over-expressing *dilp-6* resulted in progeny cells that were GFP+ as well as Elav+, and therefore were neurons.

Thus, these data demonstrate that glia reprogrammed into ectopic Dpn+ neural stem cells by Dilp-6 can produce neurons. Production of neurons was rather limited as we never found clusters larger than 3 cells. As these larvae were 120h old or as old as possible and about to pupate, pupariation likely restricts the regenerative neurogenic response of glia.

These new data are provided in new Figure 9A-D and text.

2) Calling k_on_ and ia-2 "partners" suggests a direct physical interaction. This has not been shown. There could be a dozen genes acting between these two. Please add direct interaction experiments or use wording that accurately reflects the unknown relationship between the two proteins. Multiple examples in the text; where it is stated "ia-2 is a functional partner of Kon."

We were also interested in finding out whether the interaction was direct, as both are transmembrane proteins, and could function in cell-to-cell communication. We attempted to carry out co-immunoprecipitations, by tagging Ia-2 with HA and Kon-full-length with FLAG and co-transfecting S2 cells. This involved cloning *ia-2* and k_on_ cDNAs into *pAct5* expression vectors, for which we had to generate a k_on_ cDNA by RT-PCR, as full-length cDNAs were unavailable. However, we could not get *pAct5-kon-FL-FLAG* to express. We found out there was a point mutation in the construct that introduced a stop codon. Using site directed mutagenesis, we corrected this mutation, but still *kon-FL-FLAG* would not get expressed in S2 cells. We did not understand why this was the case, but solving this would have meant starting again from scratch. This would have required far longer time than appropriate for a revision. Thus, we put considerable effort to attempt this experiment, but could not get it to work.

We modified the text instead, and used the term “functional neuronal partner”, meaning a functional link without necessarily a physical interaction. The the text has been modified accordingly in the Abstract, Introduction and Results sections.

3) Figure 2F,G says "ectopic Pros+ cells might be GMCs or neurons" but this result should be resolved by staining for Elav protein (neurons) and Asense (neuroblasts and GMCs) and Worniu (neuroblasts).

We attempted to resolve this using Anti-Elav (rat) + anti-Pros (mouse) to stain larvae of genotypes: Control: *yw,* and test: *elavGAL4>UAS-ia-2RNAi.* However, unfortunately, we obtained cross-hybridisation, and the two markers could not be resolved.

We also tried with the recommended neuroblast markers, Anti-Asense and anti-Worniu together with anti-Pros, in larvae of the same genotype as above. However, unfortunately, in our hands these antibodies gave very high background, and the data were not satisfactory. Thus, these experiments did not work either.

4) The text says loss of Ia-2 "destabilizes cell fate" but this statement is extremely vague; it seems to hide the fact that the actual phenotype is not known.

This is a point for debate. The fact that a cell previously known as glial can become a neural stem cell would imply that cell fate is not as fixed and stable as it was previously thought. The phenotype was being explored in Figure 2, at the beginning of the paper, so perhaps at that point the phenotype was not yet known. But 8 Figures later for an article of 10 main Figures and a proposed functional model, it would appear that the statement above is a misinterpretation. Still, to address this reviewer’s point, we have now changed this statement for “interference with normal neuronal Ia-2 levels up-regulated ganglion mother cell and neural stem cell markers”.

5) The use of ectopic Dpn+ cells to indicate “induced neurogenesis” is premature. The authors at a minimum need to assay the ectopic Dpn+ cells for other neuroblast (stem cell) markers (e.g. Wor, Ase) to validate stem cell identity, and PH3 or PCNA to validate their proliferative state. To make their conclusion solid it would be ideal to induce clones containing these ectopic Dpn+ cells and show the clone includes neurons.

This reviewer is correct that we had previously not provided any evidence for de novo neurogenesis. Thus, we have removed the word “neurogenesis” from the title – changing it to “neurogenic response from glia” -, have considerably restricted the use of this word and taken care to use it more accurately.

Still, an important point had been raised, i.e. whether the ectopic neural stem cells could complete de novo neurogenesis. For the revision, we have carried out more experiments – as suggested by this reviewer – and provide novel data to address this point:

1) Neural stem cell markers: We attempted to visualise the ectopic neural stem cells using other neural stem cell markers, anti-Worniu and anti-Asense. We did this in wandering third instar larvae at 120h AEL, of genotypes: control *repoGAL4/+,* and over-expression of *dilp-6* in glia: repoGAL4/UAS-dilp-6. Unfortunately, in our hands, these antibodies resulted in very high background, and we did not consider these results satisfactory enough to present.

However, we provide new data that show that not only lateral neuropile glia, but also midline can convert into Dpn+ neuroblasts. We demonstrate that all ectopic Dpn+ cells originate from glia. These new data are now reported in Figure 7C-F showing Wrp+ Dpn+ cells at the midline, and revised text.

2) Proliferation: we tested whether over-expression of *dilp-6* or *ia-2* could cause the ectopic neural stem cells to divide, testing this using the G1-phase marker PCNA-GFP and the mitosis marker anti-phospho-histone-H3 (pH3). We found that:

a) New Figure 8A-D, PCNA-GFP: Over-expression of *dilp-6* with *heat-shock GAL4* in late third instar larvae induced PCNAGFP in ectopic Dpn+ cells located around the neuropile. Genotype: *PCNAGFP, hsGAL4/+;; UAS-dilp-6/+*

b) New Figure 8 A-H, PCNA-GFP: Over-expression of *dilp-6* with *heat-shock GAL4* in late third instar larvae induced PCNAGFP in Wrp+ midline glia.

Genotype: *PCNAGFP, hsGAL4/+;; UAS-dilp-6/+*

c) New Figure 8I-L, pH3: Over-expression of *ia-2* with *repo-GAL4* induced mitosis, visualised in Wrp+ pH3+midline cells. Genotype: *repoGAL4>UAShisYFP, UAS-ia-2.* The increase was not statistically significant, but visualising mitosis is challenging as it is very brief.

Thus, we demonstrate that ectopic Dpn+ cells can be induced from two types of glial cells, Repo+ neuropile glia and Wrp+ midline glia, and these can divide. These findings are now reported in revised version text.

3) Neurogenesis: we carried out two new experiments to test whether the ectopic Dpn+ cells can generate neurons:

a) Use *pros* promoter to visualise progeny neurons, data provided in new Figure 9A,B: we had shown that ectopic Dpn+ cells originated from glial cells (Figure 6,7). Thus, we would not be able to use a glial promoter to visualise their progeny cells, since as glia reprogrammed into neural stem cells, they would stop responding to glial gene expression (e.g. driven by Repo, controlled by the *repo* promoter). Thus, to visualise the progeny cells of re-programmed glia, now Dpn+, we used the *pros* promoter instead. As the *pros* gene can be expressed in any nervous system cell type, we reasoned that it could remain switched on as cells went through a cell-state transition. With *prosGAL4> UASFlyBow* we small clusters of cells, like shown in Figure 9A,B: one cell was Dpn+ and one was Elav+. Although this does not demonstrate that these cells are related by lineage, it would be compatible with it.

b) Lineage tracing, new Figure 9C,D: To test whether Dpn+ cells could result in Elav+ cells through lineage, we used flies bearing *actinGAL4>y+ stop>UASGFP, repoGAL4* crossed to *UASFLP, UAS dilp-6*. Here, Flippase is over-expressed only in glia, where it removes the stop codon downstream of the actin promoter, thus switching reporter gene expression from being under the control of a glial promoter to a house-keeping general promoter, of *actin*. This labels all progeny cells of glia and reprogrammed glia. We found that over-expressing *dilp-6* resulted in progeny cells that were GFP+ as well as Elav+, and therefore were neurons.

Thus, these data demonstrate that glia reprogrammed into ectopic Dpn+ neural stem cells by Dilp-6 can produce neurons. Production of neurons was rather limited as we never found clusters larger than 3 cells. As these larvae were 120h old or as old as possible and about to pupate, pupariation may restrict the regenerative neurogenic response of glia.

These new data are provided in new Figure 9A-D and text.

6) Figure 3D is unconvincing. The images show large speckles in the Dpn channel that are similar to the “ectopic Dpn+ cells” plus the number of ectopic Dpn+ cells per brain is not quantified (that I could find).

We agree that our Dpn stainings were rather speckly at this stage, later ones were cleaner. Injured larval VNCs, most particularly these younger and slender ones, were extremely fragile, and stainings were rather challenging. If washed vigorously the whole injured VNC could fall apart. We tried some more samples, but unfortunately the quality of anti-Dpn at this early stage was still similar to that provided.

To address the criticism of lack of quantification, we counted all ectopic Dpn+, not just for this time point, but for all time-points. For the younger larvae (Figure 3H, 74-80h AEL), when there were still developmental neuroblasts, the crush injury caused massive lesions that removed also normal Dpn+ neuroblasts, complicating the analysis. So, we counted the VNCs in which we could see ectopic Dpn+. For the rest (Figure 3K-O) all injured samples were counted, and all Dpn+ in the abdominal VNC were counted at 120-129h AEL, when there are no remaining developmental neuroblasts except in the midline. In (Figure 3P) we show the average number of Dpn+ surrounding the lesion, for each time point.

Counts of Dpn+ cells are now provided in new graphs in Figure 3H,K,L,O,P

7) Either show data or cite prior work showing that InR-gal4 mimics the InR endogenous expression pattern.

We had visualised the expression of the *insulin receptor (InR)* using an available GAL4 line, *InR[NP2552].* We do not know whether this represents the endogenous expression pattern, as *GAL4* is inserted in an intron. To test whether other *InR-GAL4* drivers might have the same or different expression patterns, we purchased and tested all three available lines. We visualised their expression with nuclear histone-YFP and stained the larvae with the pan-glial marker anti-Repo and the pan-neuronal marker anti-Elav:

Anti-Repo and anti-Elav in InRGal4[NP2552], InRGal4[0488-G4], InRGal4[0726-G4] x UAS-HisYFP. All of these lines are insertions into introns, and they all gave rather sparsely distributed profiles, mostly in neurons, but including neuropile and surface glia sporadically. We felt it was un-necessary to include all these data as they did not add any new information.

As a result of these new data, we still used *InR[NP2552]*, but revised Figure 4E,F.

We have now cited previous references that report the expression of the *InR* in neurons and its function in axon guidance (Fernandez et al. 1995 EMBO J; Song et al., 2003 Science). Impactful publications have involved InR function in neuroblasts and neurons, without reporting its expression pattern (Chell and Brand, 2010; Sousa-Nunes et al., 2011; Fernandes et al., 2017). This suggests that InR could be expressed in many if not all cell types in the CNS.

Citations have been included in the text and reference list.

8) Figure 5B. The increased number of glia is not convincing. All panels look the same to me. The statistics show much greater significance between panels than I trust. It would be good to give the raw numbers in a supplement so that the statistics could be validated.

We wonder if perhaps there might have been a misunderstanding here. To clarify, glia were labelled with *repo>his-YFP* and his-YFP+ cells were counted automatically in 3D volumes throughout the stacks using accurate and validated DeadEasy software (Forero et al., 2012). DeadEasy counts cells using a combination of signal intensity and cell volume within validated margins. Graphs do not show mean nor error bars, but box-plots, with the box representing the distribution of 50% of samples and the line across is the median, and the whiskers the distribution of the upper and lower 25% of samples. The significance provided is factual, and all statistical details had been provided in the Supplement file 1. Thus, the reviewers “lack of trust” is not grounded on the evidence provided. Although in principle we would of course be happy to provide the raw data for these counts, we find this unnecessary. This kind of experiment is not novel, nor the software, which has been used to automatically count larval his-YFP+ cells in previous successful publications (Kato et al., 2011; Losada-Perez et al., 2016; Foldi et al., 2917). To avoid misunderstandings and make the difference more apparent, the graph has been enlarged and a dashed line has been added across the control median, to facilitate comparisons.

Please see Revised Figure 5C, and statistical details provided in Supplementary file 1.

Reviewer #3:The manuscript by Harrison et al. describes a neuronal-glial communication loop that requires Ia-2, Dilp-6, InR and k_on_ and in some circumstances, can induce the formation of Dpn+ cells from glia. Overall, the manuscript presents an interesting signaling relationship between neurons and glia that could influence neurogenesis. However, the manuscript was not easy to read or interpret and covers many details not essential for the final conclusions. Significant revisions are required to clarify and simplify the manuscript before publication. In addition, ensure that conclusions are appropriate for the data shown and do not overreach. Finally, specific suggestions are below.1) The authors state that k_on_ overexpression (which has a striking VNC overgrowth phenotype) decreases the transcript of ia-2 (not significantly?) but RNAi of ia-2 suppresses k_on_ overexpression phenotypes, which is suspicious. Do multiple RNAi lines for ia-2 give the same suppression of k_on_ overexpression? How about the other ia-2 phenotypes presented throughout the manuscript? It is possible that off target affects could account for these phenotypes and a second RNAi line should be used to verify ia-2 phenotypes. Since the authors start with k_on_ overexpression phenotypes in the VNC and suppression, the reader is left with a conflict that is not resolved. How does ia-2 knockdown suppress k_on_ overexpression phenotypes, which reduces the levels of ia-2 in the first place? If the authors choose the keep this data in the manuscript, they must address this discrepancy.

Please allow us to break this paragraph into its distinct points:

1) The reduction in *ia-2* mRNA levels caused by k_on_ over-expression in neurons is statistically significant. We had forgotten to add the star before to the graph, and this has now been corrected. Shown in revised Figure 1A. See also Supplementary file 1 for all statistical details.

2) “… k_on_ over-expression… ia-2 RNAi suppresses…” Counter-intuitively, these conditions are not incompatible. Conceivably, if *ia-2* and k_on_ were over-expressed together, the VNC could become even longer than with k_on_ gain of function alone, which would mean that a relative decrease in *ia-2* levels would weaken or partially rescue the phenotype. Accordingly, a further reduction with RNAi would rescue even further. We did not carry out this test. Nevertheless, the long VNC phenotype is not specific enough, as it has been reported by others to result from a variety of genotypes – hence the reason for having these as supplementary data. Still, it was a useful approach, as the known candidates did validate it, and for the unknown, it led to an unexpected link to the insulin pathway. We have now extensively modified this section to clarify the limitations of this screen, and resolve the conflict.

3) Do multiple RNAi lines for ia-2 give the same suppression of k_on_ overexpression?

We attempted to repeat the screen with another RNAi line from VDRC: UAS-ia-2RNAi[*KK108555-VIE-260B]* and crossing it to *elavGAL4 UASkon, UAS Flybow*, but this line caused a weaker knock-down, so the effect was lost.

Using qRT-PCR, qRT-PCR ia-2mRNA in larvae in which *ia-2* was knocked-down in neurons with this new RNAi line *elavGAL4>UAS-ia-2RNAi[KK108555-VIE-260B]* and control: elavGAL4/+ (n=10 VNC with 3 x biological replicates), we found that this new line caused a much weaker knock-down than the previous *UAS-ia-2*[*TRIPHMS00536]* line previously used. Whereas *UAS-ia-2*[*TRIPHMS00536]* caused an 80% reduction in *ia-2* mRNA levels, the new VDRC line *UAS-ia-2RNAi[KK108555-VIE-260B]* only caused a 25% reduction. This demonstrates that the original TRIP line was more effective.

Importantly, this demonstrated that both *ia-2* RNAi lines knock-down *ia-2* expression.

We now provide these new data in Figure 1E and revised text.

4) How about the other ia-2 phenotypes presented throughout the manuscript?

We only analysed *ia-2* loss of function RNAi knock-down phenotypes in Figure 1B and Figure 2D-J. That is, qRT-PCRs for k_on_ mRNA and stainings with Eve, Pros and Dpn. (Stainings with FasII and BP102 shown in Figure 2—figure supplement 1 had revealed no phenotypic differences from controls). There are no other *ia-2* RNAi data in the paper that contains 9 main figures of experiments and 5 supplementary figures. Our qRT-PCR analysis (described above) showed that the new RNAi line *UAS-ia-2RNAi*[*KK108555-VIE-260B]* was considerably weaker than the line we had originally used, as it only knocked them down by 25%. Thus, we did not anticipate remarkable phenotypes from this weak knock-down. Still, to address the reviewer’s request we repeated the most meaningful staining, which was anti-Dpn in larvae of genotype *elavGAL4> UAS-ia-2RNAi[KK108555-VIE-260B]* and control: *elavGAL4/+.* As expected, we obtained a mild increase in ectopic Dpn+ cells with this line.

These new data are provided in the revised graph Figure 2J and changes the text.

Finally, to rule out off-target effects of *ia-2* RNAi, we had previously combined it with heterozygosity for a deficiency for the locus. We now expand on this genotype and provide data for *Df ia-2 elavGAL4>UASia-2RNAi[TRIPHMS00536]* phenotypes in Figure 2D,E, I,J showing that the deficiency enhances the strength but does not alter the quality of the phenotypes.

To conclude, data show that the *ia-2* loss of function phenotypes most likely are not due to off-targets of *ia-2 RNAi.* This supported by qRT-PCR data, RNAi knock-down in a heterozygous background with a deficiency for the *ia-2* locus and using two distinct RNAi lines that partially reproduce the Dpn+ phenotype.

2) For Figure 1A and supplemental, with what genotype were the samples normalized/compared? The authors show the control in B but not A. Conclusions are made that are not necessarily supported. Since the authors state that k_on_ overexpression only marginally decreased ia-2 mRNA levels, too much emphasis is placed on this result, which leaves the reader with concerns mentioned in 4, above. Knockdown of k_on_ may not specifically regulate Ia-2 mRNA levels but may be the result of extra Ia-2 positive cells shown in Figure 2.

We agree with this reviewer that these data could have been presented better. Figure 1A is a subset of the data also presented within Figure 1—figure supplement 3. Normalisations were to house-keeping Rp132. Comparisons were to k_on_ genotypes on k_on_ mRNA levels, ie the controls for Figure 1A and supplementary are the levels of k_on_ mRNA itself. That is, how far were k_on_ mRNA levels increased with the gain of function, and reduced upon RNAi knock-down.

We have now revised these graphs, split the data into two separate groups and added the controls. On the left graph, controls show that over-expression of k_on_ resulted in a very high increase in k_on_ levels beyond the scale of the remaining data (hence their original exclusion and from Figure 1—figure supplement 3B). On the right graph, k_on_ RNAi knock-down resulted in a 50% reduction in k_on_ mRNA levels. These new graphs are shown in revised Figure 1A

The effect of k_on_ over-expression on *ia-2* mRNA is not marginal, but instead statistically significant. These data had been provided before in the Supplementary file 1 with the statistical analysis, but unfortunately we forgot to add the star p<0.05 on the figure. This has now been added to Figure 1A.

We agree with this reviewer that at least in part, the increase in *ia-2* mRNA caused by k_on_ loss of function could be due to the increase in Ia-2YFP+ cells observed along the midline in these larvae. Still, it was interesting that other genes known to be expressed in neurons (like the phosphatases) were not up-regulated with k_on_ loss of function, only *ia-2* was, suggesting a more direct functional interaction. This has now been clarified in the text.

3) Pros does not simply activate ia-2 as removal of pros or Pros expression causes ia-2 mRNA increase. This is difficult to interpret and is not a simple ARROW (below 1D). Perhaps analysis of Notch and Pros is unnecessary here as it complicates the message of the manuscript.

We regret that we did not make this point clear enough before, please allow us to explain. We had previously published that Notch and Pros depend on each other in glia, and that loss of *pros* function down-regulates Notch (Griffiths and Hidalgo, 2004; Kato et al., 2011; Losada-Perez et al., 2016). In fact, in *Notch* mutants, *pros* expression goes down, and in *pros* mutants, Notch signalling is down-regulated; conversely, over-expression of *pros* up-regulates Notch signalling, and over-expression of *Notch^ICD^* activates *pros* expression (Griffiths and Hidalgo, 2004; Kato et al., 2011; Losada-Perez et al., 2016). Thus, we considered it un-necessary to dwell on this again. It also makes sense that *Notch* loss of function caused an upregulation of *ia-2* mRNA, because k_on_ depends on *Notch* (Losada-Perez et al., 2016) and loss of k_on_ function also upregulates *ia-2* mRNA (Figure 1A). Therefore, if *pros* loss of function also up-regulates *ia-2* mRNA (Figure 1D, left), this could be an indirect effect, because loss of *pros* function will result in a decrease in *Notch* and therefore *kon*. Pros is a transcription factor, so it is more probable that the effect of the over-expression, rather than the loss of function, is a direct phenotype in this case (Figure 1D, right). Furthermore, the *pros* gain of function phenotype is statistically significant, whereas the loss of function is not. This means that the gain of function is most meaningful. In genetics language, this is expressed as *pros* positively regulates *ia-2*. The arrow does not mean that this explains everything, it’s standard genetic inference and symbolism. In our context, it provided a step to move forward.

A comparable effect was observed in the relationship between k_on_ and *pros:* both loss and gain of *pros* function down-regulated k_on_ expression, and similarly, the effect of *pros* gain of function was more pronounced (Losada-Perez et al., 2016). This meant that *pros* negatively regulates *kon*. Since now we show that *ia-2* and k_on_ antagonise each other, the fact that *pros* relates to k_on_ and *ia-2* in opposite ways, further strengthens the genetic inferences drawn, and the conclusion that *ia-2* and *k_on_* are mutually exclusive in neurons and glia, respectively.

Placing *ia-2* within the context of the *pros-Notch-kon* regenerative gene network was a very important initial step in the analysis of *ia-2* function, and therefore cannot be removed. We had previously published that Notch and Pros, together with Kon/NG2, drive a glial regenerative response to CNS injury, and we demonstrated that this response is evolutionarily conserved from flies to mammals, (Griffiths and Hidalgo, 2004; Kato et al., 2011; Kato et al., 2015; Losada-Perez et al., 2016). If Ia-2 were to be involved in regenerative responses to CNS injury, it would make sense for it to be genetically and functionally related to this gene network. Indeed, this is what the data in Figure 1C,D showed. Linking *ia-2* to *kon, pros* and *Notch* was an initial exploratory phase that enabled us to pursue the investigation of Ia-2 function in a regenerative response to injury.

We very much hope that this is clearer now, and that the reviewer is satisfied with this explanation. Accordingly, we have now revised the text.

4) When analyzing loss of ia-2, sometimes Df plus RNAi is used, sometimes only RNAi is used. Please be consistent with the strategy to remove ia-2 if you wish to compare phenotypes.

We agree with this reviewer that it would be neater to have all consistent throughout, and we have made an effort to tidy this up as much as possible, within a reasonbale time frame for a revision. Thus, we have now added new genotypes to Figure 2E, J: n=10-15 Anti-Dpn in *elavGAL4>UAS-ia-2RNAi* and Anti-Eve in *elavGAL4 x Df, UAS-ia-2RNAi*

We provide a revised Figure 2D,E,I,J.

5) What is the difference between Figure 3B and D? You provide numbers for controls in the text, so it is unclear why numbers are not provided for the injury model. Also, what is the percentage that lack Ia-2YFP?

Apologies if this was previously unclear. B are horizontal and D transverse sections of the same samples. We have now revised all our figure legends and this information has been provided. We have now quantified abdominal Dpn+ cells upon injury at each of the time-points, and all counts are provided in new graphs in Figure 3H,K,L,O. In samples with ectopic Dpn+ cells, we have also counted the number of ectopic Dpn+ specifically around the lesion, at each time point, and we provide these data in new graph Figure 3P.

We have also revised the statement to “and most, if not all, ectopic Dpn+ cells lacked Ia-2YFP”. We think it would be inaccurate to try to put a number on this. It appears that they all lack ia-2YFP, but injury samples look rather different from un-injured samples, cell morphology is different, the CNS is damaged and stainings can be messier. Thus, providing a number to this could be a misrepresentation.

In response to point 5, new graphs are provided in Figure 3H,K,L,O,P.

6) The authors make strong statements about RNA changes in ia-1 LOF and GOF models (Figure 4E-F), but no changes are statistically significant. Perhaps this data belongs in supplemental or not include at all.

We agree that data that are not statistically significant do not mean very much, therefore, we have entirely removed all of these data from the paper. We have only retained the relationship to *dilp-6,* which is the most significant and meaningful. Please find a revised Figure 4C,D.

7) Is Figure 5E the same stage as 2J? If so, the controls have very different number of Dpn + cells. If this is due to the genetic background or the GAL4, it is not the correct driver to use.

No, they are not, and we thank the reviewer for noticing this important detail that we had not explained. These experiments differed. Figure 5E was of L3 larvae at 72h AEL to see developmental neuroblasts, whereas Figure 2J was at 120h AEL to exclude developmental neuroblasts, hence the reduction in Dpn+ cells.

The controls were inscutable-GAL4/+ for Figure 5E and elavGAL4/+ for Figure 2J. As explained above, the fact that the number of abdominal Dpn+ cells differed between the two is not due to genetic background nor the GAL4, but instead it is due to the stage, i.e. age of the larvae.

Clarifying this point is very important as these figures address two different questions:

1) Figure 2I,J addressed the question of whether altering *ia-2* levels affected cell fate, whether this could be detected with anti-Dpn, whilst distinguishing any effect from Dpn+ normal developmental neuroblasts. To this aim, images were from larvae fixed at 120h After Egg Laying, when no developmental neuroblasts remain in the larval VNC, except for a few at the midline, as they have all been eliminated through apoptosis. This figure shows that altering *ia-2* levels induced ectopic Dpn+ cells surrounding the neuropile or along the midline, in the abdominal VNC, at 120h AEL.

2) Figure 5D,E addressed the question of whether k_on_ was required for normal developmental neuroblast cell fate. Images were from larvae fixed at 72h AEL, as here we wanted to see the normal, developmental Dpn+ neuroblasts of the larval VNC. All Dpn+ cells here are normal developmental neuroblasts, in the abdominal VNC, at 72h AEL.

We have now added this clarification in the main text, figure legends, Materials and methods section, and we have also indicated it on a revised Figure 5D,E.

8) In the section "Ia-2 and Dilp6 can induce neural stem cells from glia", it does not seem that everything is included or referenced correctly. I don't see images of "repoGAL4>UAS-ia-2, UAS-dilp-6-RNAi", "repoGAL4>UAS-ia-2, UAS-konRNAi", or "repoGAL4>UAS-dilp-6, UAS-InRDN" even though 6A-D is referenced for each genotype. Ideally, the authors should control for GAL4 dilution effects and use the same ia-2-YFP in all comparisons (epoGAL4>UAS-dilp-6, UAS-InRDN does not have this transgene).

Apologies for this, we thought perhaps it was not necessary to show every single image, which would make the figure massive. We now provide the images for every genotype represented in the graph, in a revised Figure 6A,B.

We have also added new anti-Dpn in *ia2YFP, repoGAL4>UAS-dilp-6, UASInR^DN^,* so that all samples now have *ia-2YFP*. These new data have been added to revised Figure 6A,B,D.

9) Dpn staining in Figure 7 is not clear. All arrows do not convincingly point to a Dpn + cell because of the high background. Use a different color for Dpn in C/D as blue cannot be seen. Better Dpn staining is presented in other figures. As this experiment is important, perhaps an image with less background or better Dpn staining can be shown.

We tried to improve these images, but these experiments were rather difficult as over-expression of GTRACE in all glia severely reduced viability. Thus, fewer progeny of the required genotype resulted from the crosses, and we could only use the oldest larvae at 120h AEL. Furthermore, anti-Dpn is not a great antibody, and stainings not always produced high signal, so although we put a considerable effort in repeating this experiment, it did not always work well enough. Thus, unfortunately, we could not provide new images for Figure 7A,B or G,H.

We realise blue is not easy to see over black, but that is why we presented the single channels in black and white instead. Still, to satisfy this reviewer, colours have been changed for GTRACE data: red to magenta, and blue to cyan. Please see revised Figure 7G, H.

We provide new additional data for Figure 7. We had previously noticed that over-expression of ia-2 and dilp-6 induced two types of ectopic Dpn+, one type around the neuropile and a second class along the midline. We had identified the lateral cells as Repo+ glia. However, we had not identified the midline cells. For the revision, we carried out new experiments to identify the midline cells, staining:

anti-Dpn and anti-Wrp in the *repoGAL4>UASdilp-6* and *repoGAL4>UAShis-YFP UASdilp-6*: new Figure 7 C-Fanti-Wrp in *PCNA-GFP, hsGAL4>UASdilp-6* in new Figure 8F-Hanti-pH3 in *repoGAL4>UAShis-YFP, UASia-2* and *repoGAL4>UAShis-YFP, UASia-2.*

These new data identified the ectopic Dpn+ Wrp+, originally midline glia. This is important, as we now demonstrate that ALL ectopic Dpn+ induced by Dilp-6 and Ia-2 – both around the neuropile and along the midline – originate from glia.

We provide these additional new data in Figure 7C-F and Figure 8E-L and revised text.

10) What is the direct evidence that there is Ia-2 regulated Dilp-6 signaling from neurons to glia?

This had been provided in the original submitted version. It appears the reviewer missed some of these data, which we recapitulate here:

Ia-2 is highly evolutionarily conserved and it functions in dense core vesicles to release insulin and neurotransmitters (Harashima et al., 2005; Kim et al., 2008; Nishimura et al., 2010; Cai et al., 2011). In *Drosophila*, it is required for the secretion of only Dilp-6 (Kim et al., 2008).We provided evidence that *ia-2* is expressed in all neurons: Figure 1E,F,G,H.We provided evidence that *dilp-6* is expressed in some neurons but mostly in glia: Figure 4A,B.We provided evidence that *dilp-6* expression depends on *kon*: Figure 4C, and we had previously published that k_on_ is expressed and functions in glia, and is required for glial cell proliferation and differentiation (Losada-Perez et al., 2016 J Cell Biol).We provided evidence that *InR* is expressed in neurons and at least some glia, including cortex glia and neuropile glia: Figure 4E,F. InR had been published by others to function in relay signalling between distinct cell populations to re-activate neuroblasts from quiescence and induce neuronal differentiation (Chell and Brand, 2010; Sousa-Nunes et al., 2011; Fernandes et al., 2017; Gil-Ranedo et al., 2019 ). May I point out that, however, these previous high profile publications did not report the expression pattern of *InR.* The expression of *InR* in the CNS was published using low resolution in situ hybridisations and antibody stainings in embryos by Fernandez et al. 1995 EMBO J. Its requirement in neurons for axon guidance was reported by Song et al., 2003. Altogether, InR is known to function in neuroblasts and neurons, and we show that it is expressed in glia too.Using genetic functional analysis, we demonstrated that InR necessarily functions in glia. InR is the only known receptor for Dilp-6, thus if Dilp-6 can induce a phenotype it means InR must be expressed and functional in that receiving cell type. Over-expression of *dilp-6* caused multiple phenotypes in glia, including:over-expression of *dilp-6* in glia increased glial cell number: Figure 5B,C;over-expression of *dilp-6* in glia induced ectopic Dpn+ cells: Figure 6A-D;over-expression of *dilp-6* in glia induced ectopic Dpn+ Repo+ cells: Figure 7A,B;over-expression of *dilp-6* induced GFP+ clones of glial origin: Figure 9C,D.Using genetic epistasis analysis, we demonstrated that InR functions in glia: over-expression of a dominant negative for of the *InR (InR^DN^)* in glia rescued the increase in cell number caused by the over-expression of *dilp-6* (*repoGAL4>UAS dilp-6, UASInR^DN^*): Figure 5B,C. And, over-expression of *InR^DN^* in glia rescued the induction of ectopic Dpn+ caused by over-expression of *dilp-6* (*ia-2YFP, repoGAL4>dilp6, InR^DN^*): Figure 6A,BD.Finally, we demonstrated that Ia-2 could induce Dpn+ cells from glia, and this depended on Dilp6, k_on_ and InR functioning in glia downstream: Figure 6A-D and Figure 7G,H.

Altogether, the above data showed that Ia-2 functions in neurons to release Dilp-6; Dilp-6 is produced by some neurons, received and amplified via InR in glia; and both Ia-2 and Dilp-6 can induce ectopic Dpn+ cells from glia. Therefore, necessarily signalling can be relayed from neurons to glia.

To ensure that these important connections between data are not missed by future readers, we have expanded on the explanation in the text in multiple points. We hope we have made it clerer now.

We have also revised the concluding Figure 10.

[Editors’ note: what follows is the authors’ response to the second round of review.]

It is required that you address the following to be reconsidered for publication:1) Additional NSC markers must be optimized. Dpn staining in some figures is unconvincing and must be verified using additional markers.

Unfortunately, there seems to have been a misunderstanding. Our samples from the first submission were appropriately stained with anti-Dpn. In the letter to reviewers, we had mentioned experiments we carried out that did not work, but this did not apply to the submitted evidence. Dpn is normally abundant in the thorax, where we detected it well, but we had not shown the thoracic signal because our work does not focus on the thorax. Our manuscript focused on the abdominal segments of the late third instar larval VNC, where there are normally no Dpn+ cells there except for the midline.

We now provide the Dpn thoracic signal for our original samples, in revised Figure 2I, Figure 3F,L, Figure 6A, Figure 6—figure supplement 1, as well as a new Figure 7A-F. Anti-Dpn in the thorax and optic lobes worked well, and differences can be attributed to genotypes (e.g. over-expression of a dominant negative form of the insulin receptor). Thus, we did not have technical problems with anti-Dpn. Relatively to the thoracic signal, the ectopic abdominal Dpn signal in glia is lower.

Similarly, anti-Worniu worked well in the thorax, but did not reveal ectopic signal in glia. We have now tested a new aliquot of anti-Ase, a kind gift of Dr Wang. Whereas it worked very well in the thorax and optic lobes, it did not reveal ectopic signal in abdominal glia.

For this revision, we obtained a new aliquot of anti-Dpn from Prof. Wang (Reviewer 1) and followed their protocol. This revealed excellent signal in the thorax, and confirmed ectopic Dpn signal in abdominal glia, which we could increase even further upon new genetic manipulations of insulin signalling. These data are provided in a new Figure 7.

Our findings are also supported by published data that we now cite:

1) TRAP-RNAseq data (Huang et al., 2015) showing that larval astrocyte glia express *dpn,* as well as many factors involved in neuroblast polarity, asymmetric cell division, asymmetric protein localisation and cell fate determination, neuroblast proliferation and neurogenesis.

2) Published after our original submission, single cell RNAseq data confirmed that LI glial cells can express *dpn, wor* and *ase* (Brunet Avalos et al., 2019).

3) Whilst revising our manuscript, Gangwani et al., 2020 reported that ectopic neuroblasts were induced in glioma models in *Drosophila*.

These references have now been cited.

2) Cell markers routinely used by many labs must work (Wor, Ase, Elav)

Antibodies to Wor, Ase and Elav worked well in our experiments, revealing standard neuroblasts in thorax and optic lobes (Wor and Ase) and neurons (Elav). However, they did not reveal ectopic expression in abdominal glia. To meet the requirements of the Editors, we repeated these experiments with a new aliquot of anti-Ase (gift from H. Wang), which again worked beautifully in thorax and optic lobes, but did not reveal ectopic expression in glia. This indicates that conversion of glia into neural stem cells does not follow the same processes as developmental neuroblasts.

We had previously shown excellent anti-Elav stainings (e.g. Figure 2C). However, when combining anti-Elav (raised in rat) with anti-Pros (raised in mouse), we originally did not realise they would cross-hybridise. For this revised version, we have used highly-cross adsorbed secondary antibodies, and provide new data in Figure 2H demonstrating that super-numerary Pros+ cells can have the neuronal marker Elav.

3) Dpn is a reagent that works well and should be clear.

As explained above, it seems it was not clear to reviewers that Dpn worked well in our experiments in developmental NBs in thorax and optic lobes. Our manuscript revealed ectopic abdominal Dpn expression in glia upon manipulating *ia-2*, insulin signalling and upon injury. This ectopic signal was lower than Dpn in thoracic NBs. To clarify this point, we now provide evidence of reliable Dpn signal in normal, developmental NBs in thorax in our original samples in new Figure 2I, Figure 3F,L, Figure 6A and Figure 6—figure supplement 1 and 2.

We have now further confirmed our data:

1) We have used a new anti-Dpn aliquot provided by reviewer 1, Prof. Wang, and used their protocol. This showed Dpn staining in developmental neuroblasts in thorax and optic lobes, in abdominal midline cells and low signal in some abdominal glia in controls. These are provided as new data in new Figure 7A,D,G.

2) We over-expressed in glia *dilp-6* alone and together with the activated forms of the downstream targets of insulin signalling – *ras* and *PI3K* – showing that they increased ectopic abdominal Dpn signal intensity and number of Dpn+ cells (using Wang anti-Dpn). That is, activation of insulin signalling in glia induces *dpn* expression and Dpn+ cell proliferation. These are provided as new data in new Figure 7A-O.

3) We have analysed publicly available RNAseq data, which demonstrate that glial cells can express neuroblast markers:

a) TRAP-RNAseq data of the LIII larval astrocyte glia showed they can express *dpn*, as wel as other neuroblast markers (Huang et al., 2015).

b) Single-cell RNAseq data of the LI larval CNS (published after our initial submission) showing that Repo+ glia can express Dpn, Wor and/or Ase (Brunet Avalos et al., 2019 ).

We conclude that the switch from a canonical glial cell state to that of expressing the neural stem cell marker *dpn* when insulin signalling is activated is an important discovery.

Please note that this decision letter does not guarantee that this manuscript will be accepted. At least one reviewer feels that their concerns were not addressed in the revised manuscript and that it is in everyone's best interest that the authors can prove that the inability to provide the appropriate expression patterns is not indicative of a deeper problem with the underlying hypothesis.

Reviewer 1 requested we carried out two experiments: (1) test whether over-expression of the *k_on_ intracellular domain (kon^ICD^)* in glia would alter the expression levels of *dilp-6* or *dpn.* (2) test whether *kon^ICD^* induced ectopic Dpn+ cells. We had originally tried those experiments, and the antibody stainings did not reveal ectopic Dpn+ cells, but the qRT-PCR data were variable. We have now repeated the qRT-PCR experiments again, which have revealed no significant differences in the levels of *dilp-6* nor *dpn* expression when kon^ICD^ is over-expressed in glia. Thus, now qRT-PCR data are consistent with stainings. We now provide these new data :

1) Over-expression of kon^ICD^ in glia does not induce ectopic Dpn+ cells

2) Over-expression of kon^ICD^ in glia does not alter dilp-6 nor dpn mRNA levels.

These data are provided in new Figure 6—figure supplement 1.

These data mean that when Kon^ICD^ translocates to the nucleus, it does not function directly as a transcription factor. It may function as a co-factor to an unknown transcription factor. Or it may participate in the cytoplasm to nucleus import/export machinery, as we have previously shown that loss of k_on_ function causes perinuclear distribution of Repo, preventing glial differentiation. We conclude that the regulation of *dilp-6* expresion by k_on_ is indirect, and have added this in the manuscript text.

It is in everyone's best interest that the authors can prove that the inability to provide the appropriate expression patterns is not indicative of a deeper problem with the underlying hypothesis.

Please see our response above. Our original findings have now been confirmed by:

1) New experiments with anti-Dpn from H. Wang, using their protocol, in Figure 7.

2) Further increase in abdominal ectopic Dpn levels and cell number by over-expressing *dilp-6* together with activated *ras* and *PI3K*, in new Figure 7.

3) Analysis of published RNAseq data confirmed that Repo+ glial cells can express *dpn, wor* and/or *ase* (Huang et al., 2015; Brunet Avalos et al., 2019).

1) The main conclusion of the paper (glia transform into NSCs which produce neurons) but is not supported by data: only one NSC marker used out of many available.The authors tried two additional NSC markers but did not observe staining, despite these reagents working for many labs in many publications. "We did not consider these results satisfactory enough to present."This is a major flaw, especially how unusual the Dpn staining looks like in the ectopic Dpn+ cells (very speckly). Failure to show additional NSC markers very concerning is a real issue; also no evidence for asymmetric cell division at mitosis (a hallmark of these NSCs).

Please allow us to break down the response to these multiple points:

The comment that “only one NSC” is insufficient evidence that “glia transform into NSCs which produce neurons” is a subjective opinion of this reviewer not grounded on evidence. We now provide RNAseq evidence from analysis of published data that larval CNS glia can express *dpn, wor* and/or *ase,* as well as other neuroblast development genes (see Huang et al., 2015; Brunet Avalos et al., 2019).

The response *“*The authors tried two additional NSC markers but did not observe staining, despite these reagents working for many labs in many publications. This is a major flaw, especially how unusual the Dpn staining looks like in the ectopic Dpn+ cells (very speckly) “, either bears an unjust implication or is a misunderstanding. We could visualise excellent, normal, canonical staining with anti-Dpn, anti-Wor and anti-Ase in thoracic and optic lobe neuroblasts. Instead, this work focuses in the abdominal ventral nerve cord of the late third instar lava, when there are normally no neuroblasts. i.e. no Dpn+, Wor+ nor Ase+ cells. No other labs had focused on nor reported ectopic Dpn signal in adbomen in the late larva before. Furthermore, in subsequent point 8 (see below) reviewers point out that all our Dpn data are good, except for (old) Figure 3D, contradicting the above statement that Dpn is *“very speckly”.* In any case, to clarify this point, we now provide the original data showing Dpn staining in thoracic NBs for comparison, including projections from either the entire VNC in new Figure 2I, Figure 6—figure supplements 1,2 and Figure 7A-C or projections from thorax (as with the thickness of the VNC a full projection loses resolution obscuring the signal) in new Figure 3F,L, new Figure 6A and new Figure 7D,E,F.

Upon genetic manipulation of insulin signalling, we could detect Dpn ectopic signal in abdominal glial cells, and this ectopic Dpn signal was not as strong as in normal neuroblasts. We could not detect clear, unambiguous ectopic abdominal Ase nor Wor.

To address the reviewer’s criticism, for this revision we have:

1) Repeated Wor, Ase and Dpn stainings, using new anti-Ase and anti-Dpn aliquots kindly donated by reviewer 1 (H. Wang), and using their protocol. We again obtained excellent stanings in thorax and optic lobes, and excellent ectopic abdominal Dpn signal, but no unambiguous ectopic abdominal signal for Wor nor Ase in glia. We conclude under our experimental conditions, converted glial cells express *dpn,* but do not reveal ectopic *wor* nor *ase.* We provide new Dpn data in new Figure 7.

2) Carried out further new tests that demonstrate that insulin signalling induces further *dpn* expression in glia and cell proliferation. We over-expressed in glia *dilp-6* alone and together with the activated forms of the downstream targets of insulin signalling – activated *ras* and *PI3K* – and showed that they increased ectopic abdominal Dpn signal intensity and number of abdominal Dpn+ cells. That is, activation of insulin signalling in glia induces *dpn* expression and Dpn+ cell proliferation, shown in new Figure 7 and in the text.

3) We analysed publicly available RNAseq data for the larval CNS which confirmed our findings:

a) TRAP-RNAseq data showed that larval astrocyte glia express *dpn,* as well as many factors involved in neuroblast polarity, asymmetric cell division, asymmetric protein localisation and cell fate determination, neuroblast proliferation and neurogenesis (Huang et al., 2015), now cited in the text.

b) Single cell RNAseq data show that some Repo+ glia express *dpn, wor* and *ase (*Brunet Avalos et al., 2019)(published after our first submission), now cited in the text.

Finally, the reviewer’s assessment that “….no evidence for asymmetric cell division at mitosis (a hallmark of these NSCs)” sounds like a subjective, biased preconception of what reprogrammed glial cells ought to look like. Instead, we provide new data evidence for this revision, including:

1) Additional evidence that Dpn+ glial cells can divide, as visualised with the midline glial marker Wrp, plus Dpn and S-phase marker PCNAGFP (new Figure 9I-J’) and mitotic marker anti-pH3 (new Figure 9K,K’), and mentioned in the text.

2) Evidence that over-expression of activated *ras^V1^*^2^ and *PI3K* together with *dilp-6* in glia increased the number of abdominal Dpn+ cells up to 3-fold, meaning these cells divided, shown in new Figure 7H,I,K,L,N,O and the text.

3) Furthermore, TRAP-RNAseq data showed that larval astrocyte glia express as well as *dpn,* also multiple factors involved in neuroblast polarity, asymmetric cell division, asymmetric protein localisation and cell fate determination, neuroblast proliferation and neurogenesis (Huang et al., 2015), now cited in the text.

2) There is no evidence for proliferation of the ectopic Dpn+ cells. The authors state that ectopic Dpn+ cells expressed the S phase marker PCNA:GFP and can be labeled with the mitotic marker pH3.However, only panes 8A-C show PCNA^+^ Dpn+ cells, which are increased following dilp-6 overexpression. No data in the figure shows ectopic Dpn+ cells that are pH3. The rest of the figure shows glial markers and PCNA or pH3, which is irrelevant to the question of whether ectopic Dpn+ cells can divide.

In response to this criticism, we have now improved these data. We have removed old Figure 9I,L and we have carried out new experiments and added these new data:

1) Over-expression of *dilp-6* and *ia-2* induced proliferation of Wrp+ Dpn+ cells, visualised as colocalisation of Dpn, the midline glia marker Wrp, the NSC marker Dpn and the S-phase marker PCNAGFP. Now shown in new Figure 9I,J’.

2) Over-expression of *ia-2* induced mitosis of Wrp+ Dpn+ cells, as visualised with the midline glia marker Wrp, the NSC marker Dpn and the mitotic marker anti-phospho-Histone-H3. Shown in new Figure 9K,K’.

3) Over-expression of activated *ras^V1^*^2^ and *PI3K* together with *dilp-6* in glia increased the number of abdominal Dpn+ cells up to 3-fold, meaning these cells divided, shown in new Figure 7H,I,K,L,N,O and the text.

These data demonstrate that over-expression of *dilp-6* and *ia-2* induce Dpn in glia and cell proliferation.

3) To show evidence that ectopic Dpn+ cells produce neuronal progeny, the authors used the pros-Gal4 line to drive flybow expression, and observed a small cluster of cells that included one Dpn+ and one Elav+ cell. As the authors say "this does not prove these cells are related by lineage, but is consistent with it."This does not show Dpn+ cells are producing neurons.

Correct, our statement accurately reflected our data. The fact that ectopic Dpn+ cells of glial origin can divide and generate neurons to a rather limited extent in our experimental conditions has been reinforced with new data showing that altering *ia-2* levels resulted in supernumerary Pros+ Elav+ cells (shown in new Figure 2H) and increased Dilp-6 and Ia-2 levels induced proliferation of Wrp+ Dpn+ cells (new Figure 9I-K’).

4) The authors also used "flip out" genetics to permanently mark glial cells.The genetics shown in the figure, legend, and reviewer response will not specifically label glia. The genotype is: actGAL4>y+>UASGFP/UAS-FLP; repoGAL4/Dilp-6. This would induce Flp widely, in all cells due to ubiquitous expression of actin-gal4. Most likely, the authors wrote down the wrong genotype in the figure, legend, Materials and methods, and reviewer response – it is probably actin promotor-FRT-stop-FRT-GFP. They cite Table 1 for more information on genotypes but there is no Table 1 provided.

We would like to thank this reviewer for spotting this typing error. The correct genotype is actin>y+STOP>GAL4 UASGFP/UASFLP; repoGAL4/UASdilp-6, and was correctly written in the manuscript text in the previous submissions, but indeed wrongly written in the figure legend. It was correctly abbreviated in the figure, as y+STOP, separating the actin promoter from the GAL4 cDNA, had been flipped-out. Nevertheless, we have now relabelled Figure 10C to remove any ambiguity, and we have corrected the genotype in the figure legend.

Supplementary file 1 had been submitted, and a revised version is now submitted too.

5) In order to call k_on_ and ia-2 partners, a direct physical interaction should be shown. The authors could not get the biochemical experiments to work for various reasons. Changed text from "partners" to "functional neuronal partner."The continued use of “partner” is inappropriate. The most accurate description of their relationship is that they show “genetic interactions” – so the first results header should be changed from "Ia-2 is a functional partner of Kon" to "Ia-2 and k_on_ show genetic interactions."

Whether to call it “genetic interaction” or “functional partner” depends on whether one stresses the experiment carried out to reveal a functional relationship or the meaning of the uncovered relationship. Nevertheless, we have modified this expression in the manuscript.

6) Saying ectopic Pros+ cells are GMCs or neurons is premature and can be definitively resolved by staining for Wor or Dpn (neuroblast-specific), Ase (neuroblast and GMC), and Elav (neurons). All have been extensively used by many labs. The authors could not get the stains to work.This is unsatisfactory.

Please allow us to correct that we could get anti-Dpn, anti-Wor and anti-Ase to work as well as other labs have shown, in canonical, developmental NBs in thorax and abdomen. No other labs reported ectopic signal in abdomen in late larvae before, therefore it is not possible to compare our data with those of other labs for this context. In response to this criticism, we now provide evidence of thoracic Dpn signal from our original samples, shown in new Figure 2I, Figure 3F,L, Figure 6A, Figure 6—figure supplements 1 and 2, and new Figure 7A-F.

Unfortunately, we had not realised that anti-Elav (raised in rat) and anti-Pros (raised in mouse) would result in non-specific cross-hybridisation using standard anti-mouse and anti-rat secondary antibodies. For this revised version, we used highly-cross adsorbed secondary antibodies, which revealed clear, specific co-localisation of Elav and Pros in some small cells in *ia-2* loss of function specimens, which are therefore neurons. We provide these new data in new Figure 2H.

7) The text says loss of ia-2 "destabilizes cell fate" – which is a vague term that obscures the phenotype. The authors changed text to "… upregulated GMC and NSC markers."They looked at Dpn but no other NSC marker, and Pros is not a specific GMC marker, also being expressed in neuropile glia near the midline (which is worrying).

The reviewer’s assumption that in vivo reprogramming of glial cells into neural stem cells ought to follow the same gene expression profile as normal developmental neuroblasts is not grounded on evidence but on pre-conception.

Regarding “other NSC markers”, a criticism that had already been raised in point 1, please see our response above. Our findings are also consistent with available RNAseq data, see response above.

Regarding the point that the reviewer finds worrying that glia express *pros*, it appears that the reviewer missed the point that Ia-2 is functionally related to Pros and that the conversion of glial cells occurs preferentially in Pros+ glial cells. This is mentioned in the text.

8) Dpn staining in Figure 3D is unconvincing; everything looks speckly. The authors state that Dpn staining is speckly in their hands.Many labs have used Dpn to mark neuroblasts, it is a very reliable reagent. The authors have good Dpn staining in other figures; this suggest to me that the ectopic Dpn+ cells are different from the normal Dpn+ NPCs, leading to different protein localization/levels. This concern is reinforced by the failure of the authors to show the ectopic Dpn+ cells express any other NSC marker.

The point that the reviewer considered our Dpn stainings speckly and the reference to “many labs” was already raised in point 1, please see our response above.

On the point “Dpn staining in Figure 3D is unconvincing; everything looks speckly… The authors have good Dpn staining in other figures”: Figure 3D was from an injured VNC at 74h AEL. No other stainings showed speckly Dpn staining, as stated by this reviewer. So that was one image, out of 25 experiments using anti-Dpn that had good Dpn staining, according to this reviewer. Thus, the reviewer makes an unfair generalisation on the quality of our work. May I point out that injured samples have to be handled with extreme care during stainings, so that they don’t fall apart, thus stainings are more challenging than normally, and younger injured larval VNCs are even more difficult to handle. Still, to satisfy the reviewer’s comment, we have now removed Figure 3D.

Regarding the reviewer’s point “this suggest to me that the ectopic Dpn+ cells are different from the normal Dpn+ NPCs, leading to different protein localization/levels”.

We did not claim that glial cells that express Dpn ectopically are identical to developmental *“normal Dpn+ NSC”*: that was a reviewer’s assumption. In fact, we claim the opposite: that regenerative neurogenesis does not involve developmental, canonical neural stem cells, but instead requires a neurogenic response from glial cells.

Dpn is the general NSC marker in *Drosophila*. We showed, and with the new data now confirm and reinforce, that genetic manipulation of Ia-2 and insulin signalling in glia increases *dpn* expression levels in glia, induces Dpn+ cell proliferation, increases Dpn+ cell number, and at least to a limited extent – constrained by the onset of pupariation – results in new neurons. We also showed that injury induces the expression of *ia-2* and ectopic Dpn+ cells. Animals that can regenerate their CNS upon injury do so by inducing neurogenesis not from “normal” NSCs, but from glia (Tanaka and Ferretti, 2009; Falk and Gotz, 2017). Our data show an important discovery that insulin signalling is involved in the injury-induced regenerative neurogenic response from glia. This important finding was discussed in the text in previous versions, and it seems the reviewer missed this important point.

To address the criticisms in point 8, we now show:

1) Dpn staining in thorax, for comparison, providing projections from either the entire VNC (including thoracic and abdominal) or for projections from thorax (as with VNC thickness resolution is lost with full projections obscuring the signal) in: new Figure 2I, Figure 3F,L, Figure 6A, Figure 6—figure supplement 1 and 2 and new Figure 7A-F.

2) Repeated Wor, Ase and Dpn stainings, using new aliquots kindly donated by reviewer 1 (H. Wang), and using their protocol. We again obtained excellent stanings in thorax and optic lobes for all, no ectopic abdominal signal for Wor nor Ase in glia, but good ectopic abdominal Dpn signal. The new stainings using the Wang anti-Dpn aliquot are shown in new Figure 7.

3) We have carried out further new tests that demonstrate that insulin signalling induces further ectopic abdominal *dpn* expression and increased Dpn+ cell number, implying proliferation. We over-expressed in glia *dilp-6* alone and together with the activate forms of the downstream targets of insulin signalling – *activated ras^V12^* and *PI3K* – and showed that they increased Dpn signal intensity and number of Dpn+ cells. That is, activation of insulin signalling in glia induces ectopic *dpn* expression and Dpn+ cell proliferation. These new data are shown in a new figure, new Figure 7.

4) We analysed publicly available RNAseq data for the larval CNS which confirmed our findings:

a) TRAP-RNAseq data showed that larval astrocyte glia express *dpn,* as well as many factors involved in neuroblast polarity, asymmetric cell division, asymmetric protein localisation and cell fate determination, neuroblast proliferation and neurogenesis (Huang et al., 2015), now cited in the text.

b) Single cell RNAseq data show that some Repo+ glia express *dpn, wor* and *ase (*Brunet Avalos et al., 2019)(published after our first submission), now cited in the text.

9) Ectopic Dpn+ cells were not quantified due to due to the disruption and variability of the abdominal crush procedure. The authors only counted the VNCs in which they could see ectopic Dpn+ (cells).Cherry-picking only VNCs that show ectopic Dpn+ cells is inappropriate.

This comment is incorrect. The ectopic Dpn+ cells were quantified in all our experiments, including all injury experiments (see graphs in current Figure 3D,I,J,O,P). It would appear that the Reviewer’s comment refers to the original submitted version (October 2019), in which we had not provided the quantifications upon injury. For the first revised version (May 2020), more experiments were carried out, and all the ectopic Dpn+ cells were counted and the data were provided in graphs.

Data were not cherry-picked. Instead, perhaps the reviewer is not familiar with genetic analysis or perhaps our data were not explicit enough for non-specialists. The question addressed was whether injury could induce ectopic abdominal Dpn+ cells, and if so, how many abdominal ectopic Dpn+ cells were produced. Standard genetic practice is to analyse phenotypes according to their penetrance and expressivity. The penetrance is the frequency with which a phenotype is manifested within a population. In our case, phenotypic penetrance was the percentage of injured VNCs that had ectopic abdominal Dpn+ cells. Not all injured samples had ectopic abdominal Dpn+ cells and some injured VNCs were severely damaged causing widespread cell loss. We had previously provided all penetrance data in the figure legend. The expressivity is the severity of the phenotype in individuals that manifest a given phenotype. In our case, expressivity was the incidence of ectopic Dpn+, and whether these cells were at the lesion site or also at a distance. Expressivity was provided in the graphs as the number of Dpn+ cells surrounding the lesion site, and the total number of abdominal Dpn+ cells, which we included to account for potential long-range effects away from lesion site. In order to make this point clearer to non-geneticists, we have now included the phenotypic penetrance as a percentage below the images in revised Figure 3G,H,J,M,N, and the expressivity above the graphs, in revised Figure 3D,I,J,O,P.

10) In response to InR-Gal4 expression concerns, the authors state "we do not know whether (InR-gal4) represents the endogenous expression pattern". It labels sparse patterns of neurons and sporadic glial cells.The authors directly state in the revised manuscript "we visualized InR expression using available GAL4 lines to drive his-YFP" but in the reviewer response they acknowledge this is not accurate.

Our statements in manuscript and response to reviewers were both accurate. It is worth pointing out that articles on the function of insulin signalling in the reactivation of developmental neuroblasts published in Cell and Nature (Chell and Brand, 2010 and Sousa-Nunes et al., 2011), whose claims depended on neuroblasts expressing the insulin receptor, did not demonstrate that *InR* was in fact expressed in those cells.

Anyway, to address this criticism, we have now analysed RNAseq data by Avalos et al. 2019 *eLife* 8: e50354, published after our manuscript was submitted, which show that 97/152 Repo+ cells in the LI CNS express the InR, validating our findings. We now cite this in the text.